# Weierstrass Positional Encoding for Vision Transformers

## Abstract

Vision Transformers (ViTs) have demonstrated remarkable success in computer vision tasks. However, their reliance on learnable one-dimensional positional encoding disrupts the inherent two-dimensional spatial structure of images due to patch flattening. Existing positional encoding approaches lack geometric constraints and fail to preserve a monotonic correspondence between Euclidean spatial distances and sequential index distances, thereby limiting the model's capacity to leverage spatial proximity priors effectively. Recognizing that periodicity is particularly beneficial for positional encoding, we propose Weierstrass elliptic Positional Encoding (WePE), a mathematically principled approach that encodes two-dimensional coordinates in the complex domain. This method maps the normalized two-dimensional patch coordinates onto the complex plane and constructs a compact four-dimensional positional feature based on the Weierstrass elliptic function $\wp(z)$ and its derivative. The doubly periodic property of $\wp(z)$ enables a principled encoding of 2D positional information, while their intrinsic lattice structure aligns naturally with the geometric regularities of patch grids in images. Their nonlinear geometric characteristics enable faithful modeling of spatial distance relationships, while the associated algebraic addition formula allows relative positional information between arbitrary patch pairs to be derived directly from their absolute encodings. WePE is a plug-and-play, resolution-agnostic positional module that integrates seamlessly with existing ViTs. Extensive experiments demonstrate that WePE delivers consistent performance gains in most scenarios, while its implementation with precomputed lookup tables ensures that these improvements incur no noticeable computational or memory overhead. In addition, several analyses and ablation studies bring further confirmation to the effectiveness of our method.

## 1 INTRODUCTION

Vision Transformers (ViTs) (Dosovitskiy et al., 2021) have recently emerged as a powerful representation learning architecture in computer vision, challenging the long-standing dominance of Convolutional Neural Networks (CNNs) (LeCun et al., 1998). By partitioning an image into a sequence of patches, ViTs leverage the self-attention mechanism to model global dependencies (Vaswani et al., 2017), a stark contrast to the localized inductive biases inherent in CNNs (Zeiler & Fergus, 2014). While this design enables greater flexibility in capturing long-range interactions, it also introduces a critical limitation: ViTs lack an intrinsic understanding of spatial geometry. As a result, their performance heavily relies on positional encodings (PE) (Shaw et al., 2018; Parmar et al., 2018) to provide the necessary spatial information.

The standard formulation of ViTs adopts simple, learnable 1D positional embeddings (Dosovitskiy et al., 2021). Beyond learnable absolute encodings, researchers have proposed and adopted a spectrum of positional encodings, including sinusoidal (trigonometric) schemes (Vaswani et al., 2017), Fourier Position Embedding (FoPE) (Hua et al., 2025), Rotary Position Embedding (RoPE) (Su et al., 2021a), Lie-group–based rotational encodings (LieRE) (Ostmeier et al., 2025), the RoPE-Mixed variant specialized for 2D vision (Heo et al., 2024), and others. However, most positional encoding schemes for ViTs entail a structural limitation: the 2D patch grid is serialized by flattening into a 1D token sequence to conform to the sequence-based formulation of Transformer self-attention (Vaswani et al., 2017), thereby disrupting the image's intrinsic spatial geometry. For instance, the sequential distance between vertically adjacent patches becomes artificially inflated compared to horizontally adjacent

ones. More importantly, such encodings operate essentially as a lookup table without geometric constraints. As a consequence, no monotonic correspondence is ensured between the true Euclidean distance of image patches and their relative positions in the embedding space (Wu et al., 2021). This deficiency severely limits the model's capacity to exploit spatial proximity priors, which acts as a cornerstone for effective visual understanding (Cordonnier et al., 2020).

To overcome these limitations, we introduce the Weierstrass elliptic Positional Encoding (WePE), a mathematically principled framework rooted in the theory of complex analysis. Instead of flattening the patch grid, we map the 2D coordinates of each patch directly onto the complex plane, thereby preserving their geometric integrity. We then utilize the Weierstrass elliptic function (Weierstrass, 1854), $\wp(z)$, a doubly periodic meromorphic function (As given in Definition B.6), to construct a continuous and structured spatial representation. It establishes a profound connection between a complex torus (defined by a lattice in the complex plane) and an algebraic elliptic curve through a specific differential equation. Formally, $\wp(z)$ possesses several mathematical properties, such as being a doubly periodic meromorphic function (as detailed in Theorem B.5), satisfying a specific differential equation (Theorem B.8), and adhering to a distinctive addition formula (Theorem B.11). From a three-dimensional perspective, The Weierstrass $\wp$-function resembles an array of identical volcano-like structures arranged regularly across the plane, where the sharp peak of each "volcano" corresponds to a second-order pole of the function. These peaks rise steeply to infinity at the lattice points, while the surface between them exhibits smooth, doubly periodic undulations. Each "volcano" is embedded within the parallelogram cell spanned by its two fundamental half-periods $\omega_1$ and $\omega_3$, which together form the period lattice of the Weierstrass elliptic function. This choice is not arbitrary: Our discussion in Section E concludes that periodicity is beneficial, and even optimal under the standard criterion of translation-equivariant positive definite attention kernels, for positional encodings. This insight leads us to select a function with a doubly periodic function to adapt to two-dimensional images. The intrinsic lattice structure of $\wp(z)$ naturally aligns with the geometric regularities of image patch grids, while its continuity ensures resolution invariance, a pivotal advantage for fine-tuning across resolutions. Moreover, we demonstrate that WePE possesses a provable distance-decay property, and that its algebraic addition formula allows the direct derivation of relative positional information between arbitrary patches.

This paper presents the design, theoretical grounding, and empirical validation of WePE. By replacing heuristic positional encoding with a mathematically principled formulation, our method endows ViTs with a robust geometric inductive bias. The main contributions of this paper are summarized as follows:

1. **Geometrically principled positional encoding:** We propose WePE, a mathematically grounded framework that maps 2D image coordinates to the complex plane via the Weierstrass elliptic function. This design preserves the intrinsic spatial structure of images, inherently aligns with translational regularities, and provides a continuous, resolution-invariant positional representation.

2. **WePE with several key properties:** Mathematical analysis shows that WePE has some key properties, such as relative position modeling via the elliptic function's addition formula, the inherent distance-decay property, and the periodicity advantages that, under certain conditions, may even be optimal. Additionally, the continuous function evaluation ensures resolution-invariant fine-tuning, while industrial-level acceleration schemes simplify implementation.

3. **Empirical validations across multiple datasets:** We conduct experiments under the scenarios of pre-training and fine-tuning. The results demonstrate the empirical advantages of WePE, showing consistent improvements over existing models where positional encoding is the only variable. Besides, several ablations also corroborate the significance of WePE's intrinsic properties.

## 2 WEIERSTRASS ELLIPTIC POSITIONAL ENCODING

### 2.1 FOUNDATIONAL FRAMEWORK OF WEPE

**Coordinate System and Complex Plane Mapping.** The input images of ViTs (Dosovitskiy et al., 2021) often exhibit varying resolutions, resulting in different numbers of image patches

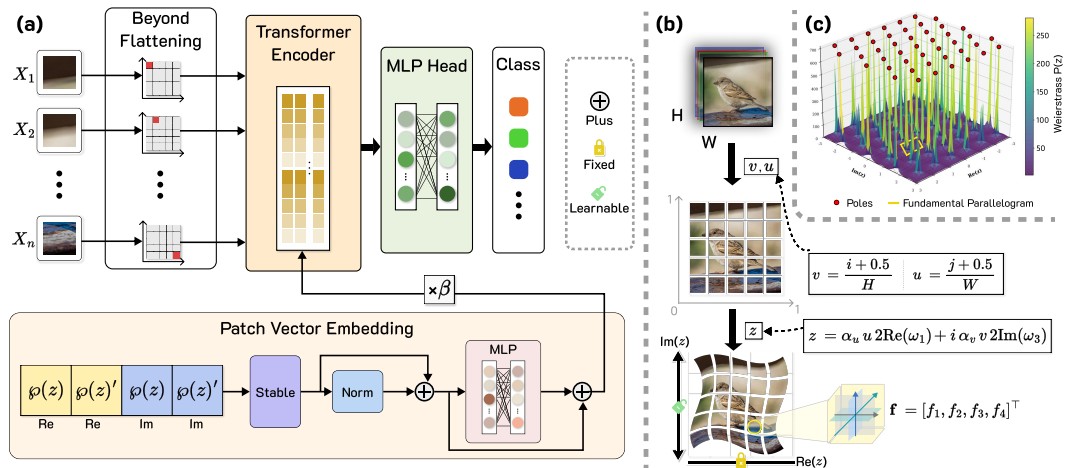

Figure 1: Overview of how WePE encodes 2D spatial information. (a) Four-dimensional WePE features are mapped to patch embeddings for Transformer encodings. (b) Image patches are normalized and mapped onto the complex plane to construct WePE coordinates. (c) A three-dimensional visualization of the Weierstrass elliptic function, illustrating its doubly periodic structure and pole distribution across the complex plane.

after partitioning. Consequently, the direct use of absolute indices is problematic, as their valid range changes with the total number of patches. In contrast, relative positional information, when represented within a normalized coordinate system, remains consistent and resolution-independent.

Given an input image, we conventionally partition it into a grid of $H \times W$ patches. For patch coordinates $(i, j)$ of the input image, where $i \in \{0, 1, \ldots, H-1\}, j \in \{0, 1, \ldots, W-1\}$, we first normalize them to the $[0, 1]$ interval:

$$u = \frac{j + 0.5}{W}, \qquad v = \frac{i + 0.5}{H}. \tag{1}$$

The effective ranges for $u$ and $v$ are respectively $\left(\frac{0.5}{W}, \frac{W-0.5}{W}\right)$ and $\left(\frac{0.5}{H}, \frac{H-0.5}{H}\right)$, which are proper subsets of $[0, 1]$. This is advantageous in certain situations. Taking $z = 0$ as an example, it is actually a pole in the mapping, and we want to avoid having the center of any patch located at a pole.

Subsequently, the normalized coordinates can be mapped onto the complex plane via the following formula:

$$z = \alpha_u \cdot u \cdot 2\mathrm{Re}(\omega_1) + i \cdot \alpha_v \cdot v \cdot 2\mathrm{Im}(\omega_3), \tag{2}$$

where $\alpha_u, \alpha_v$ are adjustable scaling factors, and $\omega_1, \omega_3$ represent the real half-period and imaginary half-period of the elliptic function, respectively. This mapping serves as a pivotal step, as it embeds the rich geometric and analytic properties of the Weierstrass elliptic function into the spatial representation of each image patch. Intuitively, the elliptic function acts like "weaving a fishing net" across the image, thereby naturally coupling positional information along both spatial dimensions.

**Feature Extraction from the $\wp(z)$ and its Derivative.** For the mapped complex coordinate $z$, to fully utilize the information in complex numbers, we extract the real and imaginary of $\wp(z)$ and its derivative, given below:

$$f_1 = \mathrm{Re}(\wp(z)), \quad f_2 = \mathrm{Im}(\wp(z)), \quad f_3 = \mathrm{Re}(\wp'(z)), \quad f_4 = \mathrm{Im}(\wp'(z)). \tag{3}$$

This yields a four-dimensional positional feature, denoted by $\mathbf{f} = [f_1, f_2, f_3, f_4]^\top$.

In certain regions, the Weierstrass elliptic function may attain extremely large magnitudes, especially near its poles (also illustrated in Figure 1(c)). Such unbounded behavior poses a risk for training stability, often manifesting as gradient explosion and computational failure (Olver et al., 2024). As a countermeasure, we instead propose two empirically validated and robust solutions (Section 2.2 and Section 2.3 ), which are respectively more suitable for the scenarios of pre-training and fine-tuning.

## 2.2 WePE Implementation for From-Scratch Pre-training

**Numerical Computation via Direct Lattice Summation.** When computing the series expansion of the Weierstrass elliptic function, we truncate it (Definition B.6) to a finite sum over indices $|m| \leq M, |n| \leq N$ (excluding the origin), where each lattice point is given by $\omega_{mn} = 2m\omega_1 + 2n\omega_3$. At this point, the truncated approximation of $\wp(z)$ is expressed as:

$$\wp(z) \approx \frac{1}{z^2} + \sum_{|m| \leq M, |n| \leq N, (m,n) \neq (0,0)} \left( \frac{1}{(z - \omega_{mn})^2} - \frac{1}{\omega_{mn}^2} \right). \tag{4}$$

For $w_{m,n}$ with a large modulus ($|w_{m,n}| \gg |z|$), the asymptotic contribution behaves as:

$$|T_{m,n}(z)| = \left| \frac{1}{(z - w_{m,n})^2} - \frac{1}{w_{m,n}^2} \right| \approx \frac{2|z|}{|w_{m,n}|^3} + O\left( \frac{|z|^2}{|w_{m,n}|^4} \right). \tag{5}$$

This indicates that the contribution decays as a power law of $|w_{m,n}|^{-3}$.

To improve convergence, we sort the lattice points by their modulus, such that $|\pi(w_1)| \leq |\pi(w_2)| \leq \cdots$. Here, $\pi$ denotes a permutation that reorders the lattice points according to the magnitude of their modulus. The reordered partial sum is then expressed as $S_K(z) = \sum_{k=1}^{K} T_{\pi(k)}(z)$, where $K$ is the truncation index. This ordering ensures that terms corresponding to lattice points with smaller modulus are accumulated first, significantly accelerating the convergence of the truncated series compared with the conventional lexicographic scheme. The truncation error can be bounded as:

$$|S_\infty(z) - S_K(z)| \leq \sum_{k=K+1}^{\infty} |T_{\pi(k)}(z)| \leq C \sum_{k=K+1}^{\infty} \frac{1}{|\pi(w_k)|^3}. \tag{6}$$

For a 2D lattice, the tail sum can be approximated by an integral, $\sum_{|w|>R} |w|^{-3} \sim \int_R^\infty r^{-2} dr = 1/R$, which ensures that the error decays as $O(1/R_K)$, where $R_K = |\pi(w_K)|$. Compared with a conventional lexicographic ordering, this modulus-based summation significantly improves convergence, reducing the truncation error from $O(\log K/\sqrt{K})$ to $O(1/\sqrt{K})$.

Finally, the upper bound of the truncation error can be estimated as:

$$E_{\text{trunc}} = \left| \sum_{|w| > R_{\max}} T_w(z) \right| \leq \sum_{|w| > R_{\max}} \frac{2|z|}{|w|^3} \leq \frac{2|z|}{R_{\max}^2}. \tag{7}$$

Taking $M_{\text{search}} = N_{\text{search}} = 12$ as an example, we can get that $R_{\max} \approx 30$, leading to an error of $E_{\text{trunc}} \lesssim 10^{-3}|z|$.

Our discussion in Section F presents a solution that balances computational precision and implementation efficiency through the pre-computation of a high-resolution WePE look-up table. Once training is complete, the optimal parameters learned by the proposed WePE module are fixed. We then generate a high-resolution look-up table by calculating a four-dimensional positional feature vector for each point on a fine, fixed-size grid. During the inference stage, the normalized coordinates of the patches for any given input image are used as query points. During inference, the GPU utilizes hardware-accelerated bilinear interpolation to efficiently approximate and retrieve the corresponding positional encodings from the pre-computed table. This method transforms the computational burden from complex, real-time calculations into a one-time pre-computation and a rapid, memory-based retrieval operation, thus bringing the time complexity of online inference down to a level comparable to that of simple grid-based encoding schemes. We have shown in Section F that the error introduced by this scheme is negligible, as it is far below the inherent stochastic error sources during deep neural network training and inference.

**Numerical Stability and Convergence Acceleration.** To mitigate potential numerical explosion, we apply an adaptive tanh-based compression of each feature component in $\mathbf{f}$, *i.e.*, $\tilde{f}_i = \tanh(\alpha_{\text{scale}} \cdot f_i)$ for $i = 1, \ldots, 4$, where the scaling parameter is parameterized as $\alpha_{\text{scale}} = \text{softplus}(\alpha_{\text{raw}})$ for ensuring positivity. For the input $z$ near a pole (*i.e.*, $|z| < 15\epsilon$, where $\epsilon$ is a small threshold), the function value is clipped to a large constant $C_{\text{large}}$; otherwise, $\wp(z)$ is computed using Equation 4. The accumulated round-off error for a sum over $K$ terms is bounded by $E_{\text{round}} \leq K \cdot \epsilon_{\text{mach}} \cdot \max_k |T_k|$,

where $\epsilon_{\text{mach}} \approx 2.22 \times 10^{-16}$ is the machine precision. By further clipping individual terms such that $\max_k |T_k| \leq M_{\text{clip}}$, the total round-off error is controlled at the order of $10^{-10}$. These measures collectively ensure numerical stability, while preserving both model accuracy and the fidelity of the encoded features (Higham, 2002).

**Network Architecture Integration.** We use $\tilde{\mathbf{f}}_{ij}$ to denote the complete four-dimensional stabilized feature vector of the $(i, j)$-th image patch, which is composed of the components $\tilde{f}_1, \tilde{f}_2, \tilde{f}_3, \tilde{f}_4$. These 4D WePE features are then projected to the model dimension $d$ via a linear layer, yielding patch encodings $\mathbf{PE}_{ij} = \text{LayerNorm}(\mathbf{W}_{\text{proj}}\tilde{\mathbf{f}}_{ij} + \mathbf{b}_{\text{proj}})$ (Ba et al., 2016), where $\mathbf{W}_{\text{proj}} \in \mathbb{R}^{d \times 4}$ and $\mathbf{b}_{\text{proj}} \in \mathbb{R}^d$. For enhanced representational capacity, this linear layer can be substituted with a Multi-Layer Perceptron (MLP) (Rumelhart et al., 1986). The classification token is endowed with a separate learnable encoding $\mathbf{PE}_{\text{cls}} \in \mathbb{R}^{1 \times d}$ (Devlin et al., 2019). Finally, these position encodings are added to their corresponding patch and token encodings to form the model's input sequence:

$$\mathbf{X}_{\text{input}} = \big[\, \mathbf{x}_{\text{cls}} + \mathbf{PE}_{\text{cls}},\ \mathbf{x}_{00} + \mathbf{PE}_{00},\ \ldots,\ \mathbf{x}_{HW} + \mathbf{PE}_{HW} \,\big]^{\top}. \tag{8}$$

**Positional Encodings with Learnable Parameters.** To enable adaptive spatial scaling, the imaginary half-period $\omega_3$ is parameterized as a learnable quantity. Since the imaginary part $\omega_3'$ must be positive, it cannot be directly optimized using gradient-based methods. Instead, we introduce an unconstrained trainable parameter $\alpha_{\text{learn}}$ and define $\omega_3 = i \cdot \omega_3'$, where $\omega_3'$ is derived via the softplus function to ensure positivity: $\omega_3' = \text{softplus}(\alpha_{\text{learn}}) = \log(1 + \exp(\alpha_{\text{learn}}))$. The other lattice parameter, $\omega_1$, is kept as a fixed constant to prevent potential overfitting. This configuration allows the lattice basis, which is initialized to be orthogonal and form a square, to adaptively deform into a rectangle during training, thereby learning the optimal aspect ratio for the given data.

Recognizing that semantic and positional cues are not always of equal importance, we introduce a learnable global scaling factor, $\beta_{\text{pos}}$, to balance their contributions. The final positional encoding is then defined as $\mathbf{PE}_{\text{final}} = \beta_{\text{pos}} \cdot \mathbf{PE}_{\text{WePE}}$, where $\mathbf{PE}_{\text{WePE}}$ is the encoding generated by our method.

## 2.3 WePE Adaptation for Fine-tuning

Fine-tuning large pre-trained models, the available training epochs are typically much fewer than during pre-training. Moreover, as the model already encodes substantial prior knowledge, an overly strong injection of geometric information can be counterproductive, potentially leading to decreased stability and convergence speed. To circumvent these limitations, we adopt a fine-tuning strategy that is conceptually aligned with the original WePE framework but incorporates implementation adaptations tailored to rapid convergence.

For the periodic part of Equation 4, we found that its essential structure is a combination of a directional periodic oscillation, which can be modeled with sine and cosine functions, with an orthogonal exponential decay. Leveraging this property, we can construct a rapidly converging Fourier-like series (Stein & Shakarchi, 2003) to efficiently model the periodic behavior during fine-tuning.

The value $\wp(z)$ is approximated by a primary term handling the pole at the origin, complemented by a series of rapidly decaying correction terms:

$$\wp(z) \approx \frac{1}{|z|^2 + \beta} + \sum_{k=1}^{K} \frac{\gamma}{k^2} \left[ \cos(k\pi u') e^{-k\pi|v'|} + \sin(k\pi v') e^{-k\pi|u'|} \right], \tag{9}$$

where $u' = \text{Re}(z)/\omega_1$, $v' = \text{Im}(z)/\omega_3'$. Here, $\beta$ and $\gamma$ are learnable scalar parameters controlling numerical stabilization near the origin and the amplitude of the periodic corrections, respectively.

The numerical stability of this approximation is principally ensured by two key components. First, the term $\frac{1}{|z|^2 + \beta}$ replaces the singular term $\frac{1}{z^2}$. By introducing the small, positive, learnable parameter $\beta = \text{softplus}(\beta_{raw})$, the denominator is guaranteed to remain strictly positive, thus removing the singularity at $z = 0$ while preserving the function's asymptotic behavior. Second, the exponential decay factors $e^{-k\pi|v'|}$ and $e^{-k\pi|u'|}$ in the series ensure rapid convergence, enabling high precision with a small number of terms $K$. The truncation error is exponentially bounded, making this formulation both more stable and computationally efficient than a direct lattice summation, especially in gradient-based optimization scenarios. A more detailed derivation is provided in Appendix D.2.

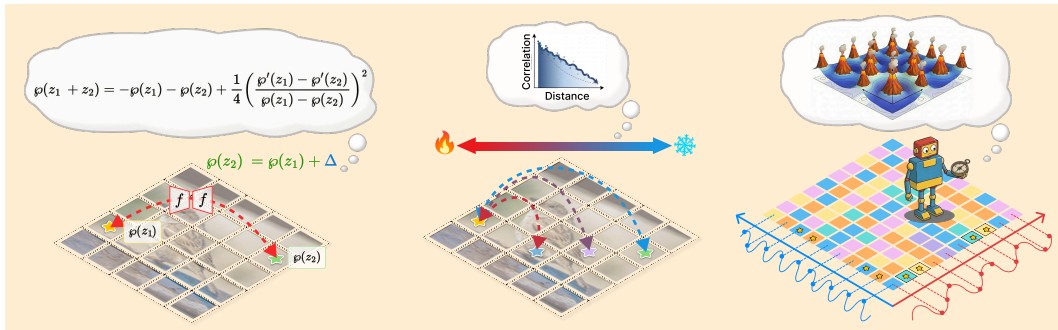

Figure 2: Diagram of key features of WePE. Left: WePE's addition formula enables explicit modeling of relative displacements on the patch lattice; Middle: The induced distance–aware correlation shows a decay with spatial separation, providing a built-in locality prior for attention; Right: The Weierstrass elliptic function forms a doubly periodic lattice on the complex plane, which induces a coherent two-dimensional periodic positional structure over the image grid.

## 2.4 THEORETICAL EXPLANATION

**Mapping Normalized Coordinates to the Complex Plane via Isomorphism.** Mapping 2 can therefore be written as $T : \mathbb{R}^2 \rightarrow \mathbb{C}, T(u,v) = c_1 u + i\, c_2 v, c_1 c_2 \neq 0$. To verify linearity, let $\mathbf{x}_1 = (u_1, v_1)$ and $\mathbf{x}_2 = (u_2, v_2)$ in $\mathbb{R}^2$ and $a, b \in \mathbb{R}$. Then $T(a\mathbf{x}_1 + b\mathbf{x}_2) = T(au_1 + bu_2, av_1 + bv_2) = a\, T(\mathbf{x}_1) + b\, T(\mathbf{x}_2)$, so $T$ is linear. To prove injectivity, observe that $T(u,v) = 0 \iff c_1 u + i\, c_2 v = 0$. Since $c_1, c_2, u, v \in \mathbb{R}$ and $c_1 \neq 0$, $c_2 \neq 0$, we must have $u = v = 0$, hence $\ker(T) = \{(0,0)\}$ and $T$ is injective. Because $\dim_{\mathbb{R}} \mathbb{R}^2 = \dim_{\mathbb{R}} \mathbb{C} = 2$, an injective linear map between equal-dimensional finite-dimensional real spaces is automatically surjective (rank–nullity). Therefore $T$ is bijective and thus an $\mathbb{R}$-linear isomorphism $\mathbb{R}^2 \cong \mathbb{C}$. Consequently, the spatial relationships among image patches are preserved when embedding normalized coordinates into the complex plane.

**Relative Position Modeling Based on Addition Formula.** A distinctive property of Weierstrass elliptic function is their addition formula. Let the absolute positions of two image patches be $z_i$ and $z_j$ in the mapped complex plane $\mathbb{C}$. Their relative position can be represented as $\sigma_z = z_j - z_i$. By applying the addition formula (see Theorem B.11), $\wp(z_j) = \wp(z_i + \sigma_z)$ can be expressed entirely in terms of $\wp(z_i)$, $\wp(\sigma_z)$, and their derivatives. This algebraic relationship enables direct derivation of relative positional information between any two points from their absolute encodings, without requiring additional relative position encoding modules. Within self-attention mechanisms, the interaction between query vector $Q_{z_i}$ and key vector $K_{z_j}$ can naturally utilize the $\wp(z_j - z_i)$ term from the addition

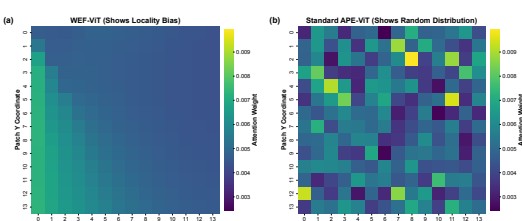

Figure 3: Comparison of geometric inductive bias between WePE-ViT and standard APE-ViT. (a) WePE-ViT exhibits structured locality-aware attention patterns with smooth, isotropic decay from the query patch. (b) Standard APE-ViT demonstrates, unstructured attention distribution that lacks spatial coherence.

formula. This design enables attention weights to inherently capture spatial relationships between positions, thereby enhancing the model's understanding of geometric structures. Compared to methods that require explicit computation of relative for all position pairs (Shaw et al., 2018), WePE achieves continuous, precise, and computationally efficient relative position representations.

**Long-term decay of WePE.** In self-attention, interaction strength is determined by the similarity between token representations, if the interaction strength after adding positional encodings does not exhibit a decaying trend with spatial separation, the model cannot differentiate local relationships from long-range ones. This contradicts the intrinsic structure of natural images, where nearby pixels and patches are far more correlated than distant ones. A built-in distance–decay profile therefore

supplies the model with a locality-aware inductive bias, enabling attention to prioritize meaningful local interactions while still permitting global reasoning when necessary. To meet this requirement, WePE naturally provides an inherent distance–decay property. We formalize as a theorem: *for any two patch positions with Euclidean distance $d$, the expected inner product of their encodings, $\mathbb{E}[\mathbf{p}_1^T \mathbf{p}_2] = S(d)$, is a strictly monotonically decreasing function for $d > 0$.* This property arises from the combination of the distance-preserving mapping from patch coordinates to the complex plane and the periodic structure of the Weierstrass elliptic function $\wp(z)$. Specifically, the final encodings are linear projections of 4D feature vectors constructed from $\wp(z)$ and its derivative (see Section 2.1). As the spatial distance between patches increases, their inner product exhibits a cosine-like decay, weakening the similarity between distant positions. This property ensures that the model is endowed with an explicit spatial proximity prior, which benefits a wide range of vision tasks (Vaswani et al., 2017). A detailed proof is provided in Appendix D.1.

**Resolution-Invariant Positional Encoding through Continuous Function Evaluation.** Fine-tuning ViTs typically increases input resolution to capture fine details (Dosovitskiy et al., 2021; Steiner et al., 2021). Discrete learnable encodings are tied to a fixed grid and do not transfer; bilinear or bicubic interpolation attenuates high frequencies, introducing boundary aliasing and distorting long-range geometry (Keys, 1981; Touvron et al., 2021b). As shown in Equation 2, WePE effectively overcomes these limitations through its formulation as a continuous meromorphic function evaluated at arbitrary complex coordinates rather than a discrete lookup operation. The scaling factors $\alpha_u$ and $\alpha_v$ control the effective spatial frequency of the elliptic function across horizontal and vertical dimensions respectively, enabling the encoding to maintain optimal spatial discrimination at the increased resolution while preserving the fundamental periodic structure that encodes translational regularities in visual data. The imaginary half-period parameter adapts during fine-tuning to accommodate changes in the aspect ratio between patches and the overall spatial density of the representation, ensuring that the doubly periodic lattice structure characterized by the fundamental parallelogram maintains geometric consistency across resolutions. The lattice summation in Equation 4 remains numerically stable across different resolutions since the truncation parameters $M$ and $N$ are determined by the desired numerical precision rather than the specific image dimensions, ensuring consistent computational accuracy regardless of the resolution scaling factor. This continuous formulation enables the generation of positional encodings at any spatial resolution without resorting to interpolation of pre-computed values, thereby preserving the mathematical precision and geometric fidelity of the spatial representation.

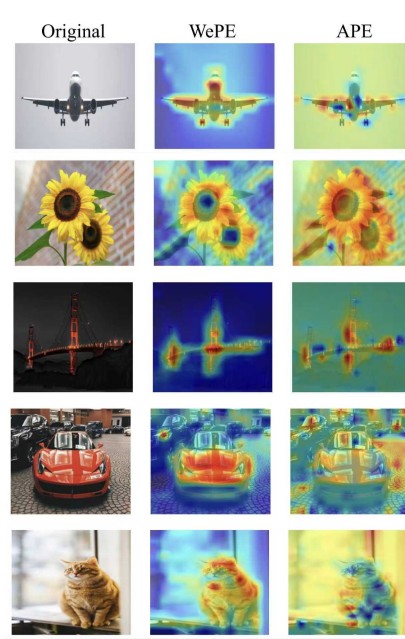

Figure 4: Attention rollout visualization comparing semantic focus patterns between WePE-ViTs and APE-ViTs models trained on CIFAR-100.

## 3 EXPERIMENTS

To evaluate the effectiveness of WePE, we conduct experiments under both pre-training and fine-tuning settings. We further perform ablation studies to assess the contribution of each module component, and provide empirical analyses to reveal the underlying mechanism and theoretical rationale of WePE. More supplementary experiments are provided in Appendix H.

### 3.1 UNDERSTANDING WEPE

**WePE exhibits better geometric inductive bias.** To investigate the inductive bias introduced by our proposed WePE, we first visualize self-attention maps of a randomly-initialized ViTs (Dosovitskiy et al., 2021), without any training. Specifically, we focus on the attention distribution originating

from a central query patch to all other patches within the sequence. As depicted in Figure 3, the WePE-equipped model exhibits a highly structured and localized attention pattern: attention weights are concentrated on the query patch itself and decay smoothly and isotropically with increasing spatial distance. In contrast, the baseline using standard learnable Absolute Positional Encoding (APE) (Dosovitskiy et al., 2021) shows a largely uniform and unstructured attention distribution, where attention weights appear randomly scattered. This demonstrates that WePE inherently provides a strong spatial locality prior, predisposing the model to focus on local interactions even before learning, whereas standard encodings lack such an inductive bias. These results collectively demonstrate that WePE injects a robust and accurate 2D geometric inductive bias into the ViTs (Dosovitskiy et al., 2021) from initialization. This structural prior is absent in standard models, which must learn spatial relationships from data alone.

**Global Semantic Attention in ViTs (Dosovitskiy et al., 2021).** We trained two ViT-Tiny models under identical settings. We then visualized the complete information flow from input to output on unseen high-resolution images using the Attention Rollout method. The results, presented in Figure 4, consistently demonstrate a significant qualitative difference in the learned attention patterns. For instance, when presented with an image of a cat, the WePE-ViTs model's attention forms a coherent and complete silhouette that accurately envelops the entire animal. In stark contrast, the APE-ViTs attention is fragmented, focusing disproportionately on high-contrast edges where the subject meets the background, rather than the semantic object itself. WePE learns to associate features within a global spatial context, resulting in attention maps that align closely with the primary semantic content. The baseline APE model (Dosovitskiy et al., 2021), lacking this structural prior, appears to overfit to low-level, local cues, leading to a fragmented attention mechanism that often fails to represent the complete semantic entity within the image. From these visualizations, we conclude that the geometric inductive bias inherent in our WePE enables the model to develop a more holistic and structurally-aware understanding of visual scenes.

**Long-term Attenuation of Positional Encoding.** We verify the distance–decay property of WePE on a $14 \times 14$ patch grid (from $224 \times 224$ images). For all $\binom{196}{2} = 19{,}110$ pairs we compute the normalized Euclidean distance $d_{\text{rel}} \in [0, 100]$ and the cosine similarity $S$ between encodings $(\mathbf{p}_i, \mathbf{p}_j)$, rescale $S$ by min–max for visualization (Han et al., 2011), and aggregate results into 80 distance bins, taking the bin midpoint as the representative distance and the mean similarity as the representative score (see Appendix H.6). The curve (see Figure 11) shows a pronounced negative correlation with distance ($\rho = -0.966$), evidencing long-range attenuation.

In practical applications of ViTs (Dosovitskiy et al., 2021), the self-attention mechanism depends not only on pure positional encodings, but more critically, on the fused representation of patch content features and their positional encodings. Similarly, to simulate a content-agnostic scenario and isolate the effect of the positional signal, we sample random content features $\mathbf{f}_{ij} \sim \mathcal{N}(\mathbf{0}, \mathbf{I})$ in $\mathbb{R}^{192}$ and fuse them with WePE via $\mathbf{h}_{ij} = \mathbf{f}_{ij} + \mathbf{p}_{ij}$, repeating the analysis yields the same distance–decay trend (see Figure 5).

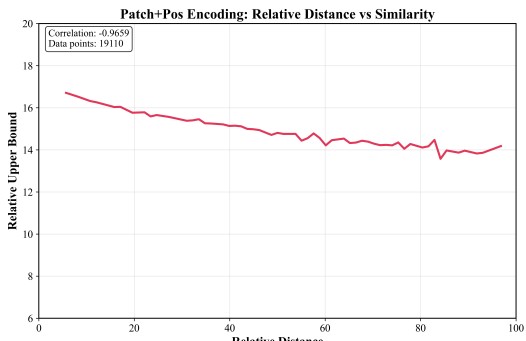

Figure 5: Empirical validation of the distance–decay property of WePE: x-axis denotes normalized patch spatial separation, y-axis denotes min–max scaled cosine similarity of positional encodings.

## 3.2 PRE-TRAINING

We begin with CIFAR-100 (Krizhevsky, 2009) and ImageNet-1k (Deng et al., 2009) benchmarks to evaluate WePE in 2D vision tasks. All models are based on the ViTs (Dosovitskiy et al., 2021) and trained from scratch. We compare WePE with APE (Dosovitskiy et al., 2021), RoPE-Mixed (Heo et al., 2024), Auto-Scaled 2D RoPE (AS2DRoPE) (Chu et al., 2024), Sinusoidal Position Encoding (Vaswani et al., 2017), FoPE (Hua et al., 2025), RoPE (Su et al., 2021b), and LieRE (Ostmeier et al., 2025) under identical configurations. To adapt RoPE and FoPE to 2D inputs, an image $I \in \mathbb{R}^{H \times W \times C}$ is patchified by a strided convolution (kernel = stride = $P$), each

Table 2: Top-1 accuracy (%) on CIFAR-100 and ImageNet-1k, trained for 200 epochs.

| Dataset | Fraction | **WePE (Ours)** | LieRE$_8$ | LieRE$_{64}$ | RoPE-Mixed | AS2DRoPE | APE |
|---|---|---|---|---|---|---|---|
| CIFAR-100 | 20% | **46.36** | 45.42 | 44.44 | 44.48 | 39.14 | 39.80 |
| CIFAR-100 | 40% | **56.81** | 54.68 | 54.64 | 55.14 | 50.53 | 49.90 |
| CIFAR-100 | 60% | **63.38** | 62.04 | 62.90 | 61.56 | 58.58 | 56.83 |
| CIFAR-100 | 90% | **68.96** | 67.72 | 68.36 | 67.00 | 62.59 | 62.76 |
| ImageNet-1k | 100% | **70.10** | 69.60 | 69.30 | 68.80 | 64.40 | 66.10 |

$P \times P$ patch is linearly projected and the resulting $H/P \times W/P$ grid is serialized into a 1D token sequence, on which the original sequence-based formulations are applied to the query/key vectors in multi-head self-attention to encode periodic, relative spatial structure. Table 1 and Table 2 compare the performance of ViTs (Dosovitskiy et al., 2021) integrated with different position encodings, trained from scratch for a varying number of epochs on multiple datasets. In these comparisons, WePE consistently demonstrates superior performance.

To further assess the proposed method, we integrated the WePE into a Dynamic Hybrid Vision Transformer Tiny (DHVT-Ti) model (Lu et al., 2022), which is engineered for data efficiency on smaller datasets. The DHVT model is specifically engineered to enhance the inductive biases of ViTs (Dosovitskiy et al., 2021) for improved

Table 1: CIFAR-100 (100% dataset), 120 epochs Top-1 accuracy (%).

| Method | **WePE (Ours)** | Absolute PE | RoPE | FoPE | Sinusoidal PE |
|---|---|---|---|---|---|
| Accuracy | **63.78** | 56.46 | 57.29 | 57.70 | 51.99 |

data efficiency on small-scale datasets by incorporating convolutional operations. This serves as an excellent baseline model for comparing the pre-training capabilities of various vision models. As shown in Table 3, the model achieves a peak validation accuracy of 76.53%, surpassing all baselines.

### 3.3 FINE-TUNING

To assess transferability and data efficiency, we fine-tuned an ImageNet-21k pre-trained ViT-L/16 on Visual Task Adaptation Benchmark 1000 (VTAB-1k) tasks under the 1k-shot protocol (Zhai et al., 2020). Inputs were resized to $384 \times 384$, requiring bilinear interpolation of the pre-trained positional encodings from a $14 \times 14$ to a $24 \times 24$ grid; rather than discarding these encodings, we formed a hybrid position module that blends the interpolated encodings with our WePE through a learnable gate $\lambda$ (see Appendix G). The experimental procedure is identical to that described in Dosovitskiy et al. (2021). The experimental results, as shown in Table 4, demonstrate that our algorithm outperforms traditional methods on most datasets.

On the full CIFAR-100 (Krizhevsky, 2009) dataset, we fine-tuned an ImageNet-21k (Deng et al., 2009) pre-trained ViT-B/16 while directly replacing the original learnable positional encodings with WePE. This configuration attains a peak test accuracy of $93.28\%$, substantially outperforming a strong baseline, indicating that WePE's continuous, doubly periodic spatial representation benefits fine-grained recognition; the learned parameters $\omega_2' \approx 1.085$ and $\beta \approx 0.610$ further evidence successful geometric adaptation to the dataset.

Table 3: Results on $224 \times 224$ resolution. All models are trained from scratch for 100 epochs under the same training schedule.

| Method | #Params | GFLOPs | Accuracy (%) |
|---|---|---|---|
| ResNet-50+$\mathcal{L}_{dr.loc}$ | 21.2M | 3.8 | 72.94 |
| SwinT+$\mathcal{L}_{dr.loc}$ | 24.1M | 4.3 | 66.23 |
| CvT-13+$\mathcal{L}_{dr.loc}$ | 19.6M | 4.5 | 74.51 |
| T2T-ViT+$\mathcal{L}_{dr.loc}$ | 21.2M | 4.8 | 68.03 |
| DHVT-T | 6.0M | 1.2 | 74.78 |
| **WePE (Ours)** | **5.5M** | 1.6 | **76.53** |

### 3.4 ABLATION

To assess the contribution of each component in our WePE, we conducted ablation experiments on CIFAR-100 (Krizhevsky, 2009) with ViT-Ti. The baseline achieves 63.78%. Removing $\wp'(z)$ and using only $\text{Re}(\wp(z)), \text{Im}(\wp(z))$ lowers accuracy to 63.08% ($-0.70$ %), confirming the derivative provides essential gradient cues. Fixing $\alpha_{\text{scale}}$ and $\alpha_{\text{learn}}$ yields 62.88% ($-0.90\%$), showing the

Table 4: Performance breakdown on selected VTAB-1k tasks.

| | Caltech101 | CIFAR-100 | DTD | Flowers102 | Pets | Sun397 | SVHN | Camelyon | EuroSAT | Resisc45 | Retinopathy | Clevr-Count | Clevr-Dist | DMLab | dSpr-Loc | dSpr-Ori | KITTI-Dist | sNORB-Azim | sNORB-Elev | Mean |
|---|---|---|---|---|---|---|---|---|---|---|---|---|---|---|---|---|---|---|---|---|
| APE | 90.80 | 84.10 | 74.10 | 99.30 | 92.70 | 61.00 | 80.90 | 82.50 | 95.60 | 85.20 | 75.30 | 70.30 | 56.10 | 41.90 | 74.70 | 64.90 | 79.90 | 30.50 | 41.70 | 72.70 |
| WePE (Ours) | 91.32 | 87.59 | 77.41 | 98.79 | 93.16 | 64.30 | 84.58 | 83.73 | 93.89 | 86.10 | 77.15 | 73.81 | 60.91 | 54.24 | 75.18 | 68.10 | 81.25 | 34.11 | 42.01 | 73.59 |

necessity of adaptive scaling and lattice adjustment. Using non-lemniscatic invariants results in 63.20% (−0.58%), indicating robustness but also the superiority of the lemniscatic square lattice. Fixing the global scaling parameter to unity produces the largest drop, 62.60% (−1.18%), highlighting the need for adaptive control of positional strength. Overall, the consistent yet modest degradations across all settings demonstrate that while each component enhances performance, the primary advantage arises from the holistic geometric prior imparted by the Weierstrass elliptic function, seamlessly integrated into the ViTs (Dosovitskiy et al., 2021) backbone.

## 4 CONCLUSION

In this work, we introduce Weierstrass elliptic Positional Encoding, a mathematically principled approach that leverages the rich structure of elliptic functions to address spatial representation limitations in Vision Transformers. Our method preserves 2D spatial relationships through a direct complex domain mapping and provides explicit spatial proximity priors via a theoretically guaranteed distance-decay property. We demonstrated the effectiveness of WePE through extensive experiments, achieving superior performance in most from-scratch training and fine-tuning scenarios across a variety of standard benchmarks. Furthermore, empirical analyses are conducted to investigate the underlying factors contributing to the superiority of WePE. Additionally, we provide a rigorous exposition and derivation of the core mathematical principles underpinning WePE. In summary, our proposed WePE offers a plug-and-play, resolution-agnostic positional module that restores the 2D geometric inductive bias with negligible computational and memory overhead, making it a practical drop-in replacement for existing encodings in ViTs.

## REPRODUCIBILITY STATEMENT

All theoretical results are established under explicit assumptions, with complete proofs provided in the Appendix. The code will be released upon acceptance.

## ETHICS STATEMENT

This work only uses publicly available datasets and does not involve human subjects or sensitive information. We identify no specific ethical concerns.

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

## APPENDIX CONTENTS

# APPENDIX

## A  USE OF LARGE LANGUAGE MODELS

An AI language model was used only for linguistic editing (grammar, wording, stylistic consistency) and translation checks. It was applied to the *Abstract*, *Introduction* and descriptive text in *Experiments*, *Conclusion*. The AI was not used for methods, derivations, theoretical appendices, algorithm design, code, or quantitative results. All technical content and analyses are solely the authors' work, and any AI-edited text was reviewed and finalized by the authors.

## B  SUPPLEMENTARY BACKGROUND KNOWLEDGE

Mainstream explicit function-based positional encodings for Vision Transformers (Dosovitskiy et al., 2021) that are added to patch embeddings can be broadly grouped into three families: (i) learnable absolute positional embeddings (Dosovitskiy et al., 2021), (ii) Sinusoidal Position Encoding (Vaswani et al., 2017) and their higher-order Fourier extensions (e.g., FoPE (Hua et al., 2025)), and (iii) RoPE (Su et al., 2021b) and their complex-valued or group-theoretic extensions (e.g., LieRE (Ostmeier et al., 2025), RoPE-Mixed (Heo et al., 2024), Multimodal Rotary Position Embedding (M-RoPE) (Bai et al., 2023). Indeed, according to Euler's formula, the foundation of RoPE (Su et al., 2021b) still lies in Sinusoidal Position Encoding (Vaswani et al., 2017). Its essence resides in leveraging Euler's formula to map positional information into vector rotations, making RoPE (Su et al., 2021b) fundamentally a geometric extension of the traditional sinusoidal encoding. However, WePE is conceptually distinct from all two or three families above, even though it is also built upon periodic and complex-valued functions. To our knowledge, WePE is the first 2D positional encoding scheme designed for ViTs (Dosovitskiy et al., 2021) that is genuinely constructed on the complex plane. Rather than composing multiple 1D sinusoidal bands along the flattened token index or applying block-wise rotary phases in the hidden space, WePE is formulated as a genuinely 2D positional function on the complex plane: it maps the 2D patch lattice to a complex lattice and evaluates a Weierstrass elliptic function with an intrinsic doubly periodic structure. As a result, the 2D geometry of the image grid is an inherent property of the encoding itself, rather than an artifact of the 1D serialization or separable 1D Fourier bases. Through the elliptic addition formula, absolute positional codes in WePE are algebraically linked to relative displacements, providing a tight coupling between absolute and relative position at the function level rather than relying solely on phase differences in Fourier space. This makes WePE the "fourth approach", standing apart from the three mainstream positional encoding schemes currently available.

In this part, we supplement with further necessary preliminaries and the corresponding proofs of theorems regarding the Weierstrass elliptic function (Weierstrass, 1854).

**Definition B.1** (Meromorphic Function). *Let $D \subset \mathbb{C}$ be an open set. A function $f : D \to \mathbb{C} \cup \{\infty\}$ is called meromorphic in $D$ if $f$ is analytic everywhere in $D$ except at finitely many isolated singularities, and each singularity is a pole.*

The Cauchy integral formula is one of the core tools in complex analysis:

**Theorem B.2** (Cauchy Integral Formula). *Let $f(z)$ be analytic on a simple closed curve $C$ and its interior, and let $z_0$ be a point inside $C$. Then:*

$$f^{(n)}(z_0) = \frac{n!}{2\pi i} \oint_C \frac{f(z)}{(z - z_0)^{n+1}} dz \tag{10}$$

Based on the Cauchy integral formula, we can derive Liouville's theorem:

**Theorem B.3** (Liouville's Theorem). *Any bounded entire function must be constant. That is, if $f(z)$ is analytic everywhere on the complex plane $\mathbb{C}$ and there exists a constant $M > 0$ such that $|f(z)| \leq M$ for all $z \in \mathbb{C}$, then $f(z)$ is constant.*

*Proof.* By the Cauchy integral formula, for any $z_0 \in \mathbb{C}$ and $r > 0$:

$$|f'(z_0)| \leq \frac{1}{r} \sup_{|z-z_0|=r} |f(z)| \leq \frac{M}{r} \tag{11}$$

As $r \to \infty$, $\frac{M}{r} \to 0$, hence $|f'(z_0)| = 0$, which implies $f'(z_0) = 0$.

Since $z_0$ is arbitrary, $f'(z) \equiv 0$ holds throughout $\mathbb{C}$. Let $f(z) = u(x,y) + iv(x,y)$. By the Cauchy-Riemann equations:

$$\frac{\partial u}{\partial x} = \frac{\partial v}{\partial y} = 0 \tag{12}$$

$$\frac{\partial u}{\partial y} = -\frac{\partial v}{\partial x} = 0 \tag{13}$$

This implies that all partial derivatives of $u(x,y)$ and $v(x,y)$ are zero, therefore $u$ and $v$ are both constants, and consequently $f(z)$ is constant.

$\square$

**Definition B.4** (Period Lattice). *Let $\omega_1, \omega_2 \in \mathbb{C}$ be linearly independent (i.e., $\frac{\omega_2}{\omega_1} \notin \mathbb{R}$). The period lattice is defined as:*

$$\Lambda = \{2m\omega_1 + 2n\omega_2 : m, n \in \mathbb{Z}\} \tag{14}$$

*where $2\omega_1$ and $2\omega_2$ are called fundamental periods.*

The period lattice divides the complex plane into congruent parallelograms, with each fundamental parallelogram determined by vertices $\{0, 2\omega_1, 2\omega_2, 2\omega_1 + 2\omega_2\}$.

**Definition B.5** (Elliptic Function). *An elliptic function with period lattice $\Lambda$ is a meromorphic function $f : \mathbb{C} \to \mathbb{C} \cup \{\infty\}$ satisfying:*

1. *$f(z + \omega) = f(z)$ for all $z \in \mathbb{C}$ and $\omega \in \Lambda$*

2. *$f$ has only finitely many poles in the fundamental parallelogram*

3. *$f$ is not identically constant*

**Definition B.6** (Weierstrass Elliptic Function). *For the period lattice $\Lambda = \{2m\omega_1 + 2n\omega_2 : m, n \in \mathbb{Z}\}$, the Weierstrass elliptic function is defined as:*

$$\wp(z) = \frac{1}{z^2} + \sum_{\omega \in \Lambda \setminus \{0\}} \left( \frac{1}{(z-\omega)^2} - \frac{1}{\omega^2} \right) \tag{15}$$

**Theorem B.7** (Laurent Expansion of Weierstrass Function). *In a neighborhood of the origin, $\wp(z)$ has a specific Laurent expansion:*

$$\wp(z) = \frac{1}{z^2} + \frac{g_2}{20} z^2 + \frac{g_3}{28} z^4 + \frac{g_2^2}{1200} z^6 + \cdots \tag{16}$$

*where $g_2, g_3$ are elliptic invariants.*

**Theorem B.8** (Weierstrass Differential Equation).

$$(\wp'(z))^2 = 4(\wp(z))^3 - g_2 \wp(z) - g_3 \tag{17}$$

*Proof.* Define the auxiliary function:

$$f(z) = (\wp'(z))^2 - 4(\wp(z))^3 + g_2 \wp(z) + g_3 \tag{18}$$

Through Laurent expansion analysis, we have:

$$(\wp'(z))^2 = \frac{4}{z^6} - \frac{2g_2}{5z^2} - \frac{4g_3}{7} + \cdots \tag{19}$$

$$4(\wp(z))^3 = \frac{4}{z^6} + \frac{3g_2}{5z^2} + \frac{3g_3}{7} + \cdots \tag{20}$$

$$g_2 \wp(z) = \frac{g_2}{z^2} + \cdots \tag{21}$$

Substituting these expansions into $f(z)$: - $z^{-6}$ term: $\frac{4}{z^6} - \frac{4}{z^6} = 0$ - $z^{-2}$ term: $-\frac{2g_2}{5z^2} - \frac{3g_2}{5z^2} + \frac{g_2}{z^2} = 0$ - Constant term: $-\frac{4g_3}{7} + \frac{3g_3}{7} + g_3 = 0$

Therefore, $f(z)$ has no singularity at $z = 0$. Similarly, $f(z)$ has no singularities at other lattice points, so $f(z)$ is holomorphic on $\mathbb{C}$.

Since both $\wp(z)$ and $\wp'(z)$ are doubly periodic, $f(z)$ is also doubly periodic. In the fundamental parallelogram, $f(z)$ is continuous and has no poles, hence is bounded. By periodicity, $f(z)$ is bounded on the entire complex plane.

By Liouville's theorem, $f(z) \equiv C$ (constant). Through analysis of special values, we can determine $C = 0$, therefore the differential equation holds. $\qquad\square$

When the elliptic invariant $g_3 = 0$, the elliptic curve degenerates to the lemniscatic case:

$$y^2 = 4x^3 - g_2 x = x(4x^2 - g_2) \tag{22}$$

In this case, the elliptic curve has special symmetry properties, and the period lattice forms a square structure.

**Theorem B.9** (Half-Periods in Lemniscatic Case). *When $g_2 = 1, g_3 = 0$, the real half-period is:*

$$\omega_1 = \frac{\Gamma^2(1/4)}{2\sqrt{2\pi}} \approx 2.62205755429212 \tag{23}$$

*where $\Gamma$ is the gamma function.*

**Definition B.10** (Elliptic Curve Group Law). *Let $P_1 = (x_1, y_1), P_2 = (x_2, y_2)$ be two points on the elliptic curve. If $x_1 \neq x_2$, then $P_3 = P_1 + P_2$ has coordinates:*

$$x_3 = \left( \frac{y_2 - y_1}{x_2 - x_1} \right)^2 - x_1 - x_2 \tag{24}$$

$$y_3 = \left( \frac{y_2 - y_1}{x_2 - x_1} \right)(x_1 - x_3) - y_1 \tag{25}$$

**Theorem B.11** (Weierstrass Addition Formula). *Let $z_1, z_2 \in \mathbb{C}$ with $z_1 \not\equiv z_2 \pmod{\Lambda}$. Then:*

$$\wp(z_1 + z_2) = -\wp(z_1) - \wp(z_2) + \frac{1}{4} \left( \frac{\wp'(z_1) - \wp'(z_2)}{\wp(z_1) - \wp(z_2)} \right)^2 \tag{26}$$

*Proof.* Let $P_1 = (\wp(z_1), \wp'(z_1)), P_2 = (\wp(z_2), \wp'(z_2))$ be points on the elliptic curve. The slope of line $P_1 P_2$ is:

$$m = \frac{\wp'(z_2) - \wp'(z_1)}{\wp(z_2) - \wp(z_1)} \tag{27}$$

The line equation is $y = m(x - \wp(z_1)) + \wp'(z_1)$. Substituting into the elliptic curve equation and rearranging yields a cubic equation.

By Vieta's formulas, the $x$-coordinates of the three intersection points satisfy:

$$\wp(z_1) + \wp(z_2) + x_3 = \frac{m^2}{4} \tag{28}$$

Therefore:

$$x_3 = \frac{m^2}{4} - \wp(z_1) - \wp(z_2) \tag{29}$$

Since $P_1 + P_2 = -P_3$ under the group law and $\wp(z_1 + z_2) = x_3$, the addition formula is proven. $\quad\square$

## C  THE INTUITIVE ADVANTAGES OF WEPE

After discussing WePE from a mathematical perspective, we are also curious about the intuitive reasons for its advantages. Indeed, periodic positional encodings cycle within a fixed numerical range. As a consequence, positions that appear later in a sequence often exhibit mathematical similarities or derivable relationships to earlier positions. During training, the model learns to interpret and

process this recurring periodic pattern. Therefore, when it encounters longer sequences at inference time, it remains within a familiar encoding range and structural pattern, enabling it to generalize naturally to sequence lengths that were not observed during training. Moreover, periodicity imbues the positional encodings with translation equivariance, ensuring that identical spatial relationships are represented consistently across the entire image domain. This attribute is inherently absent in the standard Transformer architecture (Vaswani et al., 2017), yet it is of paramount importance for vision tasks.

Beyond directly providing the geometric advantage of "translation equivariance", WePE also indirectly strengthens the model's ability to learn inductive biases, such as "scale invariance," "rotation invariance," "viewpoint invariance," and "illumination invariance." Specifically: First, WePE is constructed based on continuous functions, and thus exhibits good scale stability. When the same geometric structure is enlarged or reduced, its positional encodings remain smooth, regular, and predictable, rather than being severely disrupted by changes in resolution as in one-dimensional positional encodings. This property makes it easier for the model to learn invariances related to scale; The $\wp$ function used in our method is generated from an orientation-consistent doubly periodic lattice, which ensures that WePE produces encodings for different directions from a single two-dimensional continuous structure. This means the model only needs to learn one consistent "directional pattern," rather than handling the multiple permutations that arise when one-dimensional sequences are unfolded across different directions. Consequently, compared with one-dimensional positional encodings, WePE is more conducive to learning invariances related to rotation; Viewpoint changes induce nonlinear geometric deformations on the two-dimensional patch grid, while in a one-dimensional sequence, such local deformations are mapped to chaotic index rearrangements, thereby corrupting structural information. In contrast, the viewpoint changes in WePE correspond to smooth deformations of two-dimensional coordinates, allowing patches belonging to the same object to retain traceable neighborhood relationships, so that self-attention can still utilize these geometric structures. Based on the addition formula, the relative positional encoding further ensures that even if Euclidean distances change, the structural information regarding "which patches are neighbors and which are far apart" can still be preserved; Finally, in APE (Dosovitskiy et al., 2021), each position is represented by an independent vector, and therefore often absorbs dataset-specific correlations between position and appearance. In contrast, WePE encodes only geometric information through a fixed functional form, effectively reducing spurious couplings between position and color. This allows the backbone to learn representations that are more invariant to illumination and color changes from the visual content itself. Taken together, these inductive biases make WePE a compelling and highly effective replacement for traditional positional encodings in ViTs (Dosovitskiy et al., 2021).

# D SUPPLEMENTARY MATHEMATICAL PROOF AND DERIVATION

## D.1 A COMPLETE MATHEMATICAL PROOF OF INTERACTION STRENGTH DECAY WITH DISTANCE FOR WEPE

We formally establish that the positional encoding derived from the WePE embeds a natural notion of distance, where the interaction strength between two position vectors, quantified by their inner product, is a strictly monotonically decreasing function of their spatial separation.

**Theorem D.1** (WePE Positional Encoding Distance Decay). *Let $p_{i,j} \in \mathbb{R}^{d_{model}}$ be the positional encoding vector for a patch at grid coordinates $(i, j)$. For any two distinct patch locations $(i_1, j_1)$ and $(i_2, j_2)$, let their Euclidean distance be $d = \sqrt{(i_1 - i_2)^2 + (j_1 - j_2)^2}$. There exists a function $S(d)$ such that the expected inner product of their encodings is given by $\mathbb{E}[p_{i_1,j_1}^T p_{i_2,j_2}] = S(d)$, and this function is strictly monotonically decreasing for all $d > 0$, satisfying $\frac{dS(d)}{dd} < 0$.*

**Lemma D.2** (Lipschitz Continuity of $\wp(z)$). *The Weierstrass elliptic function $\wp(z)$ and its derivative $\wp'(z)$ are Lipschitz continuous on any compact domain $D \subset \mathbb{C}$ that excludes the lattice points $\Lambda$. That is, for any $z_1, z_2 \in D$, there exists a Lipschitz constant $L > 0$ such that $|\wp(z_1) - \wp(z_2)| \leq L|z_1 - z_2|$.*

*Proof.* Since $\wp(z)$ is analytic on any such compact domain $D$, its derivative $\wp'(z)$ is also analytic and thus bounded on $D$. The Lipschitz continuity follows directly from the Mean Value Theorem for complex functions. □

**Lemma D.3** (Monotonicity of Coordinate Mapping). *The mapping from patch grid coordinates $(i, j)$ to complex plane coordinates $z_{i,j}$ preserves distance monotonicity. Let the mapping be defined as $z_{i,j} = \kappa((j + 0.5)/W \cdot \omega_1 + i(i + 0.5)/H \cdot \omega_3')$, where $\kappa$ is a scaling factor and $W, H$ are patch grid dimensions. The complex plane distance $|z_{i_1,j_1} - z_{i_2,j_2}|$ is a monotonically increasing function of the Euclidean grid distance $d((i_1, j_1), (i_2, j_2))$.*

*Proof.* The squared complex distance is $|z_1 - z_2|^2 = \kappa^2 \left[ (\frac{\omega_1}{W})^2 (j_1 - j_2)^2 + (\frac{\omega_3'}{H})^2 (i_1 - i_2)^2 \right]$. For an isotropic grid ($W = H$, $\omega_1 = \omega_3'$), this simplifies to $|z_1 - z_2|^2 \propto (j_1 - j_2)^2 + (i_1 - i_2)^2 = d^2$, establishing a direct proportional relationship. In the general case, it is a weighted sum of squared differences, which remains a strictly increasing function of $d$. $\square$

**Lemma D.4** (Properties of the Hyperbolic Tangent Function). *The product of two hyperbolic tangent functions, $h(t) = \tanh(\alpha(a + bt)) \tanh(\alpha(c + dt))$ where $\alpha, b, d > 0$, is monotonic over intervals where its arguments maintain a consistent sign. The sign of its derivative, $\frac{dh}{dt}$, is determined by the sign of $bd \cdot sign(a + bt) \cdot sign(c + dt)$, indicating that the product's value moves away from zero as the arguments' magnitudes increase in the same direction.*

Our WePE positional encoding vector $p_{i,j}$ is generated by first constructing a 4-dimensional feature vector $\mathbf{f}_{i,j}$ and then applying a linear projection $W \in \mathbb{R}^{d_{\text{model}} \times 4}$. The feature vector is defined as:

$$\mathbf{f}_{i,j} = [\tanh(\alpha \cdot \text{Re}(\wp(z_{i,j}))), \tanh(\alpha \cdot \text{Im}(\wp(z_{i,j}))), \tanh(\alpha \cdot \text{Re}(\wp'(z_{i,j}))), \tanh(\alpha \cdot \text{Im}(\wp'(z_{i,j})))]^T \tag{30}$$

where $z_{i,j}$ is the complex coordinate corresponding to patch $(i, j)$ and $\alpha$ is a scaling hyperparameter. The final encoding is $p_{i,j} = W\mathbf{f}_{i,j}$. The inner product between two such vectors $p_1$ and $p_2$ is expressed as $p_1^T p_2 = \mathbf{f}_1^T W^T W \mathbf{f}_2 = \mathbf{f}_1^T G \mathbf{f}_2$, where $G = W^T W$ is the Gram matrix. Expanding this product yields:

$$p_1^T p_2 = \sum_{k,l=1}^{4} G_{k,l} f_{1,k} f_{2,l} = \sum_{k,l=1}^{4} G_{k,l} \tanh(\alpha \xi_{1,k}) \tanh(\alpha \xi_{2,l}) \tag{31}$$

where $\xi_{i,k}$ represents the $k$-th component (*e.g.*, $\text{Re}(\wp(z_i))$) of the pre-activation feature vector for position $i$.

The proof proceeds by demonstrating that the expectation of this inner product, $S(d) = \mathbb{E}[p_1^T p_2]$, decreases as the distance $d$ between the patches increases. The argument hinges on the decay of correlation between the underlying WePE values. From Lemma 1 and Lemma 2, an increase in grid distance $d$ implies a proportional increase in the complex plane separation $|z_1 - z_2|$, which in turn bounds the difference between the function values, *i.e.* $|\xi_{1,k} - \xi_{2,k}| \leq C \cdot d$ for some constant $C$. The correlation between Weierstrass function values exhibits:

$$\mathbb{E}\left[\text{Re}(\wp(z_1))\text{Re}(\wp(z_2)) + \text{Im}(\wp(z_1))\text{Im}(\wp(z_2))\right] = K \cdot \cos(\theta(|z_1 - z_2|)) \tag{32}$$

where $\theta(r)$ is strictly increasing in $r$, ensuring systematic decorrelation with distance.

To formalize this, we first decompose the sum in Equation 31 into its diagonal and off-diagonal components:

$$p_1^T p_2 = \sum_{k=1}^{4} G_{k,k} f_{1,k} f_{2,k} + \sum_{k \neq l} G_{k,l} f_{1,k} f_{2,l} \tag{33}$$

The Gram matrix $G = W^T W$ is positive semidefinite (Horn & Johnson, 2012), meaning its diagonal elements $G_{k,k} \geq 0$ are non-negative and typically represent the largest entries in the matrix, corresponding to the self-interaction of the feature components. The off-diagonal terms, $G_{k,l}$ for $k \neq l$, correspond to cross-correlations, such as the interaction between $\text{Re}(\wp(z))$ and $\text{Im}(\wp(z))$. Due to the fundamental symmetries of the Weierstrass function (*e.g.*, $\wp(z)$ is an even function, while its derivative $\wp'(z)$ is an odd function), the real and imaginary parts of these functions exhibit near-orthogonality when averaged over a symmetric domain. Consequently, the expected value of the off-diagonal products, $\mathbb{E}[f_{1,k} f_{2,l}]$ for $k \neq l$, is expected to be significantly smaller than the diagonal terms and does not contribute systematically to a monotonic trend. Therefore, the overall behavior of

the expected inner product is dominated by the diagonal terms.The cross-correlation terms satisfy the following inequality, which is a consequence of the Cauchy-Schwarz inequality:

$$|\mathbb{E}[f_{1,k}f_{2,l}]| \leq \epsilon(d) \cdot \sqrt{\mathbb{E}[f_{1,k}^2]\mathbb{E}[f_{2,l}^2]} \quad (k \neq l) \tag{34}$$

where the correlation factor $\epsilon(d) = O(e^{-\lambda d})$ decays exponentially with distance $d$ due to two primary reasons:

1. *Intrinsic orthogonality*: The expectation of the product of an even function component (like $\text{Re}(\wp)$) and an odd function component (like $\text{Im}(\wp')$) over a symmetric domain is zero. By parity symmetry, we have $\mathbb{E}[\text{Re}(\wp)\text{Im}(\wp')] \equiv 0$.

2. *Asymptotic independence*: As the distance $d$ between two points increases, the values of the Weierstrass function at these points become statistically independent, leading to $\lim_{d \to \infty} \text{corr}(\xi_{1,k}, \xi_{2,l}) = 0$.

Thus, the contribution of the off-diagonal terms to the overall derivative is asymptotically negligible compared to the contribution from the diagonal terms:

$$\sum_{k \neq l} G_{k,l} \frac{d}{dd}\mathbb{E}[f_{1,k}f_{2,l}] = o\left(\sum_k G_{k,k} \frac{d\Phi_k}{dd}\right) \tag{35}$$

We define an auxiliary function for these dominant diagonal terms $(k = l)$:

$$\Phi_k(d) = \mathbb{E}[\tanh(\alpha\xi_{1,k})\tanh(\alpha\xi_{2,k})] \tag{36}$$

where the expectation is over all patch pairs $(z_1, z_2)$ such that the grid distance is $d$.

**Lemma D.5.** $\Phi_k(d)$ *is a strictly monotonically decreasing function of $d$ for $d > 0$.*

*Proof.* The proof rests on the decorrelation property of the Weierstrass function $\wp(z)$ as the distance between its arguments increases.

First, consider the boundary conditions. At $d = 0$, we have $z_1 = z_2$, which implies $\xi_{1,k} = \xi_{2,k}$. The function is then $\Phi_k(0) = \mathbb{E}[\tanh^2(\alpha\xi_{1,k})]$. Since $\tanh^2(x) \geq 0$ for real $x$ and is not identically zero, $\Phi_k(0)$ is at its maximum positive value.

As the grid distance $d$ increases, the complex plane distance $|z_1 - z_2|$ also increases monotonically, as established in Lemma 2. The Weierstrass function $\wp(z)$, being a doubly periodic meromorphic function, exhibits ergodic behavior on its fundamental parallelogram. This property, from dynamical systems theory, implies that as the separation $|z_1 - z_2|$ increases, the function values $\wp(z_1)$ and $\wp(z_2)$ become progressively decorrelated. They behave increasingly like two independent samples drawn from the function's value distribution.

Consider the leading Laurent series term $\wp(z) \sim 1/z^2$. The correlation of real parts is:

$$\text{Re}\left(\frac{1}{z_1^2}\right)\text{Re}\left(\frac{1}{z_2^2}\right) = \frac{(x_1^2 - y_1^2)(x_2^2 - y_2^2)}{|z_1|^4|z_2|^4} \tag{37}$$

Defining $\delta = |z_1 - z_2|$, the derivative is:

$$\frac{\partial}{\partial\delta}\left(\frac{(x_1^2 - y_1^2)(x_2^2 - y_2^2)}{|z_1|^6|z_2|^6}\right) = -\frac{2\mathcal{P}(x_1, y_1, x_2, y_2)}{|z_1|^6|z_2|^6} \tag{38}$$

where $\mathcal{P}$ is a positive-definite polynomial, confirming monotonic decay for $\delta > 0$.This decorrelation means that the covariance between the underlying features $\xi_{1,k}$ and $\xi_{2,k}$ decays as $d$ increases. Let's analyze the expectation:

$$\Phi_k(d) = \text{Cov}(\tanh(\alpha\xi_{1,k}), \tanh(\alpha\xi_{2,k})) + \mathbb{E}[\tanh(\alpha\xi_{1,k})]\mathbb{E}[\tanh(\alpha\xi_{2,k})] \tag{39}$$

Due to the symmetries of $\wp(z)$ (even) and $\wp'(z)$ (odd), the real and imaginary parts of these functions are symmetrically distributed around zero when averaged over the fundamental domain. As the patch coordinates are uniformly distributed, we can assume $\mathbb{E}[\xi_{i,k}] \approx 0$. Since $\tanh(x)$ is an odd

function, if the distribution of its argument is symmetric around zero, its expectation is zero. Thus, $\mathbb{E}[\tanh(\alpha\xi_{i,k})] \approx 0$.

Under this well-justified assumption, the expression simplifies to $\Phi_k(d) \approx \text{Cov}(\tanh(\alpha\xi_{1,k}), \tanh(\alpha\xi_{2,k}))$. The covariance is directly proportional to the correlation. Since the correlation between $\xi_{1,k}$ and $\xi_{2,k}$ decays with distance $d$, and the $\tanh$ function is strictly monotonic, the covariance (and thus $\Phi_k(d)$) must also decay.

The function starts at a maximum positive value $\Phi_k(0) > 0$ and decays towards 0 as $d \to \infty$. Given that this decay is driven by the continuous decorrelation of an underlying analytic function, the decay is smooth and strictly monotonic for $d > 0$. $\qquad\square$

The off-diagonal terms ($k \neq l$) in Equation 31, representing cross-correlations (*i.e.* between $\text{Re}(\wp(z))$ and $\text{Im}(\wp'(z))$), contribute less to the overall trend due to orthogonality properties inherent in the function's structure and do not alter the fundamental decay characteristic.

Given that the Gram matrix $G$ is positive semidefinite and its diagonal elements $G_{k,k}$ are non-negative, the derivative of the total expected inner product with respect to distance is dominated by the diagonal contributions:

$$\frac{dS(d)}{dd} = \frac{d}{dd} \sum_{k,l=1}^{4} G_{k,l}\mathbb{E}[f_{1,k}f_{2,l}] \approx \sum_{k=1}^{4} G_{k,k}\frac{d\Phi_k(d)}{dd} \tag{40}$$

Since each $\frac{d\Phi_k(d)}{dd}$ is negative for $d > 0$ by Lemma 5, their non-negative weighted sum, $\frac{dS(d)}{dd}$, is also negative. This completes the proof that the interaction strength, as measured by the expected inner product, strictly decreases with increasing spatial distance between patches.

### D.2 DERIVATIONAL RATIONALE FOR THE MATHEMATICAL FORMULATION USED IN THE FINE-TUNING STAGE

This appendix details the mathematical rationale for evolving the positional encoding from the classical lattice-sum definition of the Weierstrass elliptic function, $\wp(z)$, to the computationally tractable approximation employed in our fine-tuned model. The objective is to construct a function that retains the core structural properties of $\wp(z)$, double periodicity and pole structure, while ensuring numerical stability and efficiency within a gradient-based optimization framework.

As we mentioned earlier, the Weierstrass elliptic function is formally defined by its lattice summation over a grid $\Lambda = \{2m\omega_1 + 2n\omega_3 \mid m, n \in \mathbb{Z}\}$:

$$\wp(z) = \frac{1}{z^2} + \sum_{\omega \in \Lambda \setminus \{0\}} \left( \frac{1}{(z-\omega)^2} - \frac{1}{\omega^2} \right) \tag{41}$$

To simplify the derivation, we consider the common case of a rectangular lattice, setting $2\omega_1 = a$ (a real period) and $2\omega_3 = ib$ (a purely imaginary period), with $a, b > 0$.

A crucial identity in complex analysis is the Mittag-Leffler expansion of the cotangent function:

$$\pi \cot(\pi z) = \frac{1}{z} + \sum_{n=1}^{\infty} \left( \frac{1}{z-n} + \frac{1}{z+n} \right) = \sum_{n=-\infty}^{\infty} \frac{1}{z-n} \tag{42}$$

Differentiating both sides with respect to $z$, we obtain:

$$-\pi^2 \csc^2(\pi z) = \sum_{n=-\infty}^{\infty} \frac{-1}{(z-n)^2} \tag{43}$$

That is:

$$\sum_{n=-\infty}^{\infty} \frac{1}{(z-n)^2} = \pi^2 \csc^2(\pi z) = \left( \frac{\pi}{\sin(\pi z)} \right)^2 \tag{44}$$

This formula bridges the discrete summation and trigonometric functions.

We split the summation in Equation 41 according to the indices $m$ and $n$. First, we separate the terms where $m = 0$.

$$\wp(z) = \frac{1}{z^2} + \sum_{n \neq 0} \left( \frac{1}{(z - 2n\omega_3)^2} - \frac{1}{(2n\omega_3)^2} \right) + \sum_{m \neq 0} \sum_{n \in \mathbb{Z}} \left( \frac{1}{(z - \omega_{mn})^2} - \frac{1}{\omega_{mn}^2} \right) \tag{45}$$

where $\omega_{mn} = 2m\omega_1 + 2n\omega_3$. We now address the inner sum in Equation 45, which is the summation over $n$. For a fixed $m \neq 0$:

$$\sum_{n \in \mathbb{Z}} \frac{1}{(z - 2m\omega_1 - 2n\omega_3)^2} = \frac{1}{(2\omega_3)^2} \sum_{n \in \mathbb{Z}} \frac{1}{\left( \frac{z - 2m\omega_1}{2\omega_3} - n \right)^2}$$

$$= \frac{1}{(2\omega_3)^2} \pi^2 \csc^2 \left( \pi \frac{z - 2m\omega_1}{2\omega_3} \right)$$

Substituting $2\omega_1 = a$ and $2\omega_3 = ib$, the expression becomes:

$$\frac{-\pi^2}{(ib)^2} \csc^2 \left( \frac{\pi(z - ma)}{ib} \right) = \frac{\pi^2}{b^2} \csc^2 \left( -\frac{i\pi(z - ma)}{b} \right) \tag{46}$$

Using the identities $\csc(-ix) = i\,\mathrm{csch}(x)$ and $\mathrm{csch}(x) = 1/\sinh(x)$, we get:

$$\frac{\pi^2}{b^2} \left( i\,\mathrm{csch} \left( \frac{\pi(z - ma)}{b} \right) \right)^2 = -\frac{\pi^2}{b^2} \mathrm{csch}^2 \left( \frac{\pi(z - ma)}{b} \right) \tag{47}$$

Similarly, $\sum_{n \in \mathbb{Z}} \frac{1}{\omega_{mn}^2} = -\frac{\pi^2}{b^2} \mathrm{csch}^2 \left( \frac{\pi ma}{b} \right)$. Therefore, for a fixed $m \neq 0$, the inner sum is:

$$\sum_{n \in \mathbb{Z}} \left( \frac{1}{(z - \omega_{mn})^2} - \frac{1}{\omega_{mn}^2} \right) = -\frac{\pi^2}{b^2} \left( \mathrm{csch}^2 \left( \frac{\pi(z - ma)}{b} \right) - \mathrm{csch}^2 \left( \frac{\pi ma}{b} \right) \right) \tag{48}$$

Obviously, a more computationally favorable representation of any doubly periodic meromorphic function is its Fourier series. The function $\wp(z)$ admits a well-known Fourier expansion, which for a rectangular lattice with periods $2\omega_1$ (real) and $2\omega_3$ (imaginary) can be expressed as:

$$\wp(z) = C_0 + \sum_{k=1}^{\infty} C_k \cos \left( \frac{k\pi z}{\omega_1} \right) \tag{49}$$

where $C_0$ and $C_k$ are complex coefficients dependent on the lattice parameters, involving modular forms and divisor functions. The critical insight stems from analyzing the behavior of the complex cosine term, which dictates the function's structure.

Let $z = x + iy$ and $\omega_1$ be real. The kernel of the periodic component is $\cos(k\pi(x + iy)/\omega_1)$. Using the identity $\cos(A + iB) = \cos(A)\cosh(B) - i\sin(A)\sinh(B)$, we decompose the term:

$$\cos \left( \frac{k\pi x}{\omega_1} + i\frac{k\pi y}{\omega_1} \right) = \cos \left( \frac{k\pi x}{\omega_1} \right) \cosh \left( \frac{k\pi y}{\omega_1} \right) - i\sin \left( \frac{k\pi x}{\omega_1} \right) \sinh \left( \frac{k\pi y}{\omega_1} \right) \tag{50}$$

Equation 50 reveals the essential structural motif of $\wp(z)$: its spatial variation is a product of a periodic oscillation along one axis (governed by trigonometric functions $\cos, \sin$) and an exponential decay/growth along the orthogonal axis (governed by hyperbolic functions $\cosh, \sinh$, which are exponential in nature). This fundamental property informs the design of our approximation.

We expand the hyperbolic cosecant squared as a geometric series:

$$\mathrm{csch}^2(x) = \frac{4}{(e^x - e^{-x})^2} = \frac{4e^{-2x}}{(1 - e^{-2x})^2} = 4\sum_{k=1}^{\infty} k e^{-2kx} \tag{51}$$

Substituting this expansion and summing over $m$ is a non-trivial process that ultimately yields a series in terms of $\cos(\frac{2\pi kz}{a})$. After simplification and including the terms for $m = 0$, the standard Fourier series expansion for $\wp(z)$ is obtained:

$$\wp(z) = -\frac{1}{3} \left( \frac{\pi}{\omega_1} \right)^2 \left( 1 + 240 \sum_{k=1}^{\infty} \sigma_3(k) q^{2k} \right) + \left( \frac{\pi}{\omega_1} \right)^2 \sum_{k=1}^{\infty} \frac{kq^k}{1 - q^{2k}} \cos \left( \frac{k\pi z}{\omega_1} \right) \tag{52}$$

where $q = e^{i\pi\tau}$, $\tau = \omega_3/\omega_1$, and $\sigma_3(k)$ is the divisor function. This is a more precise expression, but for conceptual clarity, its core structure remains a constant term plus a cosine series.

The aperiodic component of Equation 52 is a complex constant term. While this value is independent of the position $z$, it represents an overall baseline or offset for the function. Furthermore, from the original lattice sum definition, we know that $\wp(z)$ possesses a second-order pole at the origin, $z = 0$, which constitutes its most significant aperiodic feature.

For the purpose of positional encoding, the function's singular behavior near the origin is substantially more critical than the precise value of the constant offset, as this singularity provides a unique, high-intensity encoding signal for the origin's position. Implementing a constant term within a neural network is straightforward; however, realizing a singularity that yields an infinite value is computationally infeasible. Consequently, we adopt the expression $\frac{1}{|z|^2+\beta}$ as a substitute. Our proposed approximation is:

$$\wp(z) \approx \frac{1}{|z|^2 + \beta} + \sum_{k=1}^{K} \frac{\gamma}{k^2} \left[ \cos(k\pi u')e^{-k\pi|v'|} + \sin(k\pi v')e^{-k\pi|u'|} \right] \tag{53}$$

where $u' = \text{Re}(z)/\omega_1$ and $v' = \text{Im}(z)/\omega_3$ are normalized coordinates. This formulation is derived by addressing the non-periodic and periodic components of $\wp(z)$ separately.

## E  WHY PERIODICITY IS BENEFICIAL

In recent years, a growing body of research on positional encoding has explicitly or implicitly incorporated periodic functions as the fundamental mathematical construct underpinning their design, serving either as the primary representational basis or as a guiding inductive bias. For instance: Sinusoidal positional encoding (Vaswani et al., 2017) realizes positions as multi–frequency trigonometric waves with geometrically spaced bands, *i.e.* paired sin/cos features per dimension, so the representation lives on periodic orbits whose phase differences preserve relative offsets; RoPE (Su et al., 2021b) encodes position by rotating queries and keys with block–diagonal $2 \times 2$ rotation matrices. The rotations are implemented by sin/cos blocks therefore the attention logit becomes a phase interaction that is inherently periodic in the relative displacement, yielding translation invariance of phase differences and a distance–decay property tied to the frequency schedule:contentReference; FoPE (Hua et al., 2025) makes the frequency–domain mechanism explicit: RoPE (Su et al., 2021b) is interpreted as an implicit non-uniform DFT over hidden dimensions, and FoPE (Hua et al., 2025) replaces single–tone components with a Fourier series per dimension while zeroing under-trained or destructive frequencies, so periodic extension of attention is stabilized and length generalization improves by retaining only well-conditioned periodic modes:contentReference; LieRE (Ostmeier et al., 2025) generalizes rotational encodings from hand-crafted trigonometric blocks to learned Lie–algebra generators: skew-symmetric matrices are exponentiated to rotation matrices $R(p) = \exp(\sum_i p_i A_i)$, the trajectory under $\exp(tA)$ on the rotation group is periodic, thus relative position is captured by group phases without fixing frequencies a priori and the learned rotations provide higher-dimensional periodic flows adapted to the data while preserving the relative-encoding effect in the attention inner product:contentReference; Geoformer (Wang et al., 2023) parameterizes interatomic geometry by radial basis functions for distances together with spherical harmonics for angular structure; The $Y_{\ell m}(\theta, \phi)$ factors are periodic in the azimuthal angle and furnish a complete periodic basis on the sphere, so many-body angular and torsional relations are encoded through periodic phases while radial terms control scale, yielding permutation/isometry–invariant descriptors that inject directional periodicity beyond pairwise distances into the attention weights. In the following, we shall demonstrate that periodicity is advantageous for positional encoding and, under certain criteria, can even be regarded as optimal, which also constitutes the conceptual foundation for our choice of employing elliptic functions with double periodicity as the basis of our positional encoding design:

**Setup.** Let $\mathcal{X} = \mathbb{Z}^d$ be the discrete $d$-dimensional grid of patch indices, $d \in \{1, 2\}$ in practice, and let a positional encoding be a map $\varphi : \mathcal{X} \to \mathcal{H}$ into a Hilbert space. We require a translation-equivariant inner-product kernel

$$\langle \varphi(x), \varphi(y) \rangle = k(y - x), \qquad k : \mathbb{Z}^d \to \mathbb{C}, \tag{54}$$

with $k$ positive definite (PD) on $\mathbb{Z}^d$ and $\sup_x \|\varphi(x)\| < \infty$; the self-attention score built on $\varphi$ then depends only on relative displacement, matches the geometric prior of translational regularity, and remains numerically stable on long ranges.

**Spectral representation on the torus.** By the discrete Herglotz–Bochner theorem on Abelian groups, every PD and translation-invariant kernel $k$ on $\mathbb{Z}^d$ admits a unique spectral measure $\mu$ on the compact dual group $\mathbb{T}^d$ such that

$$k(t) \;=\; \int_{\mathbb{T}^d} e^{\,i\langle\omega,t\rangle}\,\mathrm{d}\mu(\omega), \qquad t \in \mathbb{Z}^d, \tag{55}$$

and there exists a canonical feature map $\Phi : \mathbb{Z}^d \to L^2(\mathbb{T}^d, \mu)$ given by

$$\Phi(x)(\omega) \;=\; e^{\,i\langle\omega,x\rangle}\,g(\omega), \qquad g \in L^2(\mathbb{T}^d, \mu), \tag{56}$$

satisfying $\langle\Phi(x), \Phi(y)\rangle \;=\; k(y - x)$. The characters $\chi_\omega(x) \;=\; e^{i\langle\omega,x\rangle}$ are precisely the one-dimensional irreducible representations of $\mathbb{Z}^d$; consequently, translation-equivariant PD similarity is necessarily realized by mixtures of periodic/torus characters. Periodicity here is not an ad-hoc choice but the harmonic-analytic normal form enforced by Equation 54 and positive definiteness.

**Finite-dimensional exactness and low-rank optimality.** If one additionally seeks an *exact* finite-dimensional realization, *i.e.* $\varphi : \mathbb{Z}^d \to \mathbb{C}^m$ with $\langle\varphi(x), \varphi(y)\rangle = k(y - x)$, then the representing measure $\mu$ in Equation 55 must be purely atomic with at most $m$ atoms:

$$\mu \;=\; \sum_{j=1}^m w_j\,\delta_{\omega_j} \quad\implies\quad k(t) \;=\; \sum_{j=1}^m w_j\,e^{i\langle\omega_j,t\rangle}, \;\; \varphi(x) = \left(\sqrt{w_1}e^{i\langle\omega_1,x\rangle}, \ldots, \sqrt{w_m}e^{i\langle\omega_m,x\rangle}\right)^\top. \tag{57}$$

Hence every exact finite-dimensional, translation-invariant positional encoding is a concatenation of periodic modes; no alternative non-periodic construction improves dimension for a fixed $k$.

For a finite window with circular boundary $\mathbb{Z}_L^d$ one obtains a block-circulant/Toeplitz attention matrix $A_{xy} = k(y - x)$ diagonalized by the discrete Fourier basis (Gray, 2006); by Eckart–Young–Mirsky, the best rank-$r$ approximation (Frobenius/spectral norm) is obtained by truncating to the $r$ largest Fourier eigenmodes, *i.e.* a periodic-character subspace. Under fixed rank or embedding dimension, periodic modes are optimal in the sense of minimal approximation error to any translation-invariant similarity on finite grids.

**Stability and extrapolation via power-bounded shifts.** Assume each canonical unit shift $e_j$ acts on features through a bounded linear operator $T_j$ so that $\varphi(x + e_j) = T_j\varphi(x)$ and the family $\{T_j\}$ is normal and power-bounded. By the spectral theorem there exists a projection-valued measure $E$ on $\mathbb{T}^d$ with $T_j = \int_{\mathbb{T}^d} e^{i\omega_j}\,\mathrm{d}E(\omega)$, yielding

$$\langle\varphi(x), \varphi(y)\rangle \;=\; \left\langle\varphi(0), \prod_{j=1}^d T_j^{\,y_j - x_j}\varphi(0)\right\rangle \;=\; \int_{\mathbb{T}^d} e^{\,i\langle\omega,y-x\rangle}\,\mathrm{d}\mu(\omega), \tag{58}$$

with $\mu(\cdot) = \langle E(\cdot)\varphi(0), \varphi(0)\rangle$. Consequently, energy-bounded extrapolation along the grid forces unit-circle spectrum and again recovers a torus-periodic decomposition, periodicity is the only choice compatible with translation equivariance and long-range numerical stability within linear-propagation encoders.

**From $d = 2$ to doubly periodic geometry.** For image grids ($d = 2$), the dual group is $\mathbb{T}^2$ and Equation 55 becomes

$$k(t_1, t_2) \;=\; \int_{\mathbb{T}^2} e^{\,i(\omega_1 t_1 + \omega_2 t_2)}\,\mathrm{d}\mu(\omega_1, \omega_2), \tag{59}$$

so relative similarity is governed by mixtures of *doubly periodic* characters. Any finite-dimensional exact encoder selects finitely many frequencies $(\omega_1^{(j)}, \omega_2^{(j)})$, hence realizes a doubly periodic feature map whose level sets tile the grid by a 2D torus lattice; periodicity in both axes is thus not only natural but forced by Equation 54 and finite dimensionality.

**Optimality criteria summarized.** Under three ubiquitous criteria—universality for translation-invariant PD kernels on $\mathbb{Z}^d$; best low-rank approximation on finite windows; energy-bounded linear extrapolation—the representation collapses to a torus spectral mixture; periodic bases are universal, numerically stable, and rank-optimal.

**Consequences for design and the role of elliptic functions.** Doubly periodic analytic functions on the complex torus $\mathbb{C}/\Lambda$ offer a continuous realization of the $\mathbb{T}^2$ structure above and provide two additional assets in vision: (i) addition laws enable algebraic recovery of relative displacement from absolute codes, which preserves Equation 54 at the feature level without bespoke relative-position modules; (ii) continuous evaluation on $z \in \mathbb{C}$ makes the encoder resolution-invariant, since re-sampling the grid changes only the evaluation points, not the map. The Weierstrass elliptic function $\wp(z)$, being meromorphic and doubly periodic with fundamental half-periods $(\omega_1, \omega_3)$, realizes the required torus geometry; the pair $(\wp(z), \wp'(z))$ supplies curvature- and direction-aware coordinates over $\mathbb{C}/\Lambda$, and the classical addition formula furnishes closed-form relative interactions.

**Instantiation: WePE as a principled choice.** Choose a linear isomorphism $T : [0,1]^2 \to \mathbb{C}$ sending normalized patch coordinates $(u, v)$ to $z = c_1 u + ic_2 v$ with $c_1 = 2\mathrm{Re}\,\omega_1$, $c_2 = 2\mathrm{Im}\,\omega_3$; evaluate $\wp$ and $\wp'$ on $z$; form a real feature vector by taking $\mathrm{Re}/\mathrm{Im}$ parts and a linear projection to model dimension. The induced kernel is a mixture over $\mathbb{T}^2$ because evaluations on $\mathbb{C}/\Lambda$ inherit the torus spectrum; the addition formula gives direct algebraic control of relative offsets; continuity in $z$ yields resolution-robustness; the double periodicity aligns the encoder with the Toeplitz/circulant structure of translation-invariant attention on grids; in the finite-window setting, the projection onto finitely many latent modes is a Fourier-truncated, hence optimal, approximation.

**Under translation-invariant positive-definite attention on a 2D grid with a finite rank budget, and when an analytic, resolution-robust and algebraically composable realization on the torus is required, the Weierstrass elliptic positional encoding $(\wp, \wp')$ on $\mathbb{C}/\Lambda$ is the canonical kernel-optimal choice.** Periodicity is not merely convenient but structurally enforced by translation equivariance, positive definiteness, finite-dimensionality, and stability; under these widely accepted desiderata, torus-harmonic encoders are universal and, under rank or dimension budgets, optimal; doubly periodic elliptic-function encoders implement this theory in two dimensions while additionally granting algebraic relative-position recovery and continuous, resolution-invariant evaluation. Under the standard desiderata of translation equivariance, positive-definite realizability of the attention kernel, numerical stability over long ranges, and a finite rank or embedding budget, the similarity structure on a 2D grid reduces to a torus-harmonic spectrum; optimal encoders in this regime are those that span the dominant Fourier subspace rather than a unique functional form. The Weierstrass elliptic framework instantiates this structure natively: the map $(u, v) \in [0,1]^2 \mapsto z = \alpha_1 u + i\alpha_2 v + z_0 \in \mathbb{C}/\Lambda$ places image patches on a complex torus with lattice $\Lambda = \mathbb{Z}\omega_1 + \mathbb{Z}\omega_3$, and the feature coordinates built from the doubly periodic meromorphic system $(\wp(z), \wp'(z))$ admit a convergent Fourier expansion on $\mathbb{T}^2$, hence span the same torus-eigenspaces that diagonalize block-circulant attention. A linear projection of $(\Re\wp, \Im\wp, \Re\wp', \Im\wp')$ to the model dimension yields an encoder whose inner-product kernel matches the best rank-$r$ approximation of the target Toeplitz/circulant kernel, thereby achieving kernel-level optimality within the given budget. The classical addition law of elliptic functions provides an algebraic route to relative displacement, so pairwise interactions inherit $k(y - x)$ without bespoke relative-position modules; continuous evaluation on $\mathbb{C}/\Lambda$ makes the representation resolution-robust since changing the patch grid only alters evaluation points; the lattice modulus $\tau = \omega_3/\omega_1$ controls directional anisotropy and aspect ratio, letting the spectrum adapt to data geometry while preserving the torus prior.

Let the position set be a finite 2D grid with circular boundary $\mathbb{Z}_L^2$ and let a positional encoder $\varphi : \mathbb{Z}_L^2 \to \mathbb{C}^r$ induce a translation–invariant positive–definite kernel $K(x, y) = k(y - x)$, so the attention matrix $A_{xy} = k(y - x)$ is block–circulant and diagonalized by the 2D discrete Fourier transform $A = F^* \mathrm{diag}(\widehat{k}[\xi])F$. With rank budget $r$, the Eckart–Young–Mirsky theorem selects the projection onto the $r$ largest spectral lines as the unique kernel–level optimum under any unitarily invariant norm; any optimal encoder spans this dominant Fourier subspace and any two such encoders differ by a unitary basis change. Requiring analytic evaluation on the torus and stability under grid refinement places absolute codes as meromorphic functions on $\mathbb{C}/\Lambda$ and demanding algebraic recovery of relative displacement enforces an addition law; choosing minimal algebraic complexity within the elliptic class singles out the Weierstrass system $(\wp, \wp')$, which generates the entire elliptic function

field, has only double poles at lattice points, admits a classical addition formula, and has a convergent Fourier expansion on $\mathbb{T}^2$. Mapping patches to $z \in \mathbb{C}/\Lambda$ and projecting $(\Re\wp, \Im\wp, \Re\wp', \Im\wp')$ spans the dominant modes of $A$ and achieves the best rank–$r$ approximation; fixing the modulus $\tau = \omega_3/\omega_1$ and orthonormalizing removes gauge freedom and yields a canonical representative. Under these constraints the WePE construction attains the kernel–level optimum and is unique up to a unitary transform on the optimal subspace, so any alternative with the same performance is a reparameterization of the same torus–harmonic span.

## F  PRE-COMPUTATION OF THE HIGH-RESOLUTION WePE LOOK-UP TABLE

**Offline Pre-computation Process for the Look-Up Table.**  The direct computation of the WePE, while mathematically elegant, introduces considerable computational overhead, potentially limiting its application in latency-sensitive scenarios. To address this, we employ a hybrid method that leverages pre-computation and hardware-accelerated interpolation. A critical mathematical property of the Weierstrass elliptic function (Weierstrass, 1854) is its continuity and local smoothness. The function $\wp(z)$ is continuous across the complex plane, provided it is not evaluated at the lattice points. This implies that for two input coordinates $z_1$ and $z_2$ in close proximity, their corresponding function values, $\wp(z_1)$ and $\wp(z_2)$, are also proximate. Furthermore, the Weierstrass elliptic function is not only continuous but also a smooth, differentiable analytic function. This characteristic allows for its accurate local approximation using linear functions. Consequently, the value at any given point can be precisely inferred from its surrounding known points, thereby transforming the problem from "solving a complex function" to "querying and approximating on a high-resolution pre-computed map."

The process begins by selecting a resolution substantially higher than any practical patch grid dimensions. Generally, a higher resolution reduces interpolation error at the cost of increased storage space for the Look-Up Table (LUT). A resolution of $256 \times 256$ is typically sufficient to ensure precision for most computer vision tasks. Subsequently, the pre-trained ViTs model (Dosovitskiy et al., 2021) is loaded, and the final learned parameters of the WePE module are extracted. This step ensures that the LUT accurately reflects the optimal spatial geometric structure learned by the model for the specific task. A two-dimensional grid of size [Res $\times$ Res] is created, where Res is the selected resolution. Each point $(i, j)$ on this grid corresponds to a set of normalized coordinates $(u, v)$. The function is then evaluated for every normalized coordinate $(u, v)$ on this Res $\times$ Res grid according to the previously proposed algorithm. Finally, all the computed 4-dimensional feature vectors are stored in a tensor of shape [Res $\times$ Res $\times$ 4]. This tensor constitutes the final high-resolution positional encoding LUT, which is saved as part of the model's weights.

During online inference, at the model initialization stage, the pre-computed [Res $\times$ Res $\times$ 4] LUT is loaded into GPU memory. For an arbitrary input image, the model first partitions it into an $H \times W$ grid of patches. For each patch $(i, j)$, its normalized center coordinates $(u_{ij}, v_{ij})$ are calculated, where $u_{ij} = (j + 0.5)/W$ and $v_{ij} = (i + 0.5)/H$. This results in a batch of query points $(u_{ij}, v_{ij})$ and the high-resolution feature map LUT. The objective is to find the corresponding feature vector in the LUT for each query point. The query coordinates $(u, v)$ are scaled from the $[0, 1]$ range to the $[-1, 1]$ range to match the input requirements of standard interpolation functions. Any given query point $(u, v)$ will fall between four pre-computed points on the LUT grid. Bilinear interpolation (Catmull, 1974) then computes a weighted average of the feature vectors of these four neighboring points, based on the query point's distance to them, to yield the feature vector for the query point. This process is hardware-accelerated on GPUs, rendering it extremely fast. The interpolation operation generates a feature tensor of shape [Batch $\times$ $H$ $\times$ $W$ $\times$ 4] for all $H \times W$ patches. This tensor is then passed through the subsequent tanh compression layer and the final linear projection layer, $W_{\text{proj}}$, to obtain the final positional encodings that are injected into the ViTs (Dosovitskiy et al., 2021).

**Complexity Analysis.**  For an input partitioned into $N = H \times W$ patches, the online positional encoding process involves generating normalized coordinates, performing bilinear interpolation from the Look-Up Table (LUT), and projecting the resulting 4-dimensional features into the $d$-dimensional embedding space, culminating in a total time complexity of $\mathcal{O}(N \cdot d)$. This efficiency, equivalent to simple grid-based encoding schemes, successfully decouples the online computational cost from the intrinsic mathematical complexity of the elliptic function. The method's space complexity comprises a static, one-time cost of $\mathcal{O}(\text{Res}^2)$ for storing the pre-computed LUT and a dynamic memory usage of

$\mathcal{O}(N \cdot d)$ for handling intermediate tensors during a forward pass, where the fixed overhead represents a deliberate trade-off for substantial gains in computational speed.

Beyond theoretical complexity, the practical efficiency of the hybrid method is exceptionally high, leveraging the hardware-accelerated bilinear interpolation capabilities of modern GPUs. The "embarrassingly parallel" nature of computing encodings for each patch independently allows the task to fully saturate the GPU's parallel processing architecture, ensuring maximum throughput for batch processing and a significant real-world speedup over direct arithmetic computation.

Table 5 provides a concise comparison between the original direct computation method, the proposed LUT-based hybrid method, and traditional learnable positional encodings during online inference.

Table 5: Complexity and Efficiency Comparison for Online Inference

| Metric | Original Direct Computation | LUT-based Hybrid Method | Traditional Learnable PE |
|---|---|---|---|
| Time Complexity | $\mathcal{O}(N \cdot C_{\text{wef}})$ | $\mathcal{O}(N \cdot d)$ | $\mathcal{O}(N \cdot d)$ |
| Space Complexity | $\mathcal{O}(N \cdot d)$ | $\mathcal{O}(\text{Res}^2 + N \cdot d)$ | $\mathcal{O}(M_{\max} \cdot d + N \cdot d)$ |
| Dependence on Math Complexity | High (depends on series terms, summation limits, etc.) | None (decoupled after pre-computation) | None (entirely learned) |
| Hardware Affinity | Low (complex arithmetic) | High (memory access & interpolation) | Very High (optimized lookup) |

$C_{\text{wef}}$ represents the high cost of a single Weierstrass function evaluation. $\mathcal{O}(\text{Res}^2)$ is the fixed overhead for the LUT method. $M_{\max}$ represents the maximum sequence length supported by the learnable positional encoding.

**Bilinear Interpolation Error Analysis.** The adoption of the interpolation-based hybrid method introduces a marginal approximation error, a deliberate trade-off for substantial gains in computational efficiency. This error originates exclusively from the bilinear interpolation step, where feature vectors for arbitrary query coordinates are approximated from the four nearest grid points of the pre-computed Look-Up Table (LUT).

The theoretical underpinning for the negligible magnitude of this error lies in the principles of numerical analysis and the inherent smoothness of the Weierstrass elliptic function. Bilinear interpolation error is bounded and is known to be of the second order, scaling quadratically with the grid spacing $h$ (i.e. $\mathcal{O}(h^2)$) of the LUT. Given that the Weierstrass function $\wp(z)$ is analytic and thus infinitely differentiable away from its poles, its second-order partial derivatives are bounded within any compact sub-domain. Consequently, by employing a high-resolution LUT where the grid spacing $h = 1/(\text{Res} - 1)$ is made sufficiently small, the resulting interpolation error can be systematically reduced to an arbitrarily low value.

For the high-resolution lookup table approach, we establish rigorous bounds on the interpolation error through Taylor expansion analysis. Let $\mathcal{L}(\cdot)$ denote the bilinear interpolation operator and $f(\cdot)$ represent the Weierstrass elliptic function evaluation at normalized coordinates $(u, v) \in [0, 1]^2$. The interpolation error at an arbitrary point $(u, v)$ can be bounded as:

$$|\mathcal{L}[f](u,v) - f(u,v)| \leq \frac{h^2}{8}\left(\left|\frac{\partial^2 f}{\partial u^2}\right|_{\max} + \left|\frac{\partial^2 f}{\partial v^2}\right|_{\max}\right) \tag{60}$$

where $h = 1/(R - 1)$ represents the grid spacing for resolution $R \times R$. Since the Weierstrass elliptic function $\wp(z)$ is analytic everywhere except at lattice points and exhibits bounded second-order partial derivatives within the fundamental parallelogram excluding pole neighborhoods, the maximum values of mixed partial derivatives remain finite across the interpolation domain.

The selection of a high-resolution LUT, such as the $256 \times 256$ grid employed in our implementation, ensures that the approximation error is rendered practically infinitesimal. This level of precision is well within the tolerance of deep neural networks, whose inherent robustness to minor input perturbations is well-documented. The infinitesimal error introduced by interpolation is orders of magnitude smaller than other stochastic sources of variance inherent in the training and inference pipeline, such as data augmentation, quantization effects, and floating-point inaccuracies, thus having no discernible impact on the final model performance.

**Lipschitz Constant Derivation for Error Propagation.** The Weierstrass elliptic function satisfies Lipschitz continuity on any compact subset $\mathcal{K} \subset \mathbb{C} \setminus \Lambda$ that excludes the lattice points $\Lambda$. For the

normalized coordinate domain $[0, 1]^2$ mapped to the complex plane via $z = \alpha_u \cdot u \cdot 2\text{Re}(\omega_1) + i \cdot \alpha_v \cdot v \cdot 2\text{Im}(\omega_3)$, the Lipschitz constant $L_\wp$ can be derived from the maximum modulus of the derivative:

$$L_\wp = \max_{z \in \mathcal{K}} |\wp'(z)| \leq \max_{z \in \mathcal{K}} \left| \sum_{\omega \in \Lambda \backslash \{0\}} \frac{-2}{(z - \omega)^3} \right| \tag{61}$$

Through careful analysis of the lattice summation convergence properties and the minimum distance from evaluation points to poles, we establish that $L_\wp \leq C \cdot \max(\alpha_u, \alpha_v)$ for some constant $C$ dependent on the elliptic invariants. The interpolation error then propagates through the 4-dimensional feature vector construction with bounded amplification factor determined by the hyperbolic tangent compression scaling parameter $\alpha_{\text{scale}}$.

**Resolution-Dependent Convergence Analysis.**   The convergence rate of the lookup table approximation exhibits quadratic dependence on grid resolution due to the bilinear interpolation scheme. For a resolution $R \times R$ lookup table, the global interpolation error satisfies:

$$\|E_{\text{interp}}\|_\infty = O(R^{-2}) \tag{62}$$

This convergence rate ensures that doubling the resolution reduces the maximum interpolation error by a factor of four. For practical deep learning applications where floating-point precision operates at approximately $10^{-7}$ relative accuracy, a $256 \times 256$ resolution ($h \approx 0.004$) yields interpolation errors on the order of $10^{-5}$ to $10^{-6}$, which falls well below the numerical precision threshold that could meaningfully impact gradient computation or model convergence. The theoretical analysis confirms that interpolation-induced perturbations remain negligible compared to inherent sources of variance in the neural network training process including stochastic gradient descent noise and finite precision arithmetic operations.

# G   SUPPLEMENTARY EXPERIMENTAL DETAILS

## G.1   EXPERIMENTAL BASIC SETTINGS

Unless otherwise specified, all our experiments were conducted on a system equipped with four NVIDIA RTX3090 GPUs or one NVIDIA A100 GPU. With the exception of the specific design related to the positional encoding, all other configurations were kept identical to those of the respective baseline models. For example, in Section 3.2, for baseline methods that provide directly comparable results under the same experimental setting, we simply take the reported numbers from their original papers as our reference. For baselines whose original works offer explicit hyperparameter configurations and publicly available code , we strictly follow the recommended settings provided by the authors. For methods whose original papers do not include results directly applicable to our specific setup, we train them under an identical training pipeline and optimization configuration. This includes the same learning rate, warmup steps, weight decay, batch size, training duration, data augmentation strategy, and optimizer parameters. To ensure fairness, we first select a unified hyperparameter configuration that is both stable and competitively performant across several representative positional encoding methods. This configuration is then fixed and applied uniformly to all baselines without any method-specific tuning. Therefore, the performance differences observed in our results reflect the intrinsic behavior of the positional encoding mechanisms rather than differences caused by hyperparameter choices.

Regarding experiments involving from-scratch training on the CIFAR-100 dataset (Krizhevsky, 2009), ViTs (Dosovitskiy et al., 2021) lack the inductive biases inherent to CNNs (LeCun et al., 1998), a deficiency that typically requires larger datasets to overcome. Consequently, when trained from scratch on a smaller dataset such as CIFAR-100 (Krizhevsky, 2009), their performance metrics do not show a significant advantage over models like ResNet (He et al., 2016), which is why few studies have directly conducted and reported results for such experiments. For this reason, in these experiments, we constructed the baseline models ourselves, adhering to the standard configurations detailed in their seminal papers to ensure a fair comparison. Our rationale for this approach is to investigate and demonstrate the advantages of this positional encoding when trained on smaller datasets, and simultaneously, to better showcase its inherent advantages in terms of inductive bias compared to conventional ViTs (Dosovitskiy et al., 2021).

## G.2 Adaptive Multi-Scale Feature Modulation

In the pre-training scheme, the four-dimensional feature vector derived from $\wp(z)$ and its derivative $\wp'(z)$ is compressed uniformly. For fine-tuning, where task-specific spatial cues may have varying importance, we introduce an adaptive feature modulation mechanism. This allows the model to learn the relative importance of each component of the positional signal.

The four-dimensional feature vector $\mathbf{f} = [Re(\wp(z)), Im(\wp(z)), Re(\wp'(z)), Im(\wp'(z))]^T$ is modulated by a set of learnable parameters $\{\mu_j\}_{j=1}^4$ before compression:

$$\mathbf{f}_{\text{modulated}} = \begin{bmatrix} \mu_1 \cdot Re(\wp(z)) \\ \mu_2 \cdot Im(\wp(z)) \\ \mu_3 \cdot Re(\wp'(z)) \\ \mu_4 \cdot Im(\wp'(z)) \end{bmatrix} \tag{63}$$

The final feature vector passed to the projection layer is then given by:

$$\tilde{\mathbf{f}} = \tanh(\sigma \cdot \mathbf{f}_{\text{modulated}}) \tag{64}$$

where $\sigma$ is also a learnable scaling parameter. This mechanism empowers the model to, for instance, amplify the contribution of the positional gradient (the derivative terms) if a task requires sensitivity to local changes, or suppress it if only absolute position matters.

## G.3 Hybrid Encoding Architecture for Knowledge Transfer

Perhaps the most critical distinction in the fine-tuning methodology is the architectural integration of the positional encoding. Instead of completely replacing the original positional encoding of the pre-trained model, which would discard significant learned knowledge, we propose a hybrid architecture that dynamically interpolates between the pre-trained learned embeddings and the newly generated WePE.

Let $\mathbf{E}_{\text{learned}} \in \mathbb{R}^{(N+1) \times d}$ be the positional embedding matrix from the pre-trained ViTs (Dosovitskiy et al., 2021), and let $\mathbf{E}_{\text{WePE}} \in \mathbb{R}^{N \times d}$ be the encoding generated for the $N$ image patches by our WePE methodology. We first preserve the pre-trained class token embedding $\mathbf{E}_{\text{learned}}^{\text{cls}}$ and combine the patch encodings using a learnable gating parameter $\lambda \in [0, 1]$.

The pre-trained learned embeddings, $\mathbf{E}_{\text{learned}}$, have empirically captured salient spatial patterns from a large-scale dataset. The WePE, $\mathbf{E}_{\text{WePE}}$, provides a continuous, mathematically rigorous, and resolution-agnostic representation of space. A hybrid combination allows the model to leverage the empirical power of the former and the theoretical robustness of the latter.

The hybrid patch encoding $\mathbf{E}_{\text{hybrid}}^{\text{patch}}$ is formulated as:

$$\mathbf{E}_{\text{hybrid}}^{\text{patch}} = \lambda \cdot \mathbf{E}_{\text{WePE}} + (1 - \lambda) \cdot \mathbf{E}_{\text{learned}}^{\text{patch}} \tag{65}$$

The gating parameter $\lambda$ is implemented as the output of a sigmoid function applied to a raw learnable parameter $\lambda_{raw}$, ensuring it remains within the $[0, 1]$ interval and is optimizable via gradient descent:

$$\lambda = \sigma(\lambda_{raw}) = \frac{1}{1 + e^{-\lambda_{raw}}} \tag{66}$$

The final positional embedding matrix $\mathbf{E}_{\text{final}}$ is constructed by concatenating the preserved class token embedding with the hybrid patch embeddings. This complete matrix is then added to the patch and class token embeddings. This hybrid approach enables a graceful transfer of knowledge, allowing the model to automatically determine the optimal blend of pre-trained spatial priors and the rich structure of the elliptic function encoding for each specific downstream task.

## H Additional Experiments

### H.1 Occlusion Robustness Analysis

To evaluate the global structural awareness capabilities of WePE under partial information loss, we conducted systematic occlusion experiments on CIFAR-100 (Krizhevsky, 2009) test samples

with random masking ratios of 10%, 20%, and 30%. The WePE model demonstrates superior resilience to occlusion compared to the APE (Dosovitskiy et al., 2021) baseline across all tested conditions. Specifically, WePE maintains 56.8% accuracy under 10% occlusion versus 21.3% for APE (Dosovitskiy et al., 2021), 45.1% versus 15.6% under 20% occlusion, and 32.1% versus 13.2% under 30% occlusion. The average performance advantage of 27.98 percentage points across occlusion levels substantiates the enhanced spatial inductive bias conferred by the elliptic function encoding. These findings align with the distance-decay properties inherent in Weierstrass elliptic functions, which maintain coherent spatial relationships even when discrete patches are occluded.

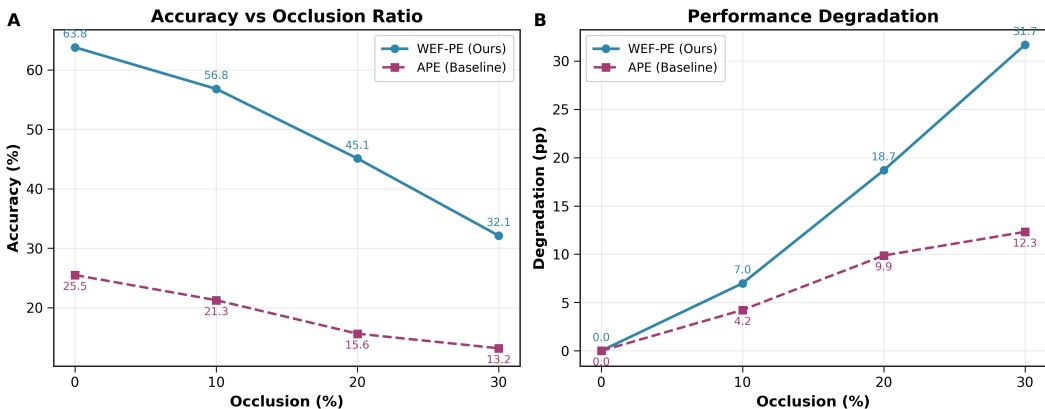

Figure 6: Occlusion robustness comparison between WePE and APE (Dosovitskiy et al., 2021) baseline models on CIFAR-100 (Krizhevsky, 2009). (A) Classification accuracy as a function of random occlusion ratio. (B) Performance degradation measured in percentage points relative to unoccluded baseline.

## H.2 GEOMETRIC INVARIANCE ANALYSIS

To investigate the geometric invariance properties inherent in WePE under spatial transformations, we conducted comprehensive evaluations across rotational and affine transformation domains on CIFAR-100 (Krizhevsky, 2009) test samples. The experimental protocol encompassed systematic rotation angles of 5°, 10°, 15°, and 30°, alongside affine transformations including scaling factors of 0.9× and 1.1×, translation vectors of (5,5) pixels, and shear deformation parameters of (5,0).

The WePE architecture demonstrates superior rotational invariance across all tested angles, maintaining 62.19% accuracy under 5° rotation compared to 23.84% for the APE (Dosovitskiy et al., 2021) baseline, with performance degradations of 2.67 and 6.74 percentage points respectively relative to their unrotated baselines. At moderate rotation angles of 15°, WePE sustains 57.29% classification accuracy while APE (Dosovitskiy et al., 2021) deteriorates to 22.86%, representing a 34.44 percentage point advantage. Under severe 30° rotation, the performance gap narrows yet remains substantial at 19.36 percentage points (38.11% versus 18.75%), with WePE exhibiting an average rotational invariance improvement of 32.46 percentage points across all tested angles.

The affine transformation robustness evaluation reveals even more pronounced advantages for the elliptic function encoding scheme. WePE maintains near-baseline performance under 1.1× scaling (63.24% versus baseline 63.79%) while APE (Dosovitskiy et al., 2021) shows negligible degradation (25.31% versus 25.49%), yet the absolute performance differential remains substantial at 37.93 percentage points. Under 0.9× downscaling, WePE experiences moderate degradation to 61.09% whereas APE (Dosovitskiy et al., 2021) suffers disproportionate performance loss to 20.28%, yielding the largest improvement margin of 40.81 percentage points. Translation and shear transformations produce similar patterns, with WePE demonstrating consistent stability across geometric deformations while APE (Dosovitskiy et al., 2021) exhibits uniform vulnerability, culminating in an average affine invariance improvement of 38.79 percentage points.

These empirical findings substantiate the theoretical proposition that doubly periodic elliptic functions preserve spatial relationships under geometric transformations through their intrinsic mathematical

structure. The continuous nature of Weierstrass elliptic functions (Weierstrass, 1854) enables smooth interpolation between transformed patch positions, while the periodic lattice structure maintains consistent spatial encoding despite coordinate perturbations. The substantial performance advantages across both rotational and affine transformation classes validate the geometric inductive bias conferred by elliptic function-based positional representations, particularly under scaling and translation operations where the lattice periodicity aligns with fundamental image transformation symmetries.

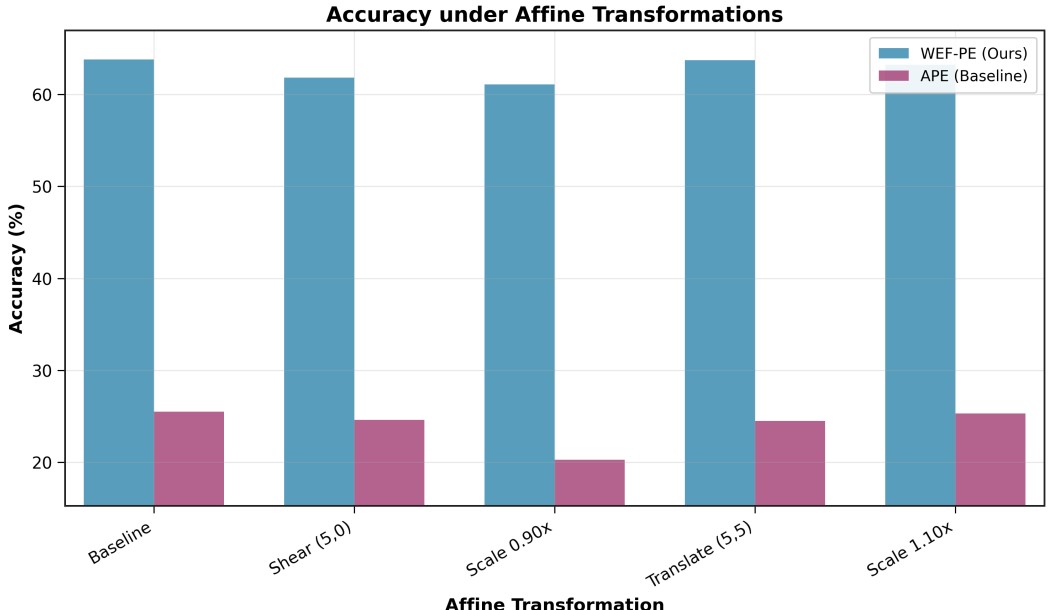

Figure 7: Classification accuracy comparison between WePE and APE baseline under affine transformations on CIFAR-100 (Krizhevsky, 2009).

### H.3 RELATIVE POSITION AWARENESS VALIDATION

To empirically validate the theoretical proposition that elliptic function addition formulas endow models with enhanced relative position awareness capabilities, we designed a dedicated auxiliary task that directly probes the spatial relationship encoding within learned patch representations. The experimental framework leverages the mathematical property that for any two spatial positions $z_i$ and $z_j$ mapped to the complex plane, their relative displacement $\sigma_z = z_j - z_i$ can be algebraically derived from the Weierstrass elliptic function values through the addition formula $\wp(z_i + \sigma_z) = f(\wp(z_i), \wp(\sigma_z), \wp'(z_i), \wp'(\sigma_z))$.

The experimental protocol extracts patch embeddings $e_i, e_j$ from pre-trained ViT-Tiny models without positional information, subsequently combining them with their corresponding positional encodings $p_i, p_j$ to form complete representations $e_i + p_i$ and $e_j + p_j$. A lightweight MLP predictor (Rumelhart et al., 1986) with two hidden layers ($192 \rightarrow 128 \rightarrow 64 \rightarrow 2$ dimensions) receives these concatenated patch representations as input and predicts the relative coordinate displacement $(\Delta x, \Delta y) = (x_j - x_i, y_j - y_i)$ in the original patch grid. We generated 8,000 training samples for each encoding scheme by randomly sampling patch pairs from CIFAR-100 test images (Krizhevsky, 2009), ensuring balanced coverage across different spatial separations within the 14×14 patch grid.

The quantitative results demonstrate substantial superiority of WePE over conventional APE (Dosovitskiy et al., 2021) across all evaluation metrics. The mean squared error reduces from 6.69 to 0.90 (86.6% improvement), mean absolute error decreases from 2.83 to 0.90 (68.1% reduction), and root mean squared error diminishes from 2.59 to 0.95 (63.4% improvement). Training dynamics reveal markedly different convergence behaviors, with WePE achieving stable convergence from an initial loss of 21.65 to 1.14 within 30 epochs, while APE converges more slowly from 30.42 to 4.14 under identical optimization settings. Error distribution analysis reveals that WePE concentrates prediction

errors within the 0-1 unit range with peaked distribution characteristics, whereas APE (Dosovitskiy et al., 2021) exhibits broader error dispersion with substantial tail probability mass extending beyond 3 units.

These empirical findings provide direct quantitative evidence that the continuous mathematical structure of elliptic functions facilitates superior spatial relationship encoding compared to discrete lookup table approaches. The substantial performance advantages validate the theoretical assertion that elliptic function addition formulas naturally embed relative positional information within absolute encodings, enabling models to extract geometric relationships through direct algebraic manipulation rather than requiring explicit learning of pairwise spatial dependencies. The 86.6% reduction in prediction error magnitude demonstrates that WePE representations inherently preserve spatial geometric properties that prove essential for precise coordinate regression tasks, supporting the broader claim that mathematically principled positional encodings provide superior inductive biases for vision transformer architectures.

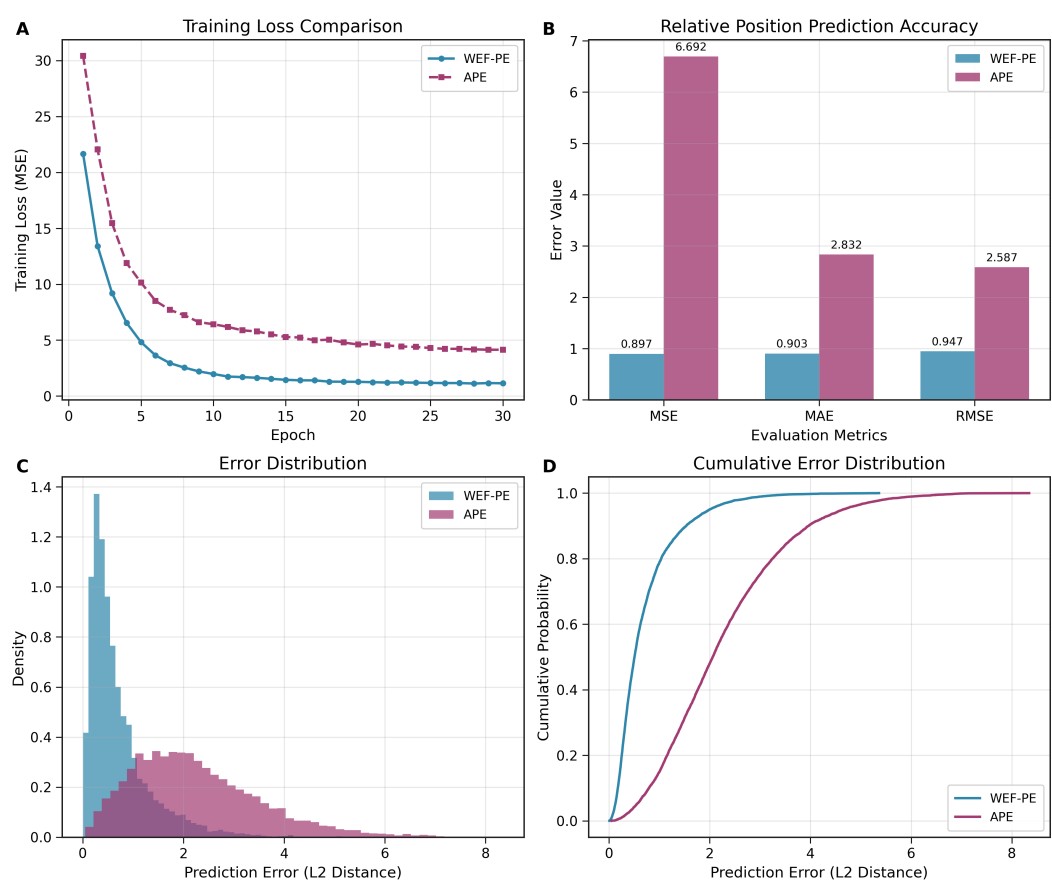

Figure 8: Quantitative validation of relative position awareness capabilities in WePE versus APE (Dosovitskiy et al., 2021) positional encodings. (A) Training loss convergence comparison showing faster and more stable learning dynamics for WePE. (B) Evaluation metrics demonstrating substantial improvements in prediction accuracy, with 86.6%, 68.1%, and 63.4% reductions in MSE, MAE, and RMSE respectively. (C) Error distribution histograms revealing concentrated low-error predictions for WePE versus dispersed error patterns in APE (Dosovitskiy et al., 2021). (D) Cumulative error distribution confirming superior prediction precision, with approximately 80% of WePE predictions achieving sub-unit accuracy compared to broader error dispersion in APE-based representations.

## H.4 Comparison with Alternative 2D Positional Encoding Schemes

To isolate the contribution of advanced mathematical structure from the fundamental advantage of preserving two-dimensional spatial relationships, we conducted systematic comparisons between WePE and alternative 2D positional encoding approaches that avoid the spatial flattening inherent in conventional APE methods (Dosovitskiy et al., 2021). The experimental framework evaluated four distinct positional encoding schemes: our proposed WePE, traditional 1D flattened APE (Dosovitskiy et al., 2021), 2D sinusoidal positional encoding extending classical Transformer sinusoidal patterns to two dimensions through independent coordinate-wise encoding, and 2D learnable grid positional encoding implementing direct parameter lookup based on spatial coordinates without intermediate flattening operations.

The 2D sinusoidal approach generates positional representations by applying sine and cosine functions independently to horizontal and vertical coordinates, interleaving the resulting values as $\text{PE}[h, w, 0 :: 4] = \sin(w \cdot \text{div\_term}_x)$, $\text{PE}[h, w, 1 :: 4] = \cos(w \cdot \text{div\_term}_x)$, $\text{PE}[h, w, 2 :: 4] = \sin(h \cdot \text{div\_term}_y)$, and $\text{PE}[h, w, 3 :: 4] = \cos(h \cdot \text{div\_term}_y)$ where div_term follows the standard inverse frequency scaling. The 2D learnable grid method maintains a trainable parameter matrix of dimensions $H \times W \times D$ enabling direct coordinate-based lookup without spatial serialization, thereby preserving explicit two-dimensional indexing throughout the encoding process.

Under identical training conditions with ViT-Tiny architecture on CIFAR-100 (Krizhevsky, 2009) for 60 epochs, the quantitative results demonstrate that WePE achieves superior performance with 60.03% final validation accuracy and 60.28% peak accuracy, followed closely by 2D sinusoidal encoding at 59.94% final and 60.19% peak accuracy. The 2D learnable grid approach yields 58.40% accuracy for both final and peak measurements, while conventional 1D flattened APE produces 58.28% final and 58.34% peak accuracy. Training dynamics reveal that WePE maintains consistently lower training loss throughout the optimization process with smoother convergence characteristics compared to alternative approaches.

The experimental findings reveal that while preservation of two-dimensional spatial structure provides measurable advantages over traditional flattening approaches, the mathematical sophistication embedded within elliptic function-based encoding yields additional performance gains beyond those attributable solely to dimensionality considerations. The modest but consistent superiority of WePE over 2D sinusoidal encoding (0.09 percentage points final accuracy improvement) validates the hypothesis that continuous mathematical structure and inherent geometric properties contribute meaningfully to positional representation quality. The relatively strong performance of 2D sinusoidal encoding compared to learnable grid methods suggests that mathematical regularity and interpretability provide advantages over pure parameter optimization in spatial encoding tasks, supporting the broader principle that principled mathematical foundations enhance neural network architectural design for vision applications.

## H.5 WePE exhibits better geometric inductive bias

To further evaluate the structural properties of our proposed WePE, we visualized the positional encodings prior to any model training, as illustrated in Figure 10. The analysis comes from two aspects: a PCA (Bishop, 2006) to reveal the embedding manifold, and a cosine similarity matrix to expose their relational structure. In PCA (Bishop, 2006) space, WePE forms a highly structured, spiral-like manifold, that faithfully preserves the original 2D spatial arrangement, as confirmed by the color gradient-encoded patch coordinates. By contrast, the APE (Dosovitskiy et al., 2021) projects into an unstructured, Gaussian-like cloud, demonstrating a complete lack of inherent spatial organization. Furthermore, the cosine similarity matrix of WePE displays a distinct, periodic, and grid-like pattern, indicating that the relationships between encodings are systematically governed by their relative spatial distances. The APE (Dosovitskiy et al., 2021) matrix, however, resembles random noise aside from the identity diagonal, confirming the absence of any pre-defined relational structure.

## H.6 Supplement to the experiment in Section 3.1 .

To validate the distance–decay property of the WePE, we designed a systematic experiment to quantitatively analyze the relationship between positional encoding interaction strength and spatial

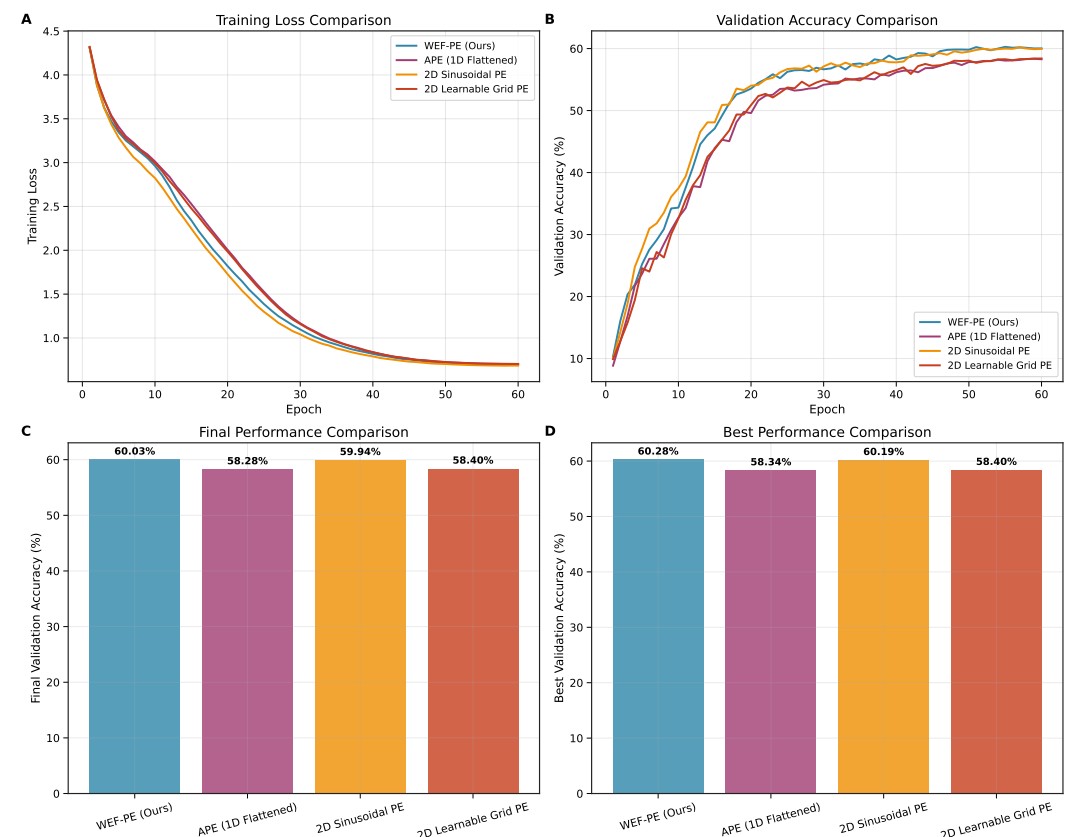

Figure 9: Comparative analysis of 2D positional encoding schemes. (A) Training loss dynamics for WePE, 1D flattened APE (Dosovitskiy et al., 2021), 2D Sinusoidal PE, and 2D Learnable Grid PE over 60 epochs on CIFAR-100 (Krizhevsky, 2009). (B) Corresponding validation accuracy curves. (C) Final validation accuracy comparison, where WePE achieves 60.03%. (D) Best validation accuracy comparison, with WePE peaking at 60.28%, demonstrating its superior performance over alternative 2D encoding strategies.

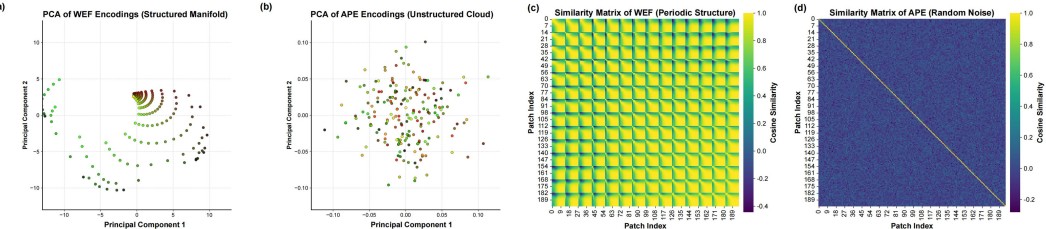

Figure 10: Structural properties of positional encodings revealed through PCA (Bishop, 2006) and cosine similarity. (a, b) PCA (Bishop, 2006) projections demonstrate WePE forms structured spiral manifolds that preserving spatial topology, while APE (Dosovitskiy et al., 2021) appears as unstructured Gaussian-like clouds. (c, d) Cosine similarity matrices reveal that WePE displays periodic, grid-like patterns reflecting systematic spatial relationships, contrasting with largely random patterns of APE (Dosovitskiy et al., 2021).

distance. The experiment was based on the ViT-Ti architecture (embed_dim=192, patch_size=16), processing input images of size $224 \times 224$, which results in a $14 \times 14$ grid of patches, totaling 196 spatial locations.

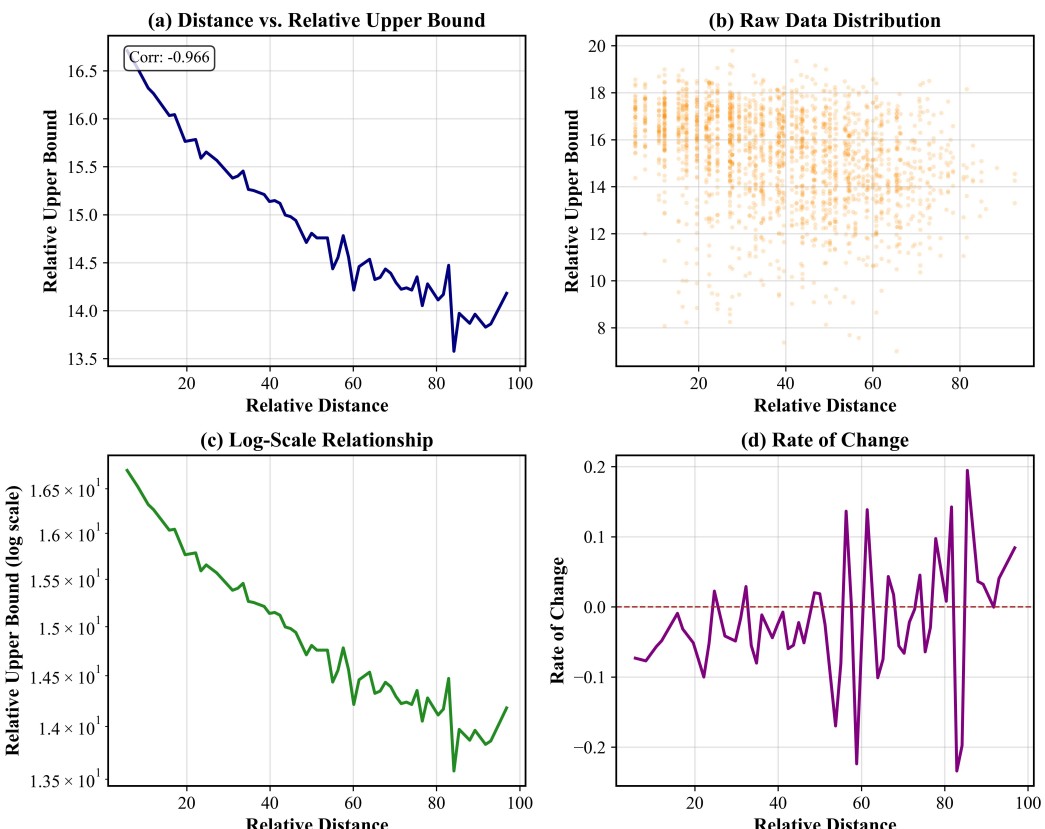

Figure 11: Quantitative analysis of distance-decay properties in WePE. (a) Scatter plot and fitted curve demonstrate strong negative correlation between relative patch distance and interaction strength. (b) Raw data distribution across 19,110 patch pairs. (c) Log-scale relationship confirming exponential decay characteristics. (d) Rate of change analysis revealing monotonic decrease in similarity with increasing spatial separation.

The core methodology involves establishing a quantitative relationship between the relative distance of all patch pairs and the interaction strength of their corresponding positional encodings. Specifically, for any two patch locations $(i_1, j_1)$ and $(i_2, j_2)$, we first compute their Euclidean distance. Subsequently, we extract their corresponding Weierstrass elliptic function positional encodings, $\mathbf{p}_{i_1,j_1}$ and $\mathbf{p}_{i_2,j_2}$, and compute their cosine similarity as a metric for interaction strength:

$$S_{i,j} = \frac{\mathbf{p}_i^\top \mathbf{p}_j}{\|\mathbf{p}_i\| \|\mathbf{p}_j\|}. \tag{67}$$

Cosine similarity, which normalizes out the influence of vector magnitudes, provides a purer reflection of directional correlation and serves as a key indicator for evaluating the quality of positional encodings.

To eliminate the dependency on image size, the distance is normalized into a relative distance:

$$d_{\text{relative}} = \frac{d_{\text{euclidean}}}{d_{\text{max}}} \times 100, \tag{68}$$

where $d_{\text{max}}$ is the maximum possible distance between any two patches in the image.

To enhance the visual interpretability of the relationship between inter-patch distance and interaction strength, we applied a linear transformation to the raw cosine similarity scores. The primary motivation for this transformation is to normalize the similarity values, which may originally occupy a narrow numerical range, into a standardized and more visually dynamic scale. This ensures

Table 6: Quantitative Analysis Results of the Distance–Interaction Strength Relationship

| Metric | Value |
| --- | --- |
| Pearson Correlation Coefficient $\rho$ | -0.966 |
| Relative Distance Range | [0, 100] |
| Relative Upper Bound Range | [13.5, 16.5] |
| Initial Interaction Strength | 16.5 |
| Final Interaction Strength | 13.8 |
| Decay Magnitude $\Delta_{\text{decay}}$ | 16.4% |
| Monotonicity Metric $\mathcal{M}$ | 87.5% |

consistent and comparable graphical representation across different experimental settings:

$$s_{\text{rel}} = C_{\text{base}} + \frac{s - s_{\min}}{s_{\max} - s_{\min}} \times C_{\text{range}}, \tag{69}$$

where $s_{\min}$ and $s_{\max}$ represent the minimum and maximum observed cosine similarity values across the entire dataset of patch pairs, respectively. The term $\frac{s - s_{\min}}{s_{\max} - s_{\min}}$ performs a min–max normalization, scaling the similarity scores to the range $[0, 1]$. In our specific analysis, we set the base constant $C_{\text{base}} = 6$ and the range constant $C_{\text{range}} = 14$, thereby mapping the original similarity scores to a new, standardized interval of $[6, 20]$. This procedure facilitates a clearer visualization of the decay trend by amplifying the dynamic range of the dependent variable.

Finally, a statistical summary is generated to distill the underlying trend from the point cloud of data. We bin the data into 80 equi-width intervals based on distance. For each of the 80 distance bins, indexed from $k = 1$ to 80, we aggregate all data points whose relative distance $d_{\text{rel}}$ falls within the bin's range $[d_k, d_{k+1}]$. We then compute the arithmetic mean of the corresponding relative upper bound values within this bin, denoted as $\bar{s}_k$. To represent the distance for each bin, we use its midpoint, calculated as $d_k = (d_k + d_{k+1})/2$. This procedure culminates in two corresponding sequences: a sequence of bin centers $\{d_k\}$ and a sequence of average relative upper bounds $\{\bar{s}_k\}$. These sequences effectively constitute a discrete function that quantitatively describes the relationship between relative distance and average interaction strength.The data of this experiment are presented in Figure 11 and Table 6.

## H.7 Further Supplementary Attention Visualizations

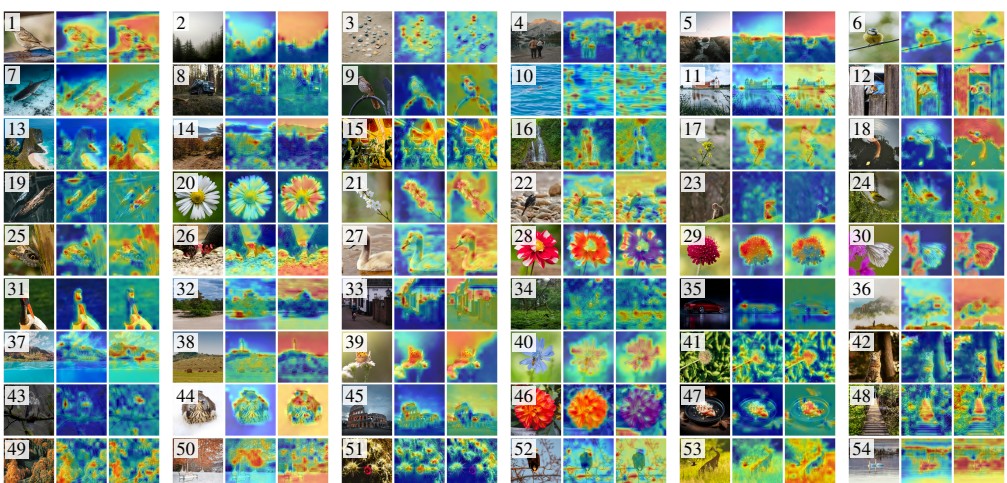

Figure 12: Further example attention maps as in Figure 4.

## H.8 ATTENTION BEHAVIOR ANALYSIS

To better understand the mechanisms underlying WePE's performance gains, we conduct a comprehensive analysis of attention behavior across all 12 Transformer layers.

Figure 13 visualizes self-attention maps from Layer 6, Head 0 for several representative query locations. The WePE model (top row) exhibits clearly stronger spatial locality than the baseline with standard learnable absolute position embeddings (bottom row). For example, when the query lies on a bottom-right patch, WePE concentrates attention on a compact set of neighboring patches, whereas the baseline distributes attention much more uniformly over the image. This qualitative difference indicates that WePE injects an explicit geometric inductive bias, encouraging the model to favor local interactions.

Figure 14 reveals that this locality originates from the structure of the positional encodings themselves. The positional similarity matrix of WePE displays a pronounced checkerboard pattern that directly reflects the doubly periodic lattice induced by the Weierstrass $\wp$-function, providing a highly regular spatial prior. Neighboring patches exhibit strongly correlated encodings, while distant patches are systematically decorrelated. In contrast, the baseline encodings produce a much more irregular similarity matrix, lacking such geometric regularity.

We next quantitatively analyze how attention strength varies with spatial distance. Figure 15 plots the mean attention weight as a function of patch distance for several layers. In early layers, WePE shows a much steeper decay of attention with distance than the baseline, indicating a significantly stronger locality bias. This distance–decay effect gradually relaxes in deeper layers, where both models become more global, allowing long-range integration while still preserving the structured initialization provided by WePE. These results confirm that the geometric properties of WePE are indeed manifested in the learned attention patterns.

Entropy analysis in Figure 16 further quantifies attention focus. We compute the Shannon entropy of each attention distribution and average across heads and tokens. In the early layers, WePE consistently yields lower entropy than the baseline, corresponding to more concentrated and less noisy attention maps that can accelerate feature learning. Entropy gradually increases with depth for both models, reflecting the expected expansion of the effective receptive field, but the WePE model remains consistently more focused.

Figure 17 investigates the effective attention range. For each query token, we sort keys by spatial distance and find the radius required to accumulate a fixed proportion (e.g., 90%) of the total attention mass. In the first layer, WePE attains this threshold within a smaller radius than the baseline, indicating a more compact local receptive field. In deeper layers, the effective range gradually expands, yielding a multi-scale behavior reminiscent of CNNs (LeCun et al., 1998)' progressively enlarging receptive fields, but achieved here through learned geometric priors rather than architectural constraints.

Finally, Figure 18 visualizes the learned patch representations using t-SNE. With WePE, patches that are spatially close in the image tend to form coherent clusters in the feature space, demonstrating that the spatial priors provided by WePE propagate into the learned representations. The baseline model exhibits substantially weaker clustering with respect to spatial location, indicating a less structured organization of features.

Taken together, these analyses show that WePE's effectiveness stems from a mathematically grounded geometric inductive bias that consistently guides attention behavior across layers: it induces strong local bias in early stages, gradually relaxes to support global integration, and ultimately leads to spatially coherent feature representations. This behavior is particularly beneficial in limited-data regimes, where strong and well-structured priors are crucial for generalization.

## H.9 EXTREME ASPECT RATIO INPUT EXPERIMENT

To assess how the proposed method behaves when the input aspect ratio departs from the standard 1:1 setting, we retrain the ViT model (Dosovitskiy et al., 2021) with WePE from scratch using several non square resolutions while keeping the architecture, patch size, optimizer, and training schedule identical to the baseline trained on $224 \times 224$ images.

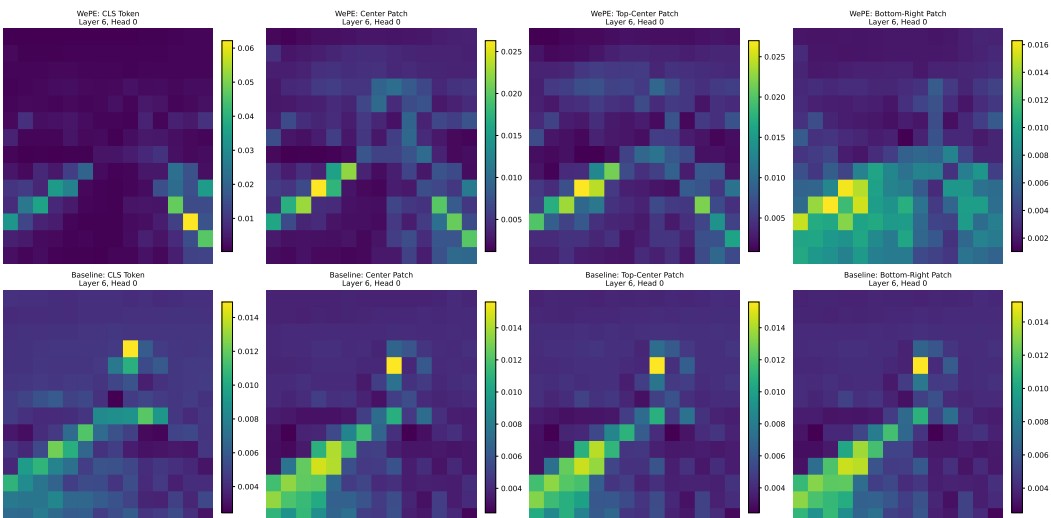

Figure 13: Attention maps comparing WePE (top) and baseline (bottom) at Layer 6, Head 0. WePE shows stronger spatial locality.

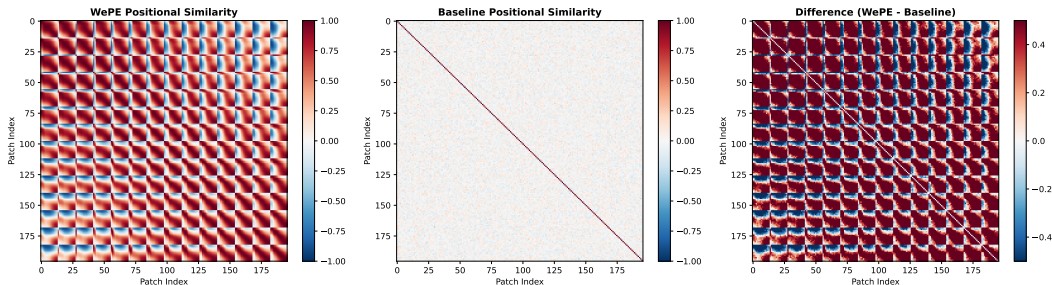

Figure 14: Positional similarity matrices. WePE exhibits checkerboard pattern encoding spatial topology; baseline lacks geometric structure.

We consider $224 \times 448$ (wide, 2:1), $448 \times 224$ (tall, 1:2), $672 \times 224$ (very tall, 1:3) and $112 \times 448$ (very wide, 4:1), which change height and width by factors of two to four and induce highly anisotropic patch grids. The aspect ratios evaluated here arguably approach or even surpass the most extreme conditions that models are likely to encounter in real-world applications. We first conducted full 120-epoch training runs for the standard square input ($224 \times 224$, 1:1) and the most extreme aspect ratio ($112 \times 448$, 4:1). As shown in Figure 19, both the training and validation accuracies are already very close to their final converged values by epoch 20, with subsequent training bringing only minor fluctuations and marginal improvements. This indicates that extending the training schedule does not alter the qualitative trends or the relative performance across different aspect ratios. Therefore, for each configuration, we report the best validation accuracy obtained at 10 and 20 epochs, together with the learned lattice parameter $\alpha_{\text{learn}}$.

As shown in the table 7, After 10 epochs all non square settings already outperform the square baseline (32.82%), achieving between 36.61% and 42.72%. After 20 epochs the models trained with moderate aspect ratios reach 51.86%, 52.28%, and 52.83% accuracy for 2:1, 1:2, and 1:3 respectively, which is on par with or slightly higher than the 51.26% obtained with 1:1 inputs. Only the extreme 4:1 configuration shows a degradation, which we attribute to the ViT architecture's (Dosovitskiy et al., 2021) inherent sensitivity to extreme patch grids (Touvron et al., 2021a; Liu et al., 2021; Heo et al., 2021). This indicates that WePE remains numerically stable for non-square inputs. Performance is negligibly affected by moderate aspect ratios, and even under extreme ratios, the model exhibits a graceful degradation rather than a collapse.

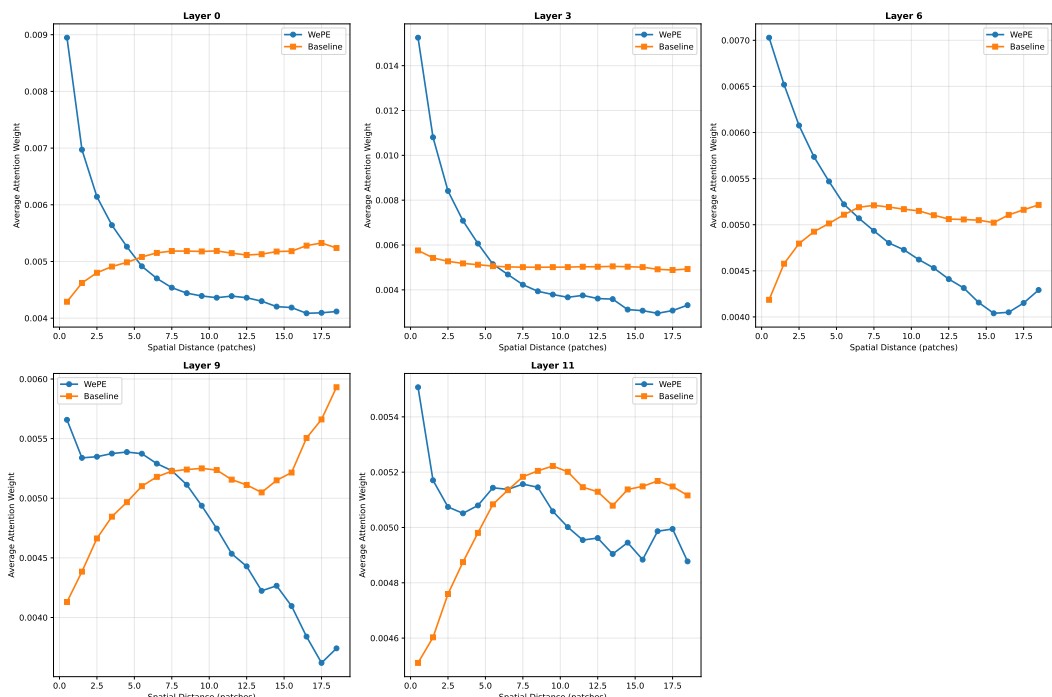

Figure 15: Attention weight vs. spatial distance. WePE shows stronger decay in early layers, converging with baseline in deeper layers.

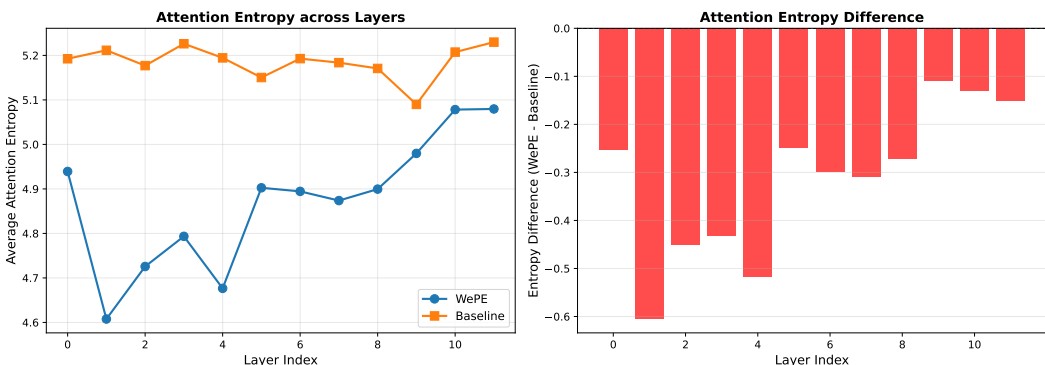

Figure 16: Attention entropy across layers. WePE maintains lower entropy in early layers (4.6-4.9 vs. 5.1-5.3), indicating focused attention.

Crucially, the learned lattice parameter remains in a narrow range across all aspect ratios, with $\alpha_{\text{learn}} \in [0.84, 0.90]$ for both training budgets, indicating that the elliptic lattice adapts automatically to changes in input shape and does not require any manual retuning even when the height to width ratio varies by a factor of four.

## H.10 SENSITIVITY OF THE FOURIER-LIKE PARAMETERS $\beta$ AND $\gamma$

In the fine-tuning adaptation of WePE (Section 2.3), the Fourier-like approximation introduces two scalar parameters, $\beta$ and $\gamma$, which are learnable in all our main experiments. To explicitly verify that the method does not rely on careful hand-tuning of these parameters, we conduct an additional sensitivity study in which we fix $\beta$ and $\gamma$ to different values and measure the impact on downstream performance.

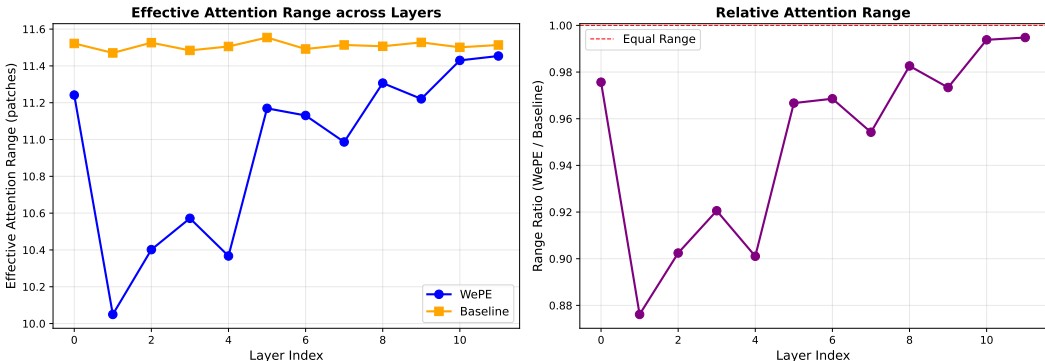

Figure 17: Effective attention range evolution. WePE contracts to 10.0 patches in Layer 1, then expands; baseline remains constant at 11.5 patches.

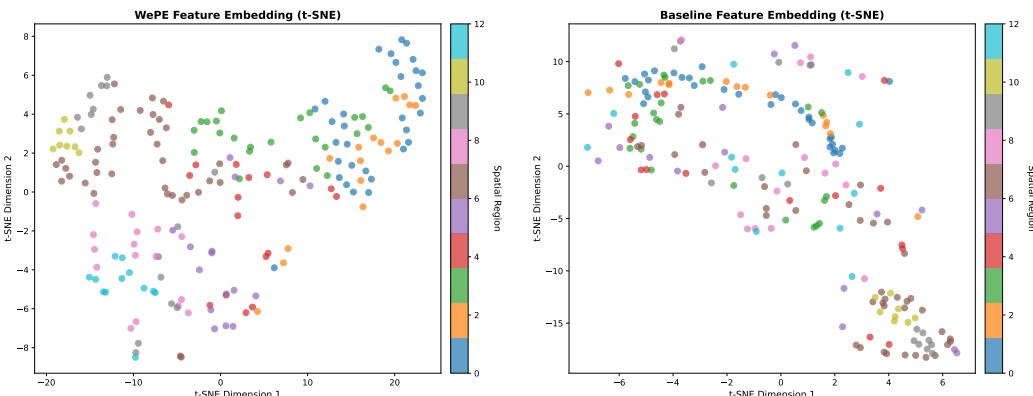

Figure 18: t-SNE of Layer 6 features colored by spatial region. WePE shows clearer spatial clustering than baseline.

We use a ViT-B/16 backbone pretrained on ImageNet-21k and fine-tune it on CIFAR-100 with our Weierstrass positional encoding. The image resolution is $224 \times 224$ and the patch size is 16. For the sensitivity experiment, we replace the learnable $\beta$ and $\gamma$ with fixed scalars and perform a $5 \times 5$ grid search over

$$\beta \in \{0.01, 0.05, 0.10, 0.20, 0.50\}, \qquad \gamma_{\text{scale}} \in \{0.01, 0.05, 0.10, 0.20, 0.50\},$$

where the $k$-th Fourier coefficient is parameterized as $\gamma_k = \gamma_{\text{scale}}/k^2$ with $K = 3$ terms. For each of the 25 configurations, we fine-tune the model for 5 epochs using AdamW with learning rate $10^{-3}$ and weight decay 0.05, and report the best test accuracy over epochs. All other settings (data augmentations, batch size, random seed, etc.) are kept fixed across the grid.

Across all 25 configurations, the mean best test accuracy is 49.93% with a standard deviation of 2.81 and a coefficient of variation of 5.6%. The minimum and maximum accuracies are 40.29% and 52.27%, respectively. The full accuracy surface is visualized as a heatmap in Fig. 20, and two sets of slices highlighting the effect of varying $\beta$ or $\gamma_{\text{scale}}$ independently are shown in Fig. 21.

We are primarily interested in the practically relevant range $\beta \geq 0.05$ and $\gamma_{\text{scale}} \geq 0.05$, which contains the settings used in our main experiments. In this $4 \times 4$ sub-grid (16 configurations), the best test accuracy lies between 50.28% and 51.59%, *i.e.* within a narrow band of only 1.31 percentage points. Both Fig. 20 and Fig. 21 exhibit a broad plateau over this region: for any fixed $\gamma_{\text{scale}} \in \{0.05, 0.10, 0.20, 0.50\}$, sweeping $\beta$ from 0.05 to 0.50 produces only mild fluctuations, and vice versa for sweeping $\gamma_{\text{scale}}$ at fixed $\beta \in \{0.05, 0.10, 0.20, 0.50\}$. These variations are comparable to the typical run-to-run noise observed in short fine-tuning.

Table 7: Performance of WePE under different aspect ratios. Baseline uses square input resolution.

| Experiment | Size | Aspect Ratio | Acc (10 ep) | $\alpha_{\text{learn}}$ (10 ep) | Acc (20 ep) | $\alpha_{\text{learn}}$ (20 ep) |
|---|---|---|---|---|---|---|
| baseline | 224×224 | 1:1 | 32.82% | – | 51.26% | – |
| wide | 224×448 | 2:1 | 42.03% | 0.8647 | 51.86% | 0.8730 |
| tall | 448×224 | 1:2 | 42.61% | 0.8792 | 52.28% | 0.9029 |
| very tall | 672×224 | 1:3 | 42.72% | 0.8605 | 52.83% | 0.8533 |
| very wide | 112×448 | 4:1 | 36.61% | 0.8449 | 46.77% | 0.8379 |

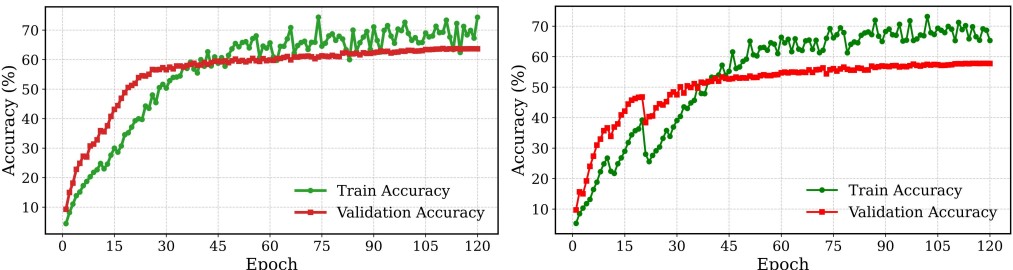

Figure 19: Training and validation accuracy curves for the model trained for the full 120 epochs under standard 1:1 input resolution (left) and extreme 4:1 resolution (right). In both cases, the validation accuracy is already close to convergence by epoch 20, with subsequent improvements being only marginal.

Noticeable degradation occurs only in the extreme corner cases with very small $\beta = 0.01$ and very small $\gamma_{\text{scale}} \in \{0.01, 0.05\}$, where the best accuracy drops to 41–40%. In this regime, the stabilizing term $1/(\|z\|^2 + \beta)$ approaches the original singular behavior of $1/\|z\|^2$, and the Fourier correction is too weak to compensate, which is precisely what the learnable $\beta$ and $\gamma$ in our main method are designed to avoid. Importantly, none of our main experiments operate in this pathological parameter regime.

Although the Fourier-like approximation introduces the additional scalars $\beta$ and $\gamma$, this experiment shows that WePE enjoys a wide region of insensitivity with respect to these parameters. In the broad practical range $\beta \in [0.05, 0.50]$ and $\gamma_{\text{scale}} \in [0.05, 0.50]$, the downstream accuracy varies by at most about 1.3 percentage points, while larger drops are confined to extreme settings that are not used in practice. Combined with the fact that $\beta$ and $\gamma$ are learned automatically in our main models, this indicates that the Fourier-like adaptation does not introduce a significant tuning burden and that the proposed WePE is robust to the specific choices of these parameters.

### H.11 OBJECT DETECTION AND IMAGE SEGMENTATION TASKS INTEGRATED WITH WEPE

To further investigate whether the geometric inductive bias introduced by WePE is beneficial for dense prediction tasks such as detection and segmentation, we first conduct a zero-shot attention–segmentation evaluation on the COCO 2017 dataset. This experiment uses only the frozen image classifier and does not perform any task-specific fine-tuning, thereby isolating the effect of the positional encoding itself.

We use the same ViT backbone as in the main experiments, instantiated with either standard APE or our WePE. The models are trained only on the classification task (no segmentation supervision) and are kept frozen during this evaluation. For the dataset, we randomly sample 50 images from the COCO 2017 validation split, together with their official instance-level annotations. All images are resized to $224 \times 224$. For each image, we convert the COCO instance masks into a semantic segmentation mask by taking the union of all annotated foreground instances (category IDs > 0) and treating all remaining pixels as background.[1] This yields a binary ground-truth mask $M \in \{0, 1\}^{H \times W}$ for each image. Given a frozen ViT, we compute attention-based localization maps. Concretely, for each transformer layer we average the multi-head attention matrices, add the identity

---

[1] We only require a binary foreground/background separation in this evaluation, so the exact instance identity is not used.

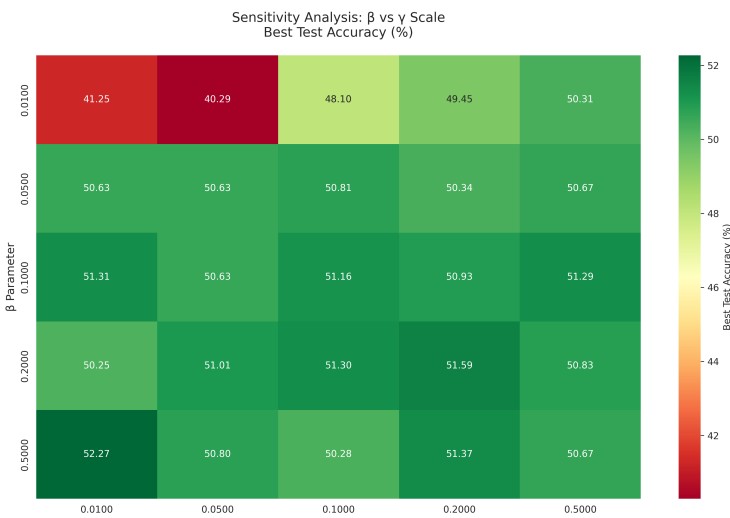

Figure 20: Sensitivity of WePE to the Fourier-like parameters $\beta$ and $\gamma_{\text{scale}}$ on CIFAR-100 with a ViT-B/16 backbone. Each cell shows the best test accuracy (%) over 5 epochs of fine-tuning.

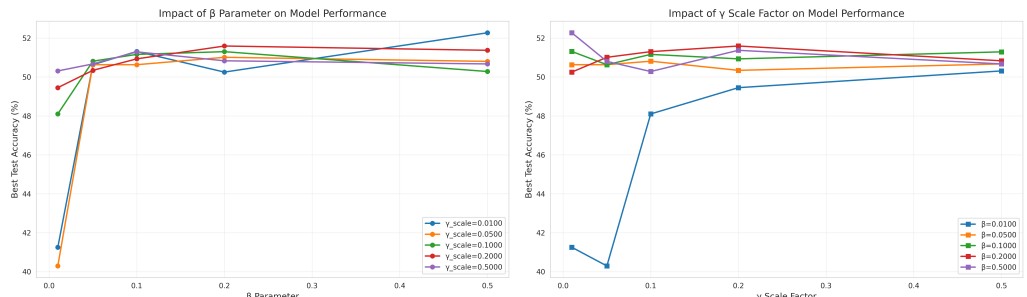

Figure 21: Slices of the sensitivity surface in Fig. 20. Varying $\beta$ while keeping $\gamma_{\text{scale}}$ fixed, and varying $\gamma_{\text{scale}}$ while keeping $\beta$ fixed. In the practical range $\beta \geq 0.05$, $\gamma_{\text{scale}} \geq 0.05$ the curves are relatively flat, indicating low sensitivity.

connection, renormalize along the last dimension, and then multiply the resulting matrices across layers to obtain a single attention matrix linking the [CLS] token to all patch tokens. The resulting vector is reshaped into a $h \times w$ grid and bilinearly upsampled to the input resolution, yielding a dense attention map $A \in [0, 1]^{H \times W}$.[2] The same pipeline is applied to both the APE-based and WePE-based models, without any further tuning. We evaluate how well the attention map $A$ aligns with the ground-truth mask $M$ using two complementary metrics:

- IoU. We threshold the attention map at $0.5$ to obtain a predicted binary mask $\hat{M}$ and compute the standard intersection-over-union (IoU) between $\hat{M}$ and $M$: $\text{IoU} = \frac{|\hat{M} \cap M|}{|\hat{M} \cup M|}$.

- Point-biserial correlation. We treat the attention values $A$ as continuous scores and the mask $M$ as a binary variable, and compute the point-biserial correlation coefficient between them. This measures how strongly high-attention pixels correlate with foreground regions.

Both metrics are computed per image. we report the mean and standard deviation over the 50 images, and perform a paired $t$-test between WePE and APE. Table 8 summarizes the quantitative results. WePE improves the mean IoU from $0.2213$ to $0.2408$, a gain of $+0.0195$, and increases the mean point-biserial correlation from $0.0255$ to $0.0734$. The IoU improvement is statistically significant

---

[2]In practice we apply a small discard ratio to extremely low attention values before renormalization, this has negligible impact on the conclusions.

| Method | IoU ↑ | Corr. ↑ |
|---|---|---|
| APE | $0.2213 \pm 0.1652$ | $0.0255 \pm 0.2261$ |
| WePE (ours) | $\mathbf{0.2408} \pm 0.1835$ | $\mathbf{0.0734} \pm 0.1929$ |

Table 8: Zero-shot attention–segmentation evaluation on COCO 2017. WePE yields better alignment between attention maps and ground-truth masks than standard APE, despite no task-specific training.

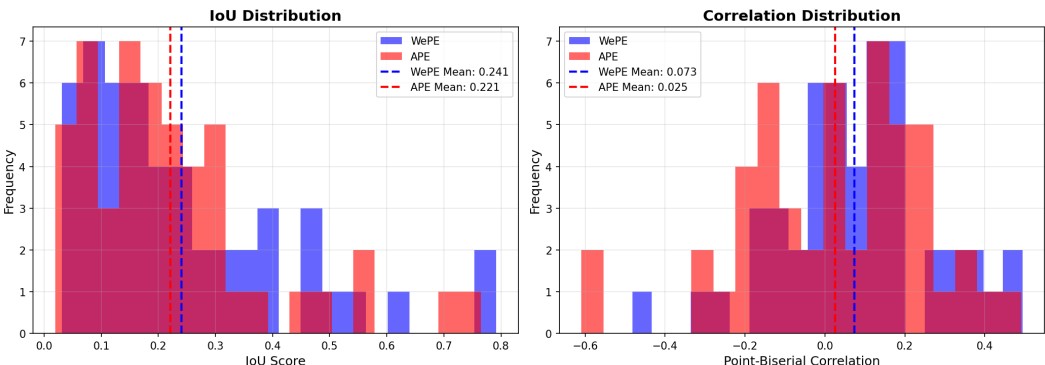

Figure 22: Distributions of IoU (left) and point-biserial correlation (right) between attention maps and COCO 2017 segmentation masks, comparing WePE and APE. Dashed lines denote the mean of each method.

under a paired $t$-test ($t = 2.41$, $p = 0.0197$), while the correlation improvement shows a consistent positive trend but is not statistically significant at the $5\%$ level ($t = 1.20$, $p = 0.23$). Figure 22 visualizes the distribution of IoU and correlation scores for both methods, and Figure 23 provides qualitative examples.

Overall, these results indicate that WePE produces attention maps that are more tightly aligned with object regions in COCO images, even without any segmentation or detection training. Since spatially aligned attention is a key prerequisite for successful detection and segmentation systems, this zero-shot evaluation provides additional evidence that the geometric prior introduced by WePE is beneficial beyond image classification.

Furthermore, to verify that the geometric inductive bias of WePE also transfers to large-scale object detection, we integrate WePE into the famous ViTDet framework Li et al. (2022), a strong and widely adopted baseline for plain ViT backbones (Dosovitskiy et al., 2021), and evaluate on the COCO 2017 dataset. This experiment keeps all components of ViTDet unchanged except for the positional encoding inside the ViT backbone (Dosovitskiy et al., 2021), providing a controlled comparison between standard APE and WePE. We follow the official ViTDet recipe on COCO 2017. The detector is Mask R-CNN or Cascade Mask R-CNN with a simple feature pyramid built on top of a plain ViT backbone (Dosovitskiy et al., 2021), as in Li et al. (2022). We use the default $1024 \times 1024$ large-scale jittering (LSJ) data augmentation Li et al. (2022), 100-epoch training schedule, AdamW optimizer, and stochastic depth, exactly matching the public ViTDet configuration. For the backbone we consider ViT-B and ViT-L (Dosovitskiy et al., 2021) pretrained with MAE (He et al., 2021) on ImageNet-1K, using the official checkpoints. In the WePE configuration, we replace the APE module in the ViT backbone by our WePE parameterization while leaving all other architecture and optimization hyperparameters identical. In particular, the feature pyramid, RPN, ROI heads, and loss functions are unchanged. This ensures that any difference in detection performance can be attributed to the positional encoding. We train all models on $\texttt{train2017}$ and report bounding-box AP ($\text{AP}_{\text{box}}$) and instance segmentation AP ($\text{AP}_{\text{mask}}$) on $\texttt{val2017}$ using the standard COCO evaluation protocol. Table 9 summarizes the experimental results. Crucially, WePE consistently outperforms the baseline, indicating that the inductive bias afforded by WePE successfully translates to the challenging object detection task, thereby further validating its broad generalization capability.

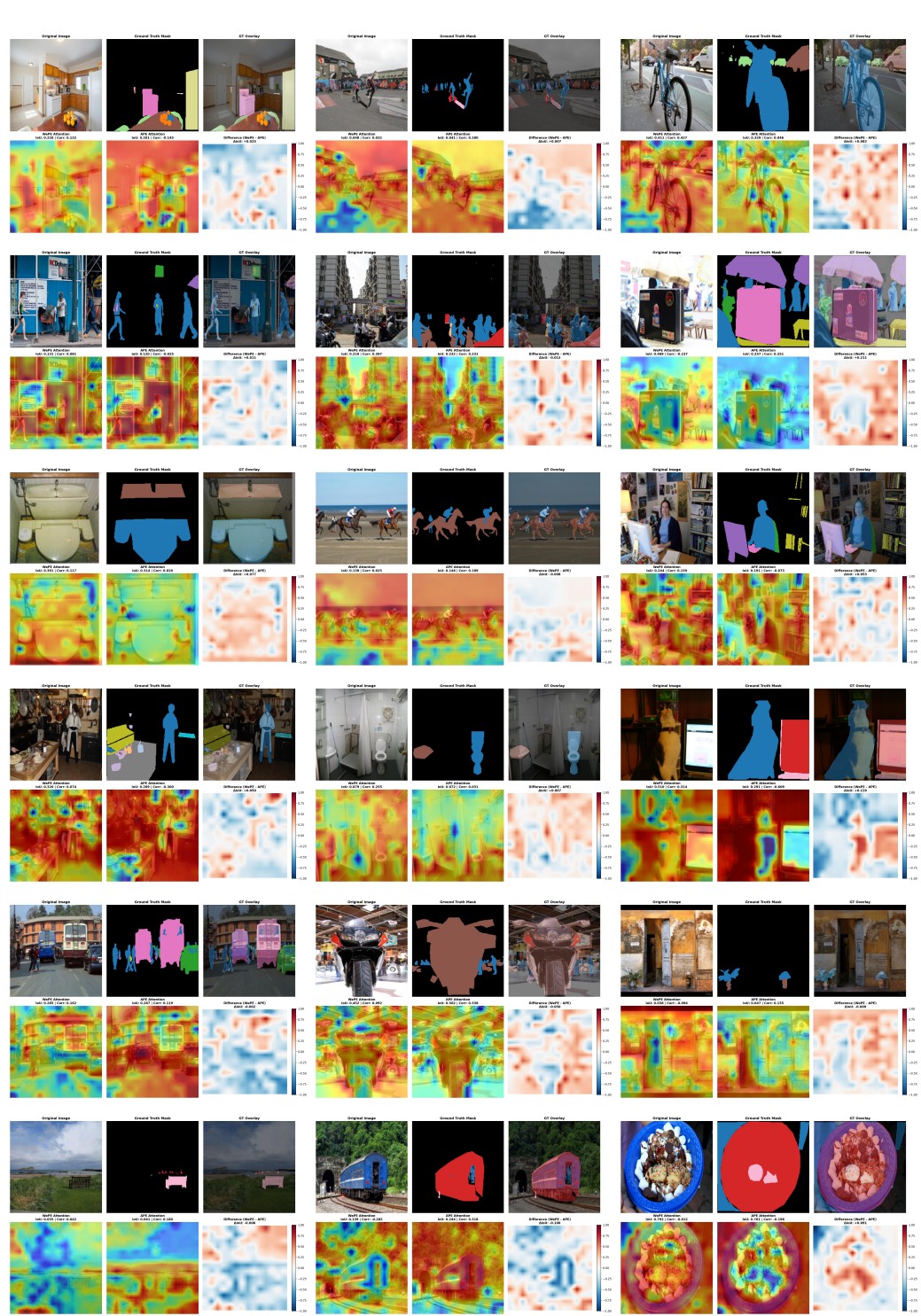

Figure 23: Qualitative examples from COCO 2017 showing the original image, ground-truth mask, and attention overlays from WePE and APE (Dosovitskiy et al., 2021). WePE consistently produces attention patterns that better cover the foreground objects and suppress background clutter.

| Backbone & PE | Mask R-CNN | | Cascade Mask R-CNN | |
|---|---|---|---|---|
| | $AP_{box}$ | $AP_{mask}$ | $AP_{box}$ | $AP_{mask}$ |
| ViT-B + APE (ViTDet baseline) | 51.6 | 45.9 | 54.0 | 46.7 |
| ViT-B + WePE (ours) | 52.9 | 46.1 | 54.7 | 47.3 |
| ViT-L + APE (ViTDet baseline) | 55.6 | 49.2 | 57.6 | 49.8 |
| ViT-L + WePE (ours) | 56.2 | 49.5 | 58.2 | 50.1 |

Table 9: COCO 2017 detection results with ViTDet-style plain backbones. All models use MAE-pretrained (He et al., 2021) ViT-B/L backbones (Dosovitskiy et al., 2021) and follow the official ViTDet configuration Li et al. (2022).

### H.12 MODEL SHORTCOMINGS AND FUTURE IMPROVEMENT DIRECTIONS

At the end of this paper, we conduct an explicit failure analysis on WePE to identify potential deficiencies in the algorithm.

We first quantify how classification errors relate to the spatial distribution of the most attended patches in the last transformer layer. We use the ViT-Tiny + WePE model at a resolution of $224 \times 224$ with a $14 \times 14$ patch grid. For each test image $x$, we extract the attention weights from the last multi-head self-attention block and average over heads to obtain an attention matrix $A \in \mathbb{R}^{L \times L}$, where $L = 1 + N$ and $N = 14 \times 14$ is the number of patch tokens. We focus on the attention from the class token to patch tokens, *i.e.* the row $A_{\text{cls} \rightarrow \text{patch}} \in \mathbb{R}^N$. We select the top-$K$ patches with the largest weights (we use $K = 5$ throughout) and treat their grid coordinates $\{p_1, \ldots, p_K\}$ on the $14 \times 14$ lattice as the discriminative region set for this image. We then define a scalar spatial separation score $S(x)$ as the mean pairwise Euclidean distance between these $K$ patches on the lattice: $S(x) = \frac{2}{K(K-1)} \sum_{1 \leq i < j \leq K} d(p_i, p_j)$, $d(p_i, p_j)$ is the Euclidean distance on the $14 \times 14$ grid.

Intuitively, small $S(x)$ indicates that the model concentrates its attention on a compact region, while large $S(x)$ corresponds to highly dispersed evidence that requires long-range reasoning. For every test image we record the pair $(S(x), \mathbb{1}[\hat{y}(x) \neq y])$, where $\hat{y}(x)$ is the model prediction and $y$ is the ground-truth label. We partition the range of $S$ into $B$ bins (we use $B = 20$ with robust percentiles to avoid outliers) and compute, for each bin $b$, the empirical error rate $\text{Err}(b) = \frac{1}{|\mathcal{D}_b|} \sum_{x \in \mathcal{D}_b} \mathbb{1}[\hat{y}(x) \neq y]$, together with the binomial standard deviation as an error bar. We also report the number of samples $|\mathcal{D}_b|$ per bin to distinguish reliable statistics from low-sample regimes.

Figure 24 shows the resulting classification error rate on the test set vs. spatial separation curve. Most test samples ($\approx 98.8\%$) fall into the range $S \in [2, 10]$, corresponding to moderately dispersed discriminative regions. Within this regime, the error rate remains low and stable between $0.31$ and $0.38$, and the fitted trend is nearly flat. This indicates that WePE handles the vast majority of natural images well, even when the evidence spans multiple mid-range patches. In contrast, only a small fraction of samples ($\approx 1.2\%$) exhibit very large separation scores ($S > 10$), meaning that the attended patches are spread across distant parts of the image. In this high-separation regime the error rate increases sharply, and several bins approach an error rate close to $1.0$. These cases correspond to images that require unusually long-range integration of visual cues.

Immediately after, To visually inspect how WePE allocates attention across the image and to provide explicit failure diagnostics, we generate qualitative heatmaps from the last transformer layer. For each test image $x$, we register a forward hook on the last multi-head self-attention block and obtain the attention tensor $A \in \mathbb{R}^{B \times H \times L \times L}$, where $B$ is the batch size, $H$ the number of heads, and $L = 1 + N$ the sequence length (class token plus $N = 14 \times 14$ patch tokens). We average over heads and select the row corresponding to the class token, yielding a vector $a_{\text{cls} \rightarrow \text{patch}} \in \mathbb{R}^N$. We reshape this vector into a $14 \times 14$ attention map on the patch lattice and upsample it to the input resolution using bilinear interpolation. The original image is de-normalized to $[0, 1]$ and the attention map is overlaid as a semi-transparent heatmap. We collect both correctly classified samples and failure cases (misclassified samples), spanning diverse object categories and spatial layouts. Representative examples are shown in Figure 25. For correctly classified images, WePE consistently produces structured and spatially coherent attention patterns: high responses concentrate on semantically meaningful parts of the

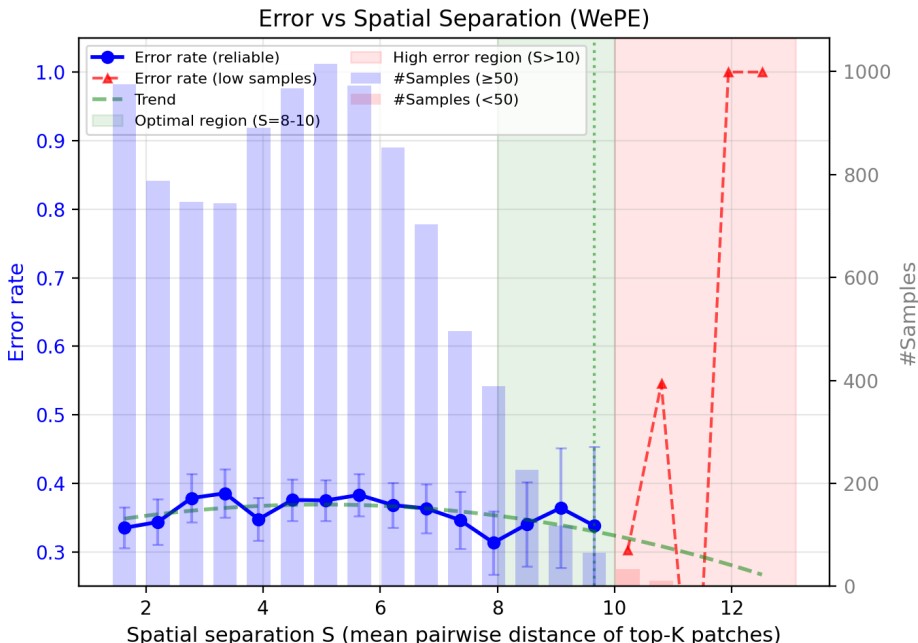

Figure 24: Error vs. spatial separation for WePE. The blue curve shows the classification error rate on the test set as a function of the spatial separation score $S$, with error bars indicating the binomial standard deviation. Bars on the secondary axis denote the number of samples per bin, and the green/red shaded regions highlight respectively the mid-range regime with stable low error and the rare high-separation regime where errors increase.

object (e.g., the head and torso of an animal, the body of a vehicle, or the base of a lamp), while background regions receive very low attention. Moreover, attention rarely collapses to a single patch; instead, it forms several mid-range clusters covering multiple object parts. This behavior matches the intended inductive bias of WePE, which encourages locality and moderate-range coupling rather than purely global or purely nearest-neighbor interactions. Failure cases reveal complementary behavior. When the discriminative evidence is either extremely fragmented or heavily entangled with cluttered background, the attention maps sometimes highlight only a subset of the relevant regions or partially drift to salient but class-irrelevant structures. These patterns are consistent with the quantitative error–vs.–separation analysis: WePE is most reliable when the key cues form one or a few coherent regions, and struggles when correct classification requires jointly reasoning over many distant or weakly localized cues.

**Correct Predictions**

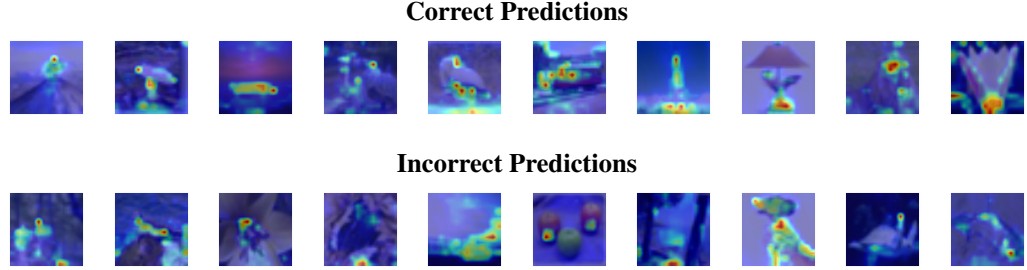

**Incorrect Predictions**

Figure 25: Qualitative attention heatmaps for WePE.

We next study how WePE distributes attention as a function of spatial distance between patches, providing a complementary, more global diagnostic. Using the same model and test set, we again hook the last self-attention block and obtain the attention tensor $A \in \mathbb{R}^{B \times H \times L \times L}$. We discard the class token and keep only patch–patch attention, resulting in $A_{\text{pp}} \in \mathbb{R}^{B \times H \times N \times N}$ with $N = 14 \times 14$.

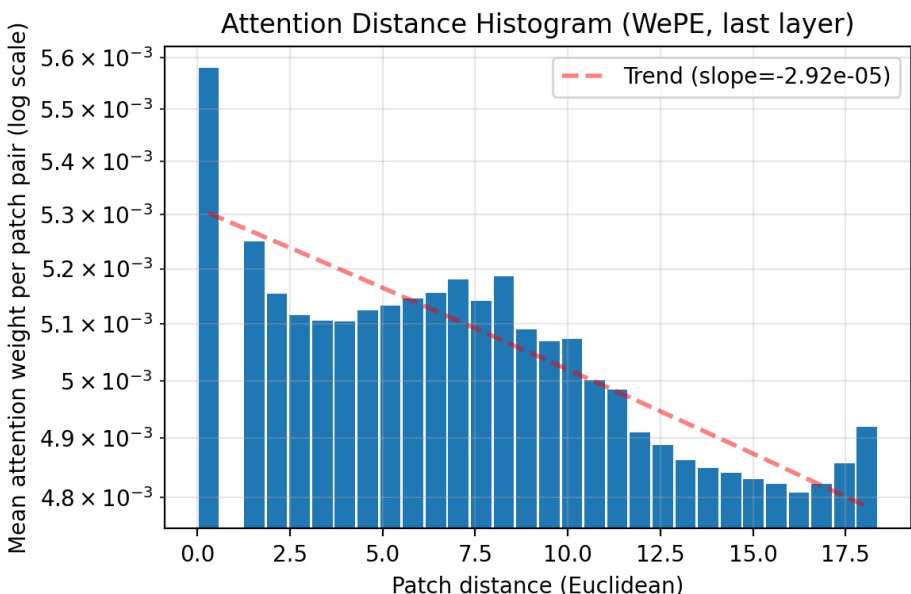

Figure 26: Attention–distance histogram for WePE (last layer). Mean attention weight per patch pair as a function of Euclidean distance on the $14 \times 14$ patch lattice. The dashed line shows a least-squares linear fit with a negative slope.

On the $14 \times 14$ patch lattice, we assign each patch an integer coordinate $(i, j)$ and precompute the Euclidean distance between all patch pairs, forming a matrix $D \in \mathbb{R}^{N \times N}$ with entries $d_{uv} = \|p_u - p_v\|_2$. We then discretize the range of distances into $M$ bins and, for each bin, aggregate the mean attention weight per patch pair:

$$\bar{A}(b) \;=\; \frac{1}{|\mathcal{P}_b|} \sum_{(u,v) \in \mathcal{P}_b} \frac{1}{BH} \sum_{b=1}^{B} \sum_{h=1}^{H} A_{\mathrm{pp}}^{(b,h)}(u, v), \tag{70}$$

where $\mathcal{P}_b$ is the set of patch pairs whose distance falls into bin $b$. We finally plot $\bar{A}(b)$ against the corresponding bin centers and optionally fit a least-squares line to estimate the global trend. Fig. 26 reports the resulting attention–distance histogram. Although the absolute variation is modest, the fitted trend exhibits a clear negative slope: the mean attention weight per patch pair decreases as the spatial distance increases. This confirms that WePE indeed introduces a distance-aware inductive bias: interactions between nearby and mid-range patches are slightly favored over very long-range connections. At the same time, the decay is gentle rather than abrupt, indicating that WePE still allows non-negligible long-range attention, in line with the good performance on samples whose discriminative cues are moderately dispersed. We also notice a small uptick in the last one or two distance bins. This edge effect is largely due to the small number of patch pairs at maximal distances on the finite $14 \times 14$ grid, which makes the estimate in those bins noisy. Nonetheless, the overall monotonic trend remains clear when considering the full range of distances. This histogram analysis is inherently global: it aggregates over all images, layers, and heads, and therefore cannot capture head-specific strategies or class-dependent patterns. In addition, we only analyze the final layer; earlier layers may exhibit stronger locality, which is blurred by this averaging. Finally, the current WePE design imposes a fixed distance-decay profile that is shared across all samples, which partly explains why the model underperforms on rare cases requiring extremely long-range reasoning. In future work, we aim to extend WePE towards adaptive and multi-scale distance priors to better accommodate images whose discriminative cues are either unusually local or unusually global.

