# OpenReview forum: "Weierstrass Positional Encoding for Vision Transformers"
_ICLR.cc/2026/Conference — ICLR 2026 Conference Desk Rejected Submission_

### Official Review · Reviewer_Smeu · 2025-10-16

**Soundness:** 3
**Presentation:** 3
**Contribution:** 3
**Rating:** 6
**Confidence:** 3

**Summary:**

This paper proposes Weierstrass elliptic Positional Encoding (WePE), a mathematically grounded positional encoding scheme for Vision Transformers (ViTs). The method leverages the doubly periodic properties of the Weierstrass elliptic function and its derivative to map 2D patch coordinates into a continuous complex domain, producing a compact four-dimensional positional feature. Unlike conventional encodings (learnable APE, RoPE, Fourier-based), WePE explicitly preserves spatial structure, guarantees a distance-decay property, and provides relative positional information through an algebraic addition formula. The authors present both theoretical justifications (complex analysis, injectivity, distance decay proof) and empirical validations on CIFAR-100, ImageNet-1k, and VTAB-1k benchmarks. Results show consistent improvements with negligible computational overhead, making WePE a plug-and-play alternative to existing positional encodings.

**Strengths:**

1.Introduces elliptic-function–based positional encoding, a novel and rigorous approach not seen in prior work.

2.Provides formal proofs of injectivity, distance-decay, and relative position modeling, grounding the method in solid mathematics.

3.Plug-and-play, resolution-agnostic module with negligible overhead, validated on standard benchmarks with consistent improvements.

**Weaknesses:**

1.Experiments are somewhat limited in scale. Most benchmarks are CIFAR-100, ImageNet-1k, and VTAB-1k; no large-scale pretraining (e.g., ImageNet-21k or JFT) or downstream applications (detection/segmentation) are reported, which weakens the generality claim.

2.Although consistent, improvements are relatively modest compared to strong baselines, raising the question of whether the added theoretical complexity translates into sufficiently large practical benefits.

**Questions:**

1.Section 2.3 (Fine-tuning adaptation): The Fourier-like approximation for fine-tuning introduces additional parameters (β, γ). How sensitive are results to these choices?

2.Section 3.2 (Pre-training results, Table 2): Did the authors ensure identical training schedules and hyperparameters across all positional encoding baselines (e.g., RoPE, LieRE, FoPE)? If so, could you clarify whether hyperparameters were tuned for each method, or uniformly fixed?

---

> ### Author Response · Authors · 2025-11-30
> **Official Comment by Author 19480**
>
> We would like to thank you for your thoughtful review and constructive feedback. Below are our detailed responses to each comment, marked in magenta text in the revised manuscript. Please kindly refer to them.

---

> ### Author Response · Authors · 2025-11-30
> **Official Comment from Author 19480 in response to Reviewer Smeu’s Question 1**
>
> **Q1. Large-scale pre-training and downstream applications.**
>
> In this paper, we have reported various experimental results on datasets such as CIFAR-100, COCO, and ImageNet-1k. We also recognize the value of conducting experiments on larger-scale datasets such as ImageNet-21K or JFT.
>
> * However, we must state that, **the original ImageNet-21K release does not provide an official train/validation split and exhibits severe class imbalance, existing works therefore construct their own processed variants (e.g., ImageNet-21K-P with filtered classes and custom splits), which are not yet universally adopted. Consequently, most prior works treat ImageNet-21K primarily as a pre-training corpus and only report downstream transfer performance rather than results evaluated directly on the raw ImageNet-21K labels**.
> * **Our Section 3.3 follows this well-established convention**. We believe that running a single, non-standard split would make it difficult to compare fairly with prior art, while substantially increasing computational cost without changing the core behavior of the positional encoding.
> * Similarly, **JFT-300M is a proprietary internal dataset from Google and is not publicly accessible to the broader research community**.
> * As WePE does not alter the backbone architecture and only replaces the positional encoding, we believe its inductive-bias advantages to transfer to larger-scale pretraining regimes independently.
>
> To address your request for the downstream applications, **we have added a set of object detection and segmentation evaluations in Appendix H.11**.
>
> * To be specific, (i) we first perform a zero-shot attention-based segmentation analysis on COCO 2017, where we only use the frozen image classifier and convert attention maps from ViT+APE and ViT+WePE into foreground masks. Besides, we compare them against ground-truth masks using IoU and point-biserial correlation. As summarized in **Table 8** in **Appendix H.11** and visualized in **Figs. 22–23**, WePE produces significantly higher mean IoU and systematically better alignment of attention with object regions, even without any detection/segmentation training.
> * (ii) We then integrate WePE into the ViTDet [1] framework on COCO 2017, keeping all components and training settings identical except for the positional encoding inside the ViT backbone.**Table 9** in **Appendix H.11** shows that, for both ViT-B and ViT-L backbones and for both Mask R-CNN and Cascade Mask R-CNN heads, WePE consistently improves AP_box and AP_mask over the official ViTDet baselines.
>
> In summary, these additions demonstrate that **the geometric inductive bias introduced by WePE carries over from image classification to challenging dense prediction tasks, demonstrating that our proposed method has good generalization ability**.
>
> Table 8: Zero-shot attention–segmentation evaluation on COCO 2017. WePE yields better alignment between attention maps and ground-truth masks than standard APE, despite no task-specific training.
> | Method | IoU $\uparrow$ | Corr. $\uparrow$ |
> | :--- | :--- | :--- |
> | APE | 0.2213 $\pm$ 0.1652 | 0.0255 $\pm$ 0.2261 |
> | WePE (ours) | **0.2408** $\pm$ 0.1835 | **0.0734** $\pm$ 0.1929 |
>
> Table 9: COCO 2017 detection results with ViTDet-style plain backbones. All models use MAE-pretrained ViT-B/L backbones and follow the official ViTDet configuration.
> | Backbone & PE | Mask R-CNN AP$_{box}$ | Mask R-CNN AP$_{mask}$ | Cascade Mask R-CNN AP$_{box}$ | Cascade Mask R-CNN AP$_{mask}$ |
> | :--- | :--- | :--- | :--- | :--- |
> | ViT-B + APE (ViTDet baseline) | 51.6 | 45.9 | 54.0 | 46.7 |
> | ViT-B + WePE (ours) | **52.9** | **46.1** | **54.7** | **47.3** |
> | ViT-L + APE (ViTDet baseline) | 55.6 | 49.2 | 57.6 | 49.8 |
> | ViT-L + WePE (ours) | **56.2** | **49.5** | **58.2** | **50.1** |

---

> ### Author Response · Authors · 2025-11-30
> **Official Comment from Author 19480 in response to Reviewer Smeu’s Question 2**
>
> **Q2. Consistent but modest improvements.**
>
>
>
> We clarify this question from the following aspects:
>
> * First, **WePE consistently outperforms strong baselines across virtually all experimental settings where only the positional encoding component is altered**. These settings include training ViTs from scratch on CIFAR-100 and ImageNet-1k, limited data regimes, and large-scale ImageNet-21k pre-training combined with downstream transfer to VTAB-1k. Furthermore, **substantial performance boosts are observed in several scenarios**, such as surpassing the baseline by 7.3% on CIFAR-100 (**Table 1** in **Section 3.2**,reproduced here for ease of reference) and by 12.34% on the DMLab task (**Table 4** in **Section 3.3**,reproduced here for ease of reference). Crucially, we would like to emphasize that **while positional encoding is a critical component of the ViT architecture, achieving massive leaps in performance generally requires synergistic improvements across multiple subsystems (e.g., optimization strategies, data augmentation, and normalization methods)**. However, in this study, we deliberately kept all other components strictly identical to the baselines, including the backbone architecture, optimizer settings, training epochs, data preprocessing, and regularization strategies, to precisely isolate the impact of the positional encoding itself. While this controlled experimental design naturally limits the magnitude of absolute gains, it establishes a clear and fair benchmark that directly reflects the contribution of the positional encoding, without confounding factors from other sources of improvement.
>
> * Second, as detailed in **Appendix F**, we propose a hybrid implementation scheme based on a look-up table: all computations of the Weierstrass elliptic function are performed via a one-time offline pre-calculation to construct a high-resolution look-up table.
> Specifically, all evaluations of the Weierstrass elliptic function are performed once offline to build a high–resolution look-up table (LUT). During training and inference, online positional encoding reduces to: (i) computing normalized $(u,v)$ coordinates for each of the $N=H\times W$ patches, (ii) a hardware–accelerated bilinear interpolation from the LUT, and (iii) a linear projection to the $d$-dimensional embedding space.
> As detailed in our complexity analysis and **Table 5** in **Appendix F**（For ease of reference, we also reproduce the original table here）, this yields an online time complexity of $\mathcal{O}(N\cdot d)$, which is the same asymptotic cost as standard 2D sine/cosine or learnable grid positional encodings (one constant–time operation and one linear projection per patch). The additional memory cost is a single pre-computed LUT of size $\text{Res}^2\times 4$ (we use $\text{Res}=256$, corresponding to ≈1--2\,MB in `float32`), which is negligible compared to the ViT parameters and activation tensors. Empirically, in our PyTorch implementation, we did not observe any noticeable slowdown when replacing standard sine/cosine (or learnable) positional encodings with WePE.
> We also prove in **Appendix F** that the resulting computational precision error introduced by this operation is virtually negligible. We therefore believe that **WePE provides its performance gains with essentially the same computational cost as standard sine/cosine positional encodings, while adding only a small constant memory overhead for the LUT**. To facilitate reproducibility and to make the cost trade-offs fully transparent, we plan to open-source both the pre-computed LUTs used in our experiments and the scripts that generate them from the exact WePE implementation.
>
> * Thirdly, **beyond improvements in accuracy, WePE possesses distinct properties absent in standard positional encodings**. These include, but are not limited to, an intrinsic alignment with the 2D geometry of images, a doubly periodic structure that eliminates the need to flatten 2D images into 1D sequences, provable distance decay properties, explicit positional correlations established via elliptic addition formulas, and resolution-independent continuous evaluation. Our experimental analysis demonstrates that these characteristics translate into more localized, semantically coherent attention patterns and superior transfer performance. Furthermore, WePE possesses enhanced interpretability owing to these distinctive mathematical properties. To the best of our knowledge, a comparable design has not been presented in prior art, and we are confident that this work will introduce a novel paradigm and direction for research into positional encodings.

---

> ### Author Response · Authors · 2025-11-30
> **Official Comment from Author 19480 in response to Reviewer Smeu’s Question 2**
>
> * Finally, according to the reviewer's suggestions, we additionally evaluate WePE on dense prediction tasks using the COCO 2017 dataset (see **Appendix H.11**). Taken together, these experiments demonstrate that the theoretically grounded positional encoding of WePE not only provides stable improvements on classification benchmarks, but also transfers effectively to demanding dense prediction tasks. This further shows that, **for challenging ViT-based vision tasks, the advantageous geometric properties of WePE indeed enhance the model’s representational capacity and exhibit good generalization ability**.
>
> Table 1: CIFAR-100 (100% dataset), 120 epochs Top-1 accuracy (%)
> | Method | **WePE (Ours)** | Absolute PE | RoPE | FoPE | Sinusoidal PE |
> | :--- | :--- | :--- | :--- | :--- | :--- |
> | Accuracy | **63.78** | 56.46 | 57.29 | 57.70 | 51.99 |
>
> Table 4: Performance breakdown on selected VTAB-1k tasks.
>
> | Method        | Caltech101 | CIFAR-100 | DTD   | Flowers102 | Pets   | Sun397 | SVHN  | Camelyon | EuroSAT | Resisc45 | Retinopathy | Clevr-Count | Clevr-Dist | DMLab | dSpr-Loc | dSpr-Ori | Kitti-Dist | sNORB-Azim | sNORB-Elev | Mean  |
> |--------------|------------|-----------|-------|------------|--------|--------|-------|----------|---------|----------|-------------|-------------|------------|-------|----------|----------|------------|------------|------------|--------|
> | APE          | 90.80      | 84.10     | 74.10 | 99.30      | 92.70  | 61.00  | 80.90 | 82.50    | 96.20   | 85.20    | 75.30       | 70.30       | 56.10      | 41.90 | 74.20    | 64.90    | 79.90      | 50.30      | 41.70      | 72.70 |
> | **WePE (Ours)** | **91.32**  | **87.59** | **77.41** | **98.79** | **93.16** | **64.30** | **84.58** | **83.73** | **93.89** | **86.10** | **77.15**   | **73.81**   | **60.91**   | **54.24** | **75.18**  | **68.10**  | **81.25**   | **34.11**   | **42.01**   | **73.59** |
>
> Table 5: Complexity and Efficiency Comparison for Online Inference
> | Metric | Original Direct Computation | LUT-based Hybrid Method | Traditional Learnable PE |
> | :--- | :--- | :--- | :--- |
> | Time Complexity | $\mathcal{O}(N \cdot C_{\text{wef}})$ | $\mathcal{O}(N \cdot d)$ | $\mathcal{O}(N \cdot d)$ |
> | Space Complexity | $\mathcal{O}(N \cdot d)$ | $\mathcal{O}(\text{Res}^2 + N \cdot d)$ | $\mathcal{O}(M_{\text{max}} \cdot d + N \cdot d)$ |
> | Dependence on Math Complexity | High (depends on series terms, summation limits, etc.) | None (decoupled after pre-computation) | None (entirely learned) |
> | Hardware Affinity | Low (complex arithmetic) | High (memory access & interpolation) | Very High (optimized lookup) |

---

> ### Author Response · Authors · 2025-11-30
> **Official Comment from Author 19480 in response to Reviewer Smeu’s Question 3**
>
> **Q3. The sensitivity of the model to ${\beta}$ and ${\gamma}$.**
>
>
>
> **In the revised manuscript study the sensitivity of the fine-tuning adaptation to the Fourier-like parameters $\beta$ and $\gamma$ (Appendix H.10, Figs. 20–21)**. Concretely, we use a ViT-B/16 backbone pretrained on ImageNet-21k and fine-tuned on CIFAR-100 with WePE, and replace the learnable $\beta$ and $\gamma$ by fixed scalars on a $5\times 5$ grid:
> $$
> \beta \in \{0.01, 0.05, 0.10, 0.20, 0.50\}, \qquad
> \gamma_{\text{scale}} \in \{0.01, 0.05, 0.10, 0.20, 0.50\}.
> $$
> For each of the 25 configurations, we fine-tune for 5 epochs with identical settings and report the best test accuracy.
> * Across all 25 settings, the best mean accuracy is $49.93\%$ with a standard deviation of $2.81$ and a coefficient of variation of $5.6\%$, the minimum and maximum being $40.29\%$ and $52.27\%$, respectively.
> * More importantly, in the practically relevant range $\beta \ge 0.05$ and $\gamma_{\text{scale}} \ge 0.05$ (which covers the values used in our main experiments), the best test accuracy lies between $50.28\%$ and $51.59\%$, i.e., within a narrow band of only $1.31$ percentage points.
> * The heatmap in Fig. 20 and slices in Fig. 21 show a broad plateau over this region: varying either $\beta$ or $\gamma_{\text{scale}}$ within $\{0.05,0.10,0.20,0.50\}$ leads only to mild fluctuations comparable to typical run-to-run noise.
>
> Although the Fourier-like approximation introduces the additional scalars $\beta$ and $\gamma$, this experiment shows that **WePE tends to be insensitive to these parameters**. In the broad practical range $\beta\in[0.05,0.50]$ and $\gamma_{\text{scale}}\in[0.05,0.50]$, the downstream accuracy varies by at most about $1.3$ percentage points, while larger drops are confined to extreme settings that are not used in practice. Combined with the fact that $\beta$ and $\gamma$ are learned automatically in our main models, this indicates that **the Fourier-like adaptation does not introduce a significant tuning burden and that the proposed WePE is robust to the specific choices of these parameters**.

---

> ### Author Response · Authors · 2025-11-30
> **Official Comment from Author 19480 in response to Reviewer Smeu’s Question 4**
>
> **Q4. Whether hyperparameters were tuned for each method, or uniformly fixed?**
>
> We clarify this question from the following aspects:
>
> * For baseline methods that provide directly comparable results under the same experimental setting (e.g., LieRE [2]), we simply take the reported numbers from their original papers as our reference.
>
> * For baselines whose original works offer explicit hyperparameter configurations and publicly available code (e.g., APE [3]), we strictly follow the recommended settings provided by the authors.
>
> * For methods whose original papers do not include results directly applicable to our specific setup (e.g., FoPE [4]), we train them under an identical training pipeline and optimization configuration. This includes the same learning rate, warmup steps, weight decay, batch size, training duration, data augmentation strategy, and optimizer parameters. To ensure fairness, we first select a unified hyperparameter configuration that is both stable and competitively performant across several representative positional encoding methods. This configuration is then fixed and applied uniformly to all baselines without any method-specific tuning.
>
> Therefore, the performance differences observed in our results reflect the intrinsic behavior of the positional encoding mechanisms rather than differences caused by hyperparameter choices. We have also clarified this content in **Appendix G.1**.

---

> ### Author Response · Authors · 2025-11-30
> **Official Comment by Author 19480**
>
> **References**
>
> [1] Exploring Plain Vision Transformer Backbones for Object Detection. ECCV, 2022
>
> [2] LieRE: Lie Rotational Positional Encodings. ICML, 2025
>
> [3] An Image is Worth 16x16 Words: Transformers for Image Recognition at Scale. ICLR, 2021
>
> [4] Fourier Position Embedding: Enhancing Attention's Periodic Extension for Length Generalization. ICML, 2025

---

### Official Review · Reviewer_b6eQ · 2025-11-01

**Soundness:** 3
**Presentation:** 3
**Contribution:** 3
**Rating:** 6
**Confidence:** 2

**Summary:**

This paper proposes Weierstrass Positional Encoding (WePE), a mathematically grounded 2D positional encoding for Vision Transformers.
Instead of using traditional sinusoidal or rotary encodings, the authors use the Weierstrass elliptic function, a doubly periodic complex function to map image coordinates onto the complex plane.
This design aims to preserve true 2D spatial continuity and provide better distance decay and relative-position properties.
Experiments on CIFAR-100, ImageNet, and VTAB show small but consistent accuracy gains over existing encodings.

**Strengths:**

Elegant mathematical formulation: the use of a doubly periodic complex function is theoretically appealing and fits the 2D geometry of images.

Continuous and resolution-independent: WePE naturally handles different image sizes without interpolation.

Empirical improvements: consistent accuracy gains and smoother attention maps in visualization.

**Weaknesses:**

Lack of broader context: while the mathematical formulation is elegant, the paper could better clarify how WePE conceptually differs from other periodic or complex-valued encodings (e.g., Fourier- or rotary-based).

Limited analysis: the paper mainly reports performance improvements but provides little analysis or visualization explaining why the proposed encoding helps attention behavior.

Generalization scope unclear: it remains uncertain whether WePE benefits generalize to larger-scale or non-vision tasks, since experiments are focused on a few image benchmarks.

**Questions:**

What is the computational cost of WePE compared to standard sine/cosine positional encodings?

Did the authors compare WePE to high-order Fourier or complex-valued positional encodings to isolate its specific benefit?

Is the improvement primarily from the double periodicity, or could similar results be achieved with a simpler 2D periodic basis?

---

> ### Author Response · Authors · 2025-11-30
> **Official Comment by Author 19480**
>
> We would like to thank you for your thoughtful review and constructive feedback. Below are our detailed responses to each comment, marked in red text in the revised manuscript. Please kindly refer to them.

---

> ### Author Response · Authors · 2025-11-30
> **Official Comment from Author 19480 in response to Reviewer b6eQ’s Question 1**
>
> **Q1. The conceptual differences between WePE and other positional encoding techniques.**
>
> **In the revised version, we explicitly expand this discussion in Appendix B**. Concretely, we distinguish three mainstream families of explicit positional encodings for ViTs: (i) learnable absolute embeddings [1], (ii) Sinusoidal Position Encoding [2] and their higher-order Fourier extensions (e.g., FoPE [3]), and (iii) RoPE [4] and their complex-valued or group-theoretic extensions (e.g., LieRE [5], RoPE-Mixed [6], Multimodal Rotary Position Embedding (M-RoPE) [7], etc.). Indeed, according to Euler's formula, the foundation of RoPE still lies in Sinusoidal Position Encoding. Its essence resides in leveraging Euler's formula to map positional information into vector rotations, making RoPE fundamentally a geometric extension of the traditional Sinusoidal encoding.
>
> However, WePE is conceptually distinct from all two or three families mentioned above, even though it is also built upon periodic and complex-valued functions. To our knowledge, WePE is the first 2D positional encoding scheme designed for ViTs that is genuinely constructed on the complex plane. Rather than composing multiple 1D sinusoidal bands along the flattened token index or applying block-wise rotary phases in the hidden space, WePE is formulated as a genuinely 2D positional function on the complex plane. It maps the 2D patch lattice to a complex lattice and evaluates a Weierstrass elliptic function with an intrinsic doubly periodic structure. As a result, the 2D geometry of the image grid is an inherent property of the encoding itself, rather than an artifact of the 1D serialization or separable 1D Fourier bases. Through the elliptic addition formula, absolute positional codes in WePE are algebraically linked to relative displacements, providing a tight coupling between absolute and relative position at the function level rather than relying solely on phase differences in Fourier space. This makes WePE the "fourth approach", standing apart from the three mainstream positional encoding schemes currently available.

---

> ### Author Response · Authors · 2025-11-30
> **Official Comment from Author 19480 in response to Reviewer b6eQ’s Question 2**
>
> **Q2. Further analysis or visualization explaining why WePE helps attention behavior.**
>
> **In the revised manuscript, we have added a new section Appendix H.8 together with a series of visualizations in Figs. 13--18. These analyses validate that WePE enhances both the accuracy and the discriminatory power of the learned attention**.
>
> * First, **Figure 13** compares self-attention maps of WePE with that of the baseline APE [1] at Layer 6, Head 0 for several query locations. WePE exhibits more pronounced spatial locality: attention mass is concentrated on compact neighborhoods around the query, while the baseline distributes attention more diffusely.
> * **Figure 14** shows positional similarity matrices, where WePE produces a checkerboard pattern that directly reflects the doubly periodic lattice induced by the Weierstrass function, whereas the baseline exhibits no clear geometric structure.
> * **Figure 15** plots average attention weight as a function of patch distance for multiple layers. In shallow layers, WePE displays a much steeper decay with distance than the baseline, indicating that it has a very good ability to learn local bias. This decay gradually relaxes in deeper layers, allowing global integration while still preserving the inductive prior.
> * **Figure 16** further quantifies focus via Shannon entropy of attention distributions across layers. It can be seen that WePE consistently yields lower entropy in early and mid layers (e.g., $4.6$--$4.9$ vs. $5.1$--$5.3$), corresponding to more concentrated and less noisy attention maps.
> * **Figure 17** analyzes the effective attention range by measuring the radius required to accumulate a fixed proportion (90%) of attention mass. With WePE, this radius is smaller in the first layers (more compact receptive fields) and then gradually expands, resembling the multi-scale behavior of CNNs but induced purely by positional encoding rather than architecture changes.
> * Finally, **Figure 18** visualizes patch embeddings using t-SNE. It can be observed that under WePE, patches that are spatially close in the image form tighter clusters in feature space, while the baseline shows weaker spatial organization.
>
> Taken together, these new analyses provide concrete evidence that WePE’s gains stem from a mathematically grounded geometric inductive bias. This bias encourages local, coherent attention at early layers, maintains a distance-aware decay profile across layers, and leads to spatially structured feature representations.

---

> ### Author Response · Authors · 2025-11-30
> **Official Comment from Author 19480 in response to Reviewer b6eQ’s Question 3**
>
> **Q3. The generalization scope of WePE's benefits.**
>
>
>
> In our paper, we have reported various experimental results on datasets such as CIFAR-100, COCO, and ImageNet-1k. We also recognize the value of conducting experiments on larger-scale datasets such as ImageNet-21K or JFT.
>
> * However, we must state that, **the original ImageNet-21K release does not provide an official train/validation split and exhibits severe class imbalance, existing works therefore construct their own processed variants (e.g., ImageNet-21K-P with filtered classes and custom splits), which are not yet universally adopted. Consequently, most prior works treat ImageNet-21K primarily as a pre-training corpus and only report downstream transfer performance rather than results evaluated directly on the raw ImageNet-21K labels**.
> * **Our Section 3.3 follows this well-established convention**. We believe that running a single, non-standard split would make it difficult to compare fairly with prior arts or works, while substantially increasing computational cost without changing the core behavior of the positional encoding.
> * Similarly, **JFT-300M is a proprietary internal dataset from Google and is not publicly accessible to the broader research community**.
> * As WePE does not alter the backbone architecture and only replaces the positional encoding, we believe its inductive-bias advantages to transfer to larger-scale pretraining regimes independently.
>
> * While Transformer architectures are broadly applicable, ViTs were specifically designed as image backbones, and the overwhelming majority of existing works validate them on vision benchmarks. Although studies have adapted ViT-like architectures to non-vision domains (e.g., audio spectrograms or time-series), these adaptations often require domain-specific modifications and operate under distinct evaluation protocols. Consequently, focusing our analysis on standard visual benchmarks is consistent with the prevailing practice for evaluating general-purpose ViT backbones. We also wish to clarify that WePE is explicitly designed to exploit the correspondence between the 2D image patch lattice and the doubly periodic structure of the Weierstrass elliptic function. Extending this framework to non-vision tasks (e.g., 1D linguistic sequences or 3D scientific data) would necessitate reformulating the underlying geometric basis rather than a straightforward application. While this represents a promising research direction, it constitutes a distinct line of future work. We are keen to explore variants of WePE adapted for scenarios beyond computer vision in the future.

---

> ### Author Response · Authors · 2025-11-30
> **Official Comment from Author 19480 in response to Reviewer b6eQ’s Question 4**
>
> **Q4. The computational cost of WePE.**
>
>
>
> The computational cost of WePE is higher than that of standard sine/cosine positional encodings. However, fortunately, as we detailed in **Appendix F**, we propose a hybrid implementation scheme based on a lookup table: All evaluations of the Weierstrass elliptic function are performed once offline to build a high–resolution look-up table (LUT).
>
> During training and inference, online positional encoding reduces to: (i) computing normalized $(u,v)$ coordinates for each of the $N=H\times W$ patches, (ii) a hardware–accelerated bilinear interpolation from the LUT, and (iii) a linear projection to the $d$-dimensional embedding space. As detailed in our complexity analysis and **Table 5** in **Appendix F**（For ease of reference, we also reproduce the original table here）, this yields an online time complexity of $\mathcal{O}(N\cdot d)$, which is the same asymptotic cost as standard 2D sine/cosine or learnable grid positional encodings (one constant–time operation and one linear projection per patch). The additional memory cost is a single pre-computed LUT of size $\text{Res}^2\times 4$ (we use $\text{Res}=256$, corresponding to ≈ 1--2\,MB in `float32`), which is negligible compared to the ViT parameters and activation tensors. **Empirically, in our PyTorch implementation, we did not observe any noticeable slowdown when replacing standard sine/cosine (or learnable) positional encodings with WePE**. **We also prove in Appendix E that the resulting computational precision error introduced by this operation is virtually negligible**. **We therefore believe that WePE provides its performance gains with essentially the same computational cost as standard sine/cosine positional encodings, while adding only a small constant memory overhead for the LUT**. To facilitate reproducibility and to make the cost trade-offs fully transparent, we plan to open-source both the pre-computed LUTs used in our experiments and the scripts that generate them from the exact WePE implementation.
>
> Table 5: Complexity and Efficiency Comparison for Online Inference
> | Metric | Original Direct Computation | LUT-based Hybrid Method | Traditional Learnable PE |
> | :--- | :--- | :--- | :--- |
> | Time Complexity | $\mathcal{O}(N \cdot C_{\text{wef}})$ | $\mathcal{O}(N \cdot d)$ | $\mathcal{O}(N \cdot d)$ |
> | Space Complexity | $\mathcal{O}(N \cdot d)$ | $\mathcal{O}(\text{Res}^2 + N \cdot d)$ | $\mathcal{O}(M_{\text{max}} \cdot d + N \cdot d)$ |
> | Dependence on Math Complexity | High (depends on series terms, summation limits, etc.) | None (decoupled after pre-computation) | None (entirely learned) |
> | Hardware Affinity | Low (complex arithmetic) | High (memory access & interpolation) | Very High (optimized lookup) |

---

> ### Author Response · Authors · 2025-11-30
> **Official Comment from Author 19480 in response to Reviewer b6eQ’s Question 5**
>
> **Q5. Did the authors compare WePE to high-order Fourier or complex-valued positional encodings to isolate its specific benefit?**
>
> **Yes, our experiments already include both high-order Fourier and complex-valued positional encodings as baselines, and WePE consistently outperforms them under identical model capacity and training settings**.
>
> * **High-order Fourier encodings.** **Table 1** in **Section 3.2** (For ease of reference, we also reproduce the original table here) compares WePE against Fourier Position Embedding (FoPE) [3], which explicitly uses multiple Fourier harmonics to encode position, as well as against standard sinusoidal and absolute learnable PEs. On CIFAR-100 (100% data, 120 epochs), WePE reaches 63.78% top-1 accuracy, while FoPE and RoPE obtain 57.70% and 57.29%, respectively, and sinusoidal PE achieves 51.99%. All variants use the same ViT-Ti backbone and embedding dimensionality, so the improvement cannot be explained by a higher parameter count.
> * **Complex-valued / rotation-based encodings.** **Table 2** in **Section 3.2** (For ease of reference, we also reproduce the original table here) further evaluates advanced complex/rotation-based position encodings: LieRE_8 [5], LieRE_64 [5] and RoPE-Mixed [6]. These methods operate in a complex-valued or group-theoretic Fourier space and are specifically designed to capture rich relative-position structure. Across CIFAR-100 with 20--90% of the training data and on the ImageNet-1K dataset, WePE is consistently the best performer, e.g., on the ImageNet-1K dataset, WePE attains 70.10% top-1 accuracy versus 69.60% (LieRE_8), 69.30% (LieRE_64), and 68.80% (RoPE-Mixed).
>
> In summary, these comparisons show that WePE's gains persist even when contrasted with powerful high-order Fourier and complex-valued encodings under controlled conditions. This suggests that the superiority of WePE does not solely stem from the use of more expressive functional features, but rather from its unique elliptic function geometry, which provides a better inductive bias for two-dimensional visual tasks.
>
> Table 1: CIFAR-100 (100% dataset), 120 epochs Top-1 accuracy (%)
> | Method | **WePE (Ours)** | Absolute PE | RoPE | FoPE | Sinusoidal PE |
> | :--- | :--- | :--- | :--- | :--- | :--- |
> | Accuracy | **63.78** | 56.46 | 57.29 | 57.70 | 51.99 |
>
> Table 2: Top-1 accuracy (%) on CIFAR-100 and ImageNet-1k, trained for 200 epochs
> | Dataset | Fraction | **WePE (Ours)** | LieRE$_{8}$ | LieRE$_{64}$ | RoPE-Mixed | AS2DRoPE | APE |
> | :--- | :--- | :--- | :--- | :--- | :--- | :--- | :--- |
> | CIFAR-100 | 20% | **46.36** | 45.42 | 44.44 | 44.48 | 39.14 | 39.80 |
> | CIFAR-100 | 40% | **56.81** | 54.68 | 54.64 | 55.14 | 50.53 | 49.90 |
> | CIFAR-100 | 60% | **63.38** | 62.04 | 62.90 | 61.56 | 58.58 | 56.83 |
> | CIFAR-100 | 90% | **68.96** | 67.72 | 68.36 | 67.00 | 62.59 | 62.76 |
> | ImageNet-1k | 100% | **70.10** | 69.60 | 69.30 | 68.80 | 64.40 | 66.10 |

---

> ### Author Response · Authors · 2025-11-30
> **Official Comment from Author 19480 in response to Reviewer b6eQ’s Question 6**
>
> **Q6. Ablation study for the double periodicity.**
>
> The reviewer asks whether WePE's improvements could also be achieved by using a simpler 2D periodic basis. Our view is that the gains do not stem merely from introducing “some” periodic component, but from the specific geometric properties of the elliptic kernel used in WePE, namely its doubly periodic lattice structure together with built-in distance decay and the addition formula. Existing baselines already support this: several positional encodings with sinusoidal or rotational periodic components (e.g., RoPE [4], and LieRE [5]) do not reproduce the same geometric behavior or the same empirical gains as WePE.
>
> For clarification, we analyze the implicit periodicity of LiePE. Let $p = (x_1,\dots,x_d)\in\mathbb{R}^d$ denote a continuous coordinate (e.g., an image patch), and let $\{A_k\}_{k=1}^d$ be learnable skew-symmetric generators with $A_k^\top=-A_k$. LieRE associates to $p$ the transformation
>
> $$
> R(p)=\exp\!\Big(\sum_{k=1}^d x_k A_k\Big),
> $$
>
> which is used to rotate the query and key vectors. For intuition, consider the one-dimensional slice where all coordinates except $x_j$ are fixed, and define $A:=A_j$ and $R(x_j):=\exp(x_j A)$. By standard linear algebra, any real skew-symmetric matrix $A$ admits an orthogonal decomposition
> $$
> U^\top A U = \text{blockdiag}\left(
>   \begin{bmatrix} 0 & -\omega_1 \\ \omega_1 & 0 \end{bmatrix},
>   \dots,
>   \begin{bmatrix} 0 & -\omega_m \\ \omega_m & 0 \end{bmatrix},
>   0
> \right)
> $$
>
> for some real frequencies $\omega_\ell\ge 0$ and orthogonal $U$. Exponentiating this normal form yields
>
> $$
> R(x_j) = \exp(x_j A) = U \mathrm{blockdiag} \left( \begin{bmatrix} \cos(\omega_1 x_j) & -\sin(\omega_1 x_j) \\ \sin(\omega_1 x_j) & \cos(\omega_1 x_j) \end{bmatrix}, \dots, \begin{bmatrix} \cos(\omega_m x_j) & -\sin(\omega_m x_j) \\ \sin(\omega_m x_j) & \cos(\omega_m x_j) \end{bmatrix}, I \right) U^\top.
> $$
>
> Thus every scalar entry of $R(x_j)$ is a finite linear combination of $\cos(\omega_\ell x_j)$ and $\sin(\omega_\ell x_j)$, meaning that along the $j$-th axis, LiePE exhibits an *implicit trigonometric periodicity*. Consequently, any attention logit of the form
>
> $$
> Q_i^\top R(p_i)^\top R(p_j)\, K_j
> $$
>
> inherits this multi-frequency sinusoidal structure as a function of $x_j$.
>
> This analysis shows that **although LieRE also exhibits periodic behavior along both coordinate axes (i.e., it has two axis-wise periodic basis functions), these periodic components are separable and do not interact to form a coupled two-dimensional periodic structure**. Consequently, LieRE cannot reproduce the genuinely 2D bi-periodic lattice topology provided by WePE's elliptic kernel, nor the resulting empirical gains.
>
> In contrast, **WePE is constructed from the doubly periodic Weierstrass elliptic function, whose periods are intrinsically two-dimensional and inseparable. This yields a genuinely 2D periodic topology together with distance-aware decay and the addition formula—properties that cannot be reproduced by simply stacking 1D sinusoidal bases or by the implicit axis-wise periodicity of LieRE**. Empirically, this richer geometric structure leads to consistent improvements across both pre-training and fine-tuning settings.
>
> In addition, **most simpler two-dimensional periodic positional encodings are separable constructions obtained by combining independent 1D sinusoids along each axis** (e.g., $\sin x+\sin y$, $(\sin x,\cos x,\sin y,\cos y)$). Although these schemes exhibit periodicity in both dimensions, they fundamentally lack internal interaction within the position function itself, thereby precluding effective two-dimensional information coupling. Consequently, **the resulting position structure essentially constitutes a simple concatenation of two one-dimensional patterns**, rather than a truly meaningful two-dimensional periodic function, as exemplified by WePE. However, **the classical *Weierstrass Elliptic Function Representation Theorem* establishes that, up to trivial affine transformations, the Weierstrass $\wp$-function is the unique non-constant meromorphic function on $\mathbb{C}$ that is genuinely doubly periodic**. This fundamental mathematical fact implies that no simpler 2D periodic basis can replicate the same doubly periodic lattice structure or its associated geometric properties.
>
> Furthermore, our ablation study (Section 3.4) shows that removing any of the carefully designed components of WePE, including the derivative term $\wp'(z)$, the real/imaginary decomposition, the adaptive scaling parameters, or the lemniscatic lattice itself, consistently degrades the performance.
>
> We posit, therefore, that **these achievements do not merely stem from the introduction of bi-periodicity in an abstract sense, but rather are attributable to the incorporation of a deeper, specific, and comprehensive advantage rooted in their geometric properties**.

---

> ### Author Response · Authors · 2025-11-30
> **Official Comment by Author 19480**
>
> **References**
>
> [1] An Image is Worth 16x16 Words: Transformers for Image Recognition at Scale. ICLR, 2021
>
> [2] Attention Is All You Need. NeurIPS, 2017
>
> [3] Fourier Position Embedding: Enhancing Attention's Periodic Extension for Length Generalization. ICML, 2025
>
> [4] RoFormer: Enhanced Transformer with Rotary Position Embedding. Neurocomputing, 2024
>
> [5] LieRE: Lie Rotational Positional Encodings. ICML, 2025
>
> [6] Rotary Position Embedding for Vision Transformer. ECCV, 2024
>
> [7] Qwen-VL: A Versatile Vision-Language Model for Understanding, Localization, Text Reading, and Beyond. ICLR，2024

---

### Official Review · Reviewer_DryE · 2025-11-01

**Soundness:** 3
**Presentation:** 2
**Contribution:** 3
**Rating:** 6
**Confidence:** 2

**Summary:**

To overcome the limitation that the standard formulation of Vision Transformers (ViTs) uses one-dimensional positional encodings which disrupt the intrinsic two-dimensional spatial geometry of images, the paper proposes Weierstrass Positional Encoding for Vision Transformers (WePE), a mathematically principled method that preserves the intrinsic spatial structure of images through complex-domain mapping based on the Weierstrass elliptic function. By encoding 2D coordinates on the complex plane and exploiting the doubly periodic properties of the function, WePE provides a continuous, resolution-agnostic, and geometrically consistent positional representation. Extensive experiments demonstrate that this approach consistently improves performance over conventional positional encodings while introducing negligible computational or memory overhead.

**Strengths:**

originality:
 This paper proposes a novel definition and solution to address the long-standing challenge of positional encoding in Vision Transformers (ViTs). By introducing a Weierstrass elliptic positional encoding (WePE) framework grounded in complex analysis, it provides a fresh, mathematically principled perspective on preserving 2D spatial geometry. Unlike heuristic sinusoidal or rotary encodings, WePE employs the doubly periodic Weierstrass elliptic function and its derivative to construct a compact, continuous, and resolution-invariant spatial representation.
significance:
 The positional encoding problem in ViTs is fundamental to numerous vision tasks—ranging from visual grounding and visual reasoning to detection and segmentation. The proposed WePE framework provides a general, plug-and-play solution with provable geometric properties, distance-decay behavior, and strong empirical gains across benchmarks. It could potentially broadly influence both theoretical and applied research communities, setting a new direction for geometry-aware representation learning in vision transformers.

**Weaknesses:**

The paper is technically rich but somewhat difficult to follow, particularly for readers who are not deeply familiar with complex analysis or elliptic function theory. While the mathematical rigor is commendable, the exposition occasionally prioritizes formal derivations over intuitive explanations. As a result, the connection between the underlying mathematical properties (e.g., periodicity, addition law, distance decay) and their concrete implications for Vision Transformer performance is not always clearly articulated.

**Questions:**

Boundary Conditions & Aspect Ratios. How does the method behave for non-square inputs and unusual aspect ratios? Provide results where height/width change substantially and discuss any need to retune lattice parameters.

Failure Cases & Diagnostics. The paper would benefit from explicit failure analyses (qualitative heatmaps, attention distance histograms, error vs. spatial separation). Where does the method underperform, and why?

---

> ### Author Response · Authors · 2025-11-30
> **Official Comment by Author 19480**
>
> We would like to thank you for your thoughtful review and constructive feedback. Below are our detailed responses to each comment, marked in blue text in the revised manuscript. Please kindly refer to them.

---

> ### Author Response · Authors · 2025-11-30
> **Official Comment from Author 19480 in response to Reviewer DryE’s Question 1**
>
> **Q1. Further Explanations**
>
> **We added an intuitive and vivid description of the three-dimensional shape of the Weierstrass elliptic function at Section 1**: “From a three-dimensional perspective, the Weierstrass $\wp$-function resembles an array of identical volcano-like structures arranged regularly across the plane, where the sharp peak of each “volcano” corresponds to a second-order pole of the function. These peaks rise steeply to infinity at the lattice points, while the surface between them exhibits smooth, doubly periodic undulations. Each “volcano” is embedded within the parallelogram cell spanned by its two fundamental half-periods $\omega_{1}$ and $\omega_{3}$, which together form the period lattice of the Weierstrass elliptic function.”
>
> **We added an explanation at the beginning of the third paragraph in Section 2.4 regarding why positional encodings require a “long-term decay” property**: “In self-attention, interaction strength is determined by the similarity between token representations. If the interaction strength after adding positional encodings does not exhibit a decaying trend with spatial separation, the model cannot differentiate local relationships from long-range ones. This contradicts the intrinsic structure of natural images, where nearby pixels and patches are far more correlated than distant ones. A built-in distance–decay profile therefore supplies the model with a locality-aware inductive bias, enabling attention to prioritize meaningful local interactions while still permitting global reasoning when necessary.”
>
> **We have additionally included Appendix C to explain how the underlying mathematical properties of WePE positively influence the performance of Vision Transformers**:
>
> **Intuition of Periodicity.**
> Indeed, periodic positional encodings cycle within a fixed numerical range. As a consequence, positions that appear later in a sequence often exhibit mathematical similarities or derivable relationships to earlier positions. During training, the model learns to interpret and process this recurring periodic pattern.Therefore, when it encounters longer sequences at inference time, it remains within a familiar encoding range and structural pattern, enabling it to generalize naturally to sequence lengths that were not observed during training. Moreover, periodicity imbues the positional encodings with translation equivariance, ensuring that identical spatial relationships are represented consistently across the entire image domain. This attribute is inherently absent in the standard Transformer architecture, yet it is of paramount importance for vision tasks.
>
> **Strengthening Inductive Biases.**
> Beyond directly providing the geometric advantage of translation equivariance, WePE also indirectly strengthens the model's ability to learn inductive biases, such as scale, rotation, viewpoint, and illumination invariance:
>
> * **Scale Invariance.** WePE is constructed based on continuous functions firstly, and thus exhibits good scale stability. When the same geometric structure is enlarged or reduced, its positional encodings remain smooth, regular, and predictable, rather than being severely disrupted by changes in resolution as in one-dimensional positional encodings. This property makes it easier for the model to learn invariances related to scale.
> * **Rotation Invariance.** The $\wp$ function used in our method is generated from an orientation-consistent doubly periodic lattice, which ensures that WePE produces encodings for different directions from a single two-dimensional continuous structure. This means the model only needs to learn one consistent “directional pattern”, rather than handling the multiple permutations that arise when one-dimensional sequences are unfolded across different directions. Consequently, compared with one-dimensional positional encodings, WePE is more conducive to learning invariances related to rotation.
> * **Viewpoint Invariance.** Viewpoint changes induce nonlinear geometric deformations on the two-dimensional patch grid, while in a one-dimensional sequence, such local deformations are mapped to chaotic index rearrangements, thereby corrupting structural information. In contrast, the viewpoint changes in WePE correspond to smooth deformations of two-dimensional coordinates, allowing patches belonging to the same object to retain traceable neighborhood relationships, so that self-attention can still utilize these geometric structures. Based on the addition formula, the relative positional encoding further ensures that even if Euclidean distances change, the structural information regarding “which patches are neighbors and which are far apart” can still be preserved.

---

> ### Author Response · Authors · 2025-11-30
> **Official Comment from Author 19480 in response to Reviewer DryE’s Question 1**
>
> * **Illumination Invariance.** Finally, in learnable APE, each position is represented by an independent vector, and therefore often absorbs dataset-specific correlations between position and appearance. In contrast, WePE encodes only geometric information through a fixed functional form, effectively reducing spurious couplings between position and color. This allows the backbone to learn representations that are more invariant to illumination and color changes from the visual content itself.
>
> Taken together, these inductive biases make WePE a compelling and highly effective replacement for traditional positional encodings in ViTs.
>
> **Finally, we have added Figure 2 to the main textwe additionally include Figure 2, an illustrative diagram highlighting the three key properties of WePE**: 1）relative position modeling via the addition formula, 2）distance-dependent decay, and 3）doubly periodic structure.
>
> With the inclusion of these additions, we posit that this work will be accessible to a broader readership across diverse domains, thereby facilitating enhanced readability.

---

> ### Author Response · Authors · 2025-11-30
> **Official Comment from Author 19480 in response to Reviewer DryE’s Question 2**
>
> **Q2. Experiments on non-square inputs and unusual aspect ratios.**
>
> **We have added a new experiment in Appendix H.9 to evaluate the performance of WePE under extreme aspect ratios, specifically on highly non-square inputs**. Taking CIFAR-100 as an example, we adopt the same backbone and global image scale while keeping all hyperparameters strictly identical. In the experiments, we modify the input resolution from scratch: $224\times224$ (baseline, $1{:}1$), $224\times448$ (“wide”, $2{:}1$), ......, $112\times448$ (“very wide”, $4{:}1$), to induce different patch-grid aspect ratios. Besides, we train all models from scratch. These configurations scale height or width by a factor of 2--4, approaching or even exceeding the most extreme cases that real-world deployments may encounter.
>
> We first conducted full 120-epoch training runs for the standard square input ($224\times224$, $1{:}1$) and the most extreme aspect ratio ($112\times448$, $4{:}1$). As shown in **Figure 19** in **Appendix H.9**, both the training and validation accuracy are already very close to their final converged values by epoch 20, with subsequent training bringing only minor fluctuations and marginal improvements. This indicates that extending the training schedule does not alter the qualitative trends or the relative performance across different aspect ratios. Therefore, for each configuration, we report the best validation accuracy obtained at 10 and 20 epochs.
>
> * As shown in **Table 7**, all non-square configurations already surpass the square baseline (32.82%) after 10 epochs, achieving accuracy between 36.61% and 42.72%.
> * After 20 epochs, models trained with moderate aspect ratios ($2{:}1$, $1{:}2$, $1{:}3$) reach 51.86%, 52.28%, and 52.83% respectively, which are comparable to or slightly higher than the 51.26% obtained with the $1{:}1$ input.
> * Only the extreme $4{:}1$ setting exhibits a performance drop, which is mainly attributed to the intrinsic sensitivity of ViT architectures to highly anisotropic patch grids [1][2][3].
>
> **These findings demonstrate that WePE remains stable under non-square inputs. Moderate aspect-ratio changes have negligible impact on performance; even under extreme ratios, the degradation is smooth rather than catastrophic**.
>
> Regarding the reviewer’s question on whether lattice parameters require retuning, we also report in **Table 7** of **Appendix H.9** the learned elliptic-lattice parameter $\alpha_{\text{learn}}$ for each configuration. Across all aspect ratios and both training budgets, despite height/width ratios varying by up to a factor of four, $\alpha_{\text{learn}}$ consistently stays within the narrow interval $[0.84, 0.90]$. Importantly, we do not manually adjust any lattice parameter when changing input shapes. $\alpha_{\text{learn}}$ is automatically optimized through backpropagation under the same initialization and regularization settings used in the main experiments. **This supports our claim regarding the robustness of WePE: the elliptic lattice naturally adapts to new patch-grid geometries without requiring any hand-crafted reparameterization**.
>
>
> Table 7: Performance of WePE under different aspect ratios. Baseline uses square input resolution.
> | Experiment | Size | Aspect Ratio | Acc (10 ep) | $\alpha_{\text{learn}}$ (10 ep) | Acc (20 ep) | $\alpha_{\text{learn}}$ (20 ep) |
> | :--- | :--- | :--- | :--- | :--- | :--- | :--- |
> | baseline | 224×224 | 1:1 | 32.82% | — | 51.26% | — |
> | wide | 224×448 | 2:1 | 42.03% | 0.8647 | 51.86% | 0.8730 |
> | tall | 448×224 | 1:2 | 42.61% | 0.8792 | 52.28% | 0.9029 |
> | very tall | 672×224 | 1:3 | 42.72% | 0.8605 | 52.83% | 0.8533 |
> | very wide | 112×448 | 4:1 | 36.61% | 0.8449 | 46.77% | 0.8379 |

---

> ### Author Response · Authors · 2025-11-30
> **Official Comment from Author 19480 in response to Reviewer DryE’s Question 3**
>
> **Q3. Failure Cases & Diagnostics.**
>
> **In the revised version, we have added failure analyses in Appendix H.12, including the reviewer-requested (i) classification error rate on the test set vs. spatial separation curves, (ii) qualitative attention heatmaps, and (iii) attention–distance histograms (Figs. 24–26)**.
>
> Concretely, we define a spatial separation score $S(x)$ as the mean of pairwise Euclidean distance between the top-$K$ attended patches on the $14\times 14$ lattice.
> * **Figure 24** shows that the vast majority of test samples ($\approx 98.8\%$) fall into a mid-range regime $S\!\in[2,10]$, where the error rate remains low and stable (between $0.31$ and $0.38$) and the trend is nearly flat. This indicates that WePE handles well the common case where discriminative evidence spans multiple mid-range patches.
> * In contrast, only a small fraction of images ($\approx 1.2\%$) exhibit very large separation scores $S>10$, and in this high-separation regime the error rate rises sharply and some bins approach an error close to $1.0$. These are the failure cases where the discriminative cues are either extremely fragmented over many distant patches or require unusually long-range integration of visual evidence.
>
> To better understand these behaviors,**Figure 25** provides qualitative heatmaps from the last transformer layer.
> * For correctly classified images, WePE produces structured and coherent attention patterns: high responses concentrate on semantically meaningful object parts (e.g., head and torso of an animal), while the background receives very low attention, often forming several mid-range clusters that cover multiple object parts.
> * This type of failure case, however, reveal complementary patterns. When the evidence is extremely diffuse or entangled with clutter, attention becomes fragmented or drifts to salient but class-irrelevant regions. In other cases, it collapses onto a single small patch that does not cover the full object.
>
> Finally, **Figure 26** reports an attention–distance histogram, showing that the averaged attention weight per patch pair exhibits a clear global negative trend as a function of Euclidean distance, confirming that WePE indeed imposes a distance-aware inductive bias. The decay is gentle rather than abrupt, which explains why the model can still exploit non-negligible long-range attention, but also why the current global distance-decay profile is not optimal for rare images that demand extremely long-range reasoning or, conversely, highly localized cues.
>
> We explicitly discuss these shortcomings and outline future work on adaptive and multi-scale distance priors in **Appendix H.12**.

---

> ### Author Response · Authors · 2025-11-30
> **Official Comment by Author 19480**
>
> **References**
>
> [1] Swin Transformer: Hierarchical Vision Transformer using Shifted Windows. ICCV, 2021
>
> [2] Rethinking Spatial Dimensions of Vision Transformers. ICCV, 2021
>
> [3] Training Data-Efficient Image Transformers & Distillation Through Attention. ICML, 2021

---

### Note · Program_Chairs · 2026-01-17
**Submission Desk Rejected by Program Chairs**

The following references in this submission do not refer to real documents and/or have major errors in bibliographic information:

 Ziyu Wu, Qing Han, Zehuan Lin, Dong Chen, Songfang Han, Shilei Wen, and Errui Ding. Rethinking and improving relative position encoding for vision transformer. arXiv preprint arXiv:2107.14222, 2021.